# Exposing Outlier Exposure: What Can Be Learned From Few, One, and Zero Outlier Images

**Philipp Liznerski**[*]                                          *liznerski@cs.uni-kl.de*
*TU Kaiserslautern*

**Lukas Ruff**[*]                                                *lukas.ruff@aignostics.com*
*Aignostics, Berlin*

**Robert A. Vandermeulen**[*]                                  *vandermeulen@tu-berlin.de*
*ML Group TU Berlin, Berlin Institute for the Foundations of Learning and Data*

**Billy J. Franks**                                              *franks@cs.uni-kl.de*
*TU Kaiserslautern*

**Klaus-Robert Müller**                                  *klaus-robert.mueller@tu-berlin.de*
*Google Research, Brain Team, ML Group TU Berlin, MPII, and Korea University*

**Marius Kloft**                                                *kloft@cs.uni-kl.de*
*TU Kaiserslautern*

**Reviewed on OpenReview:** *https://openreview.net/forum?id=3v78awEzyB*

## Abstract

Due to the intractability of characterizing *everything* that looks unlike the normal data, anomaly detection (AD) is traditionally treated as an unsupervised problem utilizing only normal samples. However, it has recently been found that unsupervised image AD can be drastically improved through the utilization of huge corpora of random images to represent anomalousness; a technique which is known as *Outlier Exposure*. In this paper we show that specialized AD learning methods seem unnecessary for state-of-the-art performance, and furthermore one can achieve strong performance with just a small collection of Outlier Exposure data, contradicting common assumptions in the field of AD. We find that standard classifiers and semi-supervised one-class methods trained to discern between normal samples and relatively few random natural images are able to outperform the current state of the art on an established AD benchmark with ImageNet. Further experiments reveal that even *one* well-chosen outlier sample is sufficient to achieve decent performance on this benchmark (79.3% AUC). We investigate this phenomenon and find that one-class methods are more robust to the choice of training outliers, indicating that there are scenarios where these are still more useful than standard classifiers. Additionally, we include experiments that delineate the scenarios where our results hold. Lastly, no training samples are necessary when one uses the representations learned by CLIP, a recent foundation model, which achieves state-of-the-art AD results on CIFAR-10 and ImageNet in a zero-shot setting.

## 1 Introduction

Anomaly detection (AD) (Chandola et al., 2009) is the task of determining whether a sample is anomalous compared to a corpus of data. Recently there has been a great interest in developing novel deep methods for

---

[*] equal contribution.
Our code is available at: `https://github.com/liznerski/eoe`.
Part of this work has been presented in the ICML 2021 UDL Workshop (Ruff et al., 2021b).

AD (Ruff et al., 2021a; Pang et al., 2021). Most prior work performs AD in an *unsupervised* way utilizing only an unlabeled corpus of mostly normal data (Golan & El-Yaniv, 2018; Hendrycks et al., 2019b; Bergman et al., 2020; Tack et al., 2020). While AD can be interpreted as a classification problem of "normal vs. anomalous," it is classically treated as an unsupervised problem due to the rather tricky issue of finding or constructing a dataset that captures *everything different* from the normal dataset.

One often has, in addition to normal data, access to some data which is known to be anomalous. Hendrycks et al. (2019a) noted that, for an image AD problem, one has access to a virtually limitless amount of random natural images from the internet that are presumably not normal. They term the utilization of such auxiliary data *Outlier Exposure* (OE). Many top-performing AD methods on standard image AD benchmarks utilize tens of thousands of OE samples combined with self-supervised learning (Hendrycks et al., 2019b) or transfer learning (Reiss et al., 2021; Deecke et al., 2021) to achieve state-of-the-art detection performance.

For clarity, we here delineate three basic approaches to anomaly detection:
- *Unsupervised*: These are methods trained on unlabeled data that is assumed to be mostly normal. This is the classic and most common approach to AD.
- *Unsupervised OE*: These are adaptations of unsupervised methods that incorporate auxiliary OE data that is not normal. Elsewhere this is also called "semi-supervised" AD (Görnitz et al., 2013; Ruff et al., 2020).
- *Supervised OE*: This indicates standard classification methods trained to discern between normal data and an auxiliary OE dataset that is not normal.

Using unsupervised OE rather than supervised OE to discern between the normal data and OE samples seems intuitive since the presented anomalies likely do not completely characterize "anomalousness." Figure 1 illustrates this classic intuition and highlights the differences between these approaches on a 2D toy dataset. The benefits of unsupervised OE when incorporating (a few specific) known anomalies has also been observed in previous works (Tax, 2001; Görnitz et al., 2013; Ruff et al., 2020).

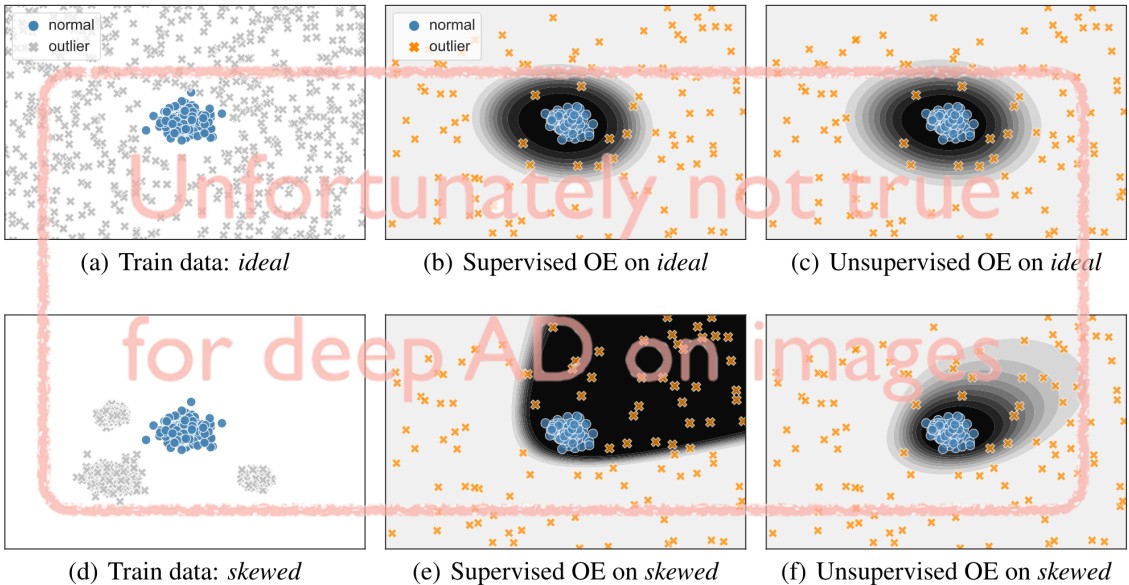

(a) Train data: *ideal*    (b) Supervised OE on *ideal*    (c) Unsupervised OE on *ideal*

(d) Train data: *skewed*    (e) Supervised OE on *skewed*    (f) Unsupervised OE on *skewed*

Figure 1: This figure visualizes a deceptively reasonable intuition that one might have for deep AD. It shows the decision boundaries of a supervised OE method and an unsupervised OE method on two toy datasets: *ideal* (a–c) and *skewed* (d–f). The skewed scenario occurs naturally when there are not enough OE samples to cover the ambient space, the data dimensionality is high, or the OE data is clustered. Unsupervised OE (c + f) learns a compact decision region for the normal class that generalizes well in both scenarios. A supervised OE approach (b + e), on the other hand, learns a decision region that generalizes well in the *ideal* case where the outlier training data is fully representative, but not in the *skewed* case. While this intuition is true for shallow AD settings, like in this 2D toy example (a–f), our results suggest that a deep approach does not follow these phenomena for which supervised OE performs remarkably well.

In this paper we present surprising experimental results that challenge the assumption that deep AD on images needs an unsupervised approach (with or without OE). Using the same OE setup as Hendrycks et al. (2019b), which is common in the literature, we find that a standard classifier outperforms current state-of-the-art AD methods on the one vs. rest AD benchmark with CIFAR-10 and ImageNet. The one vs. rest benchmark has been recommended as a standard evaluation protocol to validate AD methods (Emmott et al., 2013) and is used as a litmus test in virtually all deep AD papers published at top-tier venues; see e.g. (Ruff et al., 2018; Deecke et al., 2018; Golan & El-Yaniv, 2018; Akcay et al., 2018; Hendrycks et al., 2019b; Abati et al., 2019; Perera et al., 2019; Wang et al., 2019a; Ruff et al., 2020; Bergman & Hoshen, 2020; Kim et al., 2020; Liznerski et al., 2021; Deecke et al., 2021). Further challenging common assumptions, we find that OE does *not* seem to require huge amounts of data to represent "anomalousness." A classifier requires only 256 random OE samples to outperform the state of the art on ImageNet and only *one* well-chosen OE sample to score reasonably compared to unsupervised methods and classical AD approaches.

OE, however, does not solve all types of AD problems, in particular when the normal dataset is highly diverse or when anomalies are very subtle. For instance, we demonstrate that the methods need more OE samples to achieve top performance on the less studied, more challenging leave-one-class-out AD benchmark Bergman & Hoshen (2020); Ahmed & Courville (2020); Deecke et al. (2021) where many classes are combined to form a multimodal normal class. On the DTD (describable textures) one vs. rest benchmark and on MVTec-AD, a recent manufacturing dataset, we show that random natural images are not very informative as OE.

We also investigate transfer learning approaches to AD that have recently improved AD on images (Reiss et al., 2021; Deecke et al., 2021). Using CLIP (Radford et al., 2021), a recent foundation model, we find that it is possible to set a state of the art on CIFAR-10 and ImageNet *without any additional training data.* While transfer learning and standard classification work well, we still find advantages of unsupervised OE over supervised OE. When there are very few ($< 32$) OE samples or the OE samples are not very informative, unsupervised OE approaches outperform classifiers, indicating a certain robustness with respect to the training outliers. On low-dimensional datasets like Fashion-MNIST and MNIST, where shallow approaches without OE perform well, the intuition described in Figure 1 is still valid: Standard classifiers require a lot of OE data to perform well, whereas one-class methods perform competitively even with just one OE sample.

In conclusion, the primary message of this paper is neither that we propose yet another state-of-the-art method nor that one of our investigated methodologies is of general superiority, but that there is a surprisingly strong performance of off-the-shelf classifiers, transfer learning, and just a few OE samples—contradicting widespread common assumptions for deep AD on well-established AD benchmarks. Through this work, we want to encourage rethinking how previous AD results extend to deep learning.

## 2   Related work

We here briefly review recent developments in deep AD, including self-supervision, transfer learning, and outlier exposure. We further clarify the differences between out-of-distribution detection and AD and discuss non-natural image AD benchmarks.

**Deep anomaly detection**   While there exist many shallow methods for AD, it has been observed that these methods perform poorly on high-dimensional data (Huang & LeCun, 2006; Kriegel et al., 2008; Erfani et al., 2015; 2016). To address this, deep approaches to AD that scale well with higher dimensions have been proposed (Ruff et al., 2021a; Pang et al., 2021). The most common approaches to deep AD employ autoencoders trained on normal data, where samples not reconstructed well at test time are deemed anomalous (Hawkins et al., 2002; Sakurada & Yairi, 2014; Chen et al., 2017; Zhou & Paffenroth, 2017; Nguyen et al., 2019; Kim et al., 2020). Deep generative models detect anomalies via a variety of methods (Schlegl et al., 2017; Deecke et al., 2018; Zenati et al., 2018; Schlegl et al., 2019), yet their effectiveness has been called into question (Nalisnick et al., 2019).

A recent avenue of research uses self-supervision for deep AD on images (Gidaris et al., 2018; Golan & El-Yaniv, 2018; Wang et al., 2019b; Hendrycks et al., 2019b; Tack et al., 2020; Sohn et al., 2021). One of the best-performing AD methods is the self-supervised approach from Tack et al. (2020), which combines Hendrycks et al. (2019b)'s AD method with contrastive representation learning (Chen et al., 2020). Tack

et al. (2020) train their network on transformed normal data so that it maps similar transformations of a sample close together, while sufficiently distorted transformations and other samples are mapped away. The network then has to classify each sample's type of transformation as in (Hendrycks et al., 2019b). For a test sample, both the trained network's certainty on predicting correct transformations and the similarity of the sample with its nearest neighbor in feature space determine its anomaly score: the larger the certainty and similarity, the smaller the anomaly score.

More recently, transfer learning-based approaches to AD (Bergman et al., 2020; Reiss et al., 2021; Deecke et al., 2021) that fine-tune supervised classifiers trained on ImageNet have shown to outperform self-supervised methods such as Tack et al. (2020) on common benchmarks. To the best of our knowledge, Reiss et al. (2021) is the best performing AD method on CIFAR-10 that does not use OE (Hendrycks et al., 2019a). Their method fine-tunes a ResNet pre-trained on ImageNet on a deep one-class loss (Ruff et al., 2018) and applies continual learning to avoid a feature collapse. Since they use ImageNet pre-trained models, they do not validate their method on the ImageNet one vs. rest benchmark.

The state of the art on the CIFAR-10 benchmark, however, takes advantage of OE (Hendrycks et al., 2019a), which follows the idea of using a large unstructured corpus of images as "auxiliary anomalies" during training. For example, Hendrycks et al. (2019b) use OE to improve their self-supervised method by training the network to predict the uniform distribution for all transforms on OE samples, while leaving training on normal samples unchanged. Reiss et al. (2021) and Deecke et al. (2021) combined transfer learning and OE, which yields the currently best performing methods.

**Out-of-distribution detection and anomaly detection**  A field of research related to AD is out-of-distribution (OOD) detection, where the aim is to detect anomalous samples that do not belong to any of the given classes of a multi-class classification task (Lee et al., 2018). One can always apply an AD method to OOD by using it separately from the classifier, treating all training samples of all given classes as normal, ignoring the available class labels. However, AD methods typically (and expectedly) perform worse than specialized OOD methods that take advantage of the in-distribution labels and confidence scores of a trained classifier (Liang et al., 2018; Tack et al., 2020; Hendrycks et al., 2022). Such methods define the anomaly score to be large when the maximum of the softmax outputs (Liang et al., 2018) or logits (Hendrycks et al., 2022) is small, i.e. when the classifier is uncertain about the classification of a sample. Conversely to AD methods, which are applicable to OOD problems, OOD methods cannot be applied to AD setups due to the absence of in-distribution class labels. Note that the type of auxiliary supervision via OE we utilize in this paper hence differs from the kind of supervision applied in OOD, as we do not discriminate between different classes of normality but only between normal samples and auxiliary outliers.

**Non-natural image AD**  Recently there has been increasing attention on image AD on "non-natural" images (e.g. medical images or technical images from manufacturing), where anomalies tend to be more subtle. For example, the MVTec-AD dataset consists of photos from manufacturing with, for instance, screws being normal and defective screws being anomalous (Bergmann et al., 2019). In this paper we instead focus on the common and well-established one vs. rest benchmarks with natural images, aiming to detect images of natural classes that are semantically different from the normal class. For other types of image data, random natural images from the web are likely not informative as OE. We show this in Section 5.2 and Appendix I for the example of MVTec-AD and, less prominently, for DTD. However, one can see that both transfer learning and OE *can* work well in other settings as many state-of-the-art methods on MVTec-AD rely on one of these. For instance, Liznerski et al. (2021); Schlüter et al. (2021); Li et al. (2021) employ OE in the form of synthetically generated anomalies and Defard et al. (2021); Gudovskiy et al. (2022); Roth et al. (2022) use transfer learning-based methods.

## 3  Methods

In this section, we introduce the methods that we will use for our experimental evaluation. We first motivate why AD typically follows an unsupervised approach and is not viewed as a binary classification problem. Afterwards, we introduce deep one-class classification as well as CLIP (Radford et al., 2021) for zero-shot AD.

### 3.1 AD as a classification problem

Traditionally AD is understood as the problem of estimating the support (or level sets of the support) of the normal data-generating distribution. This is known as *density level set estimation* (Polonik, 1995; Tsybakov, 1997). This follows the assumption that normal data is concentrated whereas anomalies are not concentrated (Schölkopf & Smola, 2002). Steinwart et al. (2005) remark that density level set estimation can also be interpreted as a binary classification problem between the normal and an anomalous distribution. Many classic AD methods (e.g., kernel density estimation or one-class SVMs) implicitly assume the anomalies to follow a uniform distribution, i.e. they make an uninformative prior assumption on the anomalous distribution (Steinwart et al., 2005). These methods, as well as a binary classifier trained to discriminate between normal samples and uniform noise, are in fact asymptotically consistent density level set estimators (Steinwart et al., 2005; Vert & Vert, 2006). Practically, however, it is preferable to estimate the level set directly rather than classifying against uniform noise. Such a classification approach is particularly ineffective and inefficient in high dimensions since it would require massive amounts of noise samples to properly fill the sample space. As we show in our experiments, however, we find that this intuition does not seem to extend to deep anomaly detection on images.

### 3.2 Deep one-class classification

Deep one-class classification (Ruff et al., 2018) was introduced as a deep learning extension of the one-class classification approach to anomaly detection (Schölkopf et al., 2001; Tax, 2001). Deep SVDD (Ruff et al., 2018) is trained to map normal samples close to a center $c$ in feature space, thereby following the concentration assumption (Schölkopf & Smola, 2002) mentioned above. For a neural network $\phi_\theta$ with parameters $\theta$, the Deep SVDD objective is given by $\min_\theta \frac{1}{n} \sum_{i=1}^n \|\phi_\theta(\boldsymbol{x}_i) - \boldsymbol{c}\|^2$. Ruff et al. (2020) proposed an extension of Deep SVDD that incorporates known anomalies, called *Deep Semi-supervised Anomaly Detection* (Deep SAD). Deep SAD trains a network to concentrate normal data near the center $c$, while mapping anomalous samples away from that center. Hence, this follows an unsupervised OE approach to AD. Here, we present a principled modification of Deep SAD based on the cross-entropy loss, which we call *hypersphere classification* (HSC). We find that this modification improves performance over Deep SAD and use it in our experiments as a prototypical representative of the unsupervised OE approach to AD. Potentially, one could further improve the unsupervised OE approach by developing an OE variant of CSI (Tack et al., 2020), which we leave to future work and is out of scope of this paper.

Let $\mathcal{D} = \{(\boldsymbol{x}_1, y_1), \ldots, (\boldsymbol{x}_n, y_n)\}$ be a dataset with $\boldsymbol{x}_i \in \mathbb{R}^d$ and $y \in \{0, 1\}$ where $y = 1$ denotes normal and $y = 0$ anomalous instances. Let $\phi_\theta : \mathbb{R}^d \to \mathbb{R}^r$ be a neural network and $l : \mathbb{R}^r \to [0, 1]$ a function that maps the output to a probabilistic score. Then, the cross-entropy loss is given by

$$-\frac{1}{n} \sum_{i=1}^n y_i \log l(\phi_\theta(\boldsymbol{x}_i)) + (1-y_i) \log\left(1 - l(\phi_\theta(\boldsymbol{x}_i))\right). \tag{1}$$

For a standard binary classifier, $l$ is a linear layer followed by a sigmoid, and the decision region of the mapped samples $\phi_\theta(\boldsymbol{x}_1), \ldots, \phi_\theta(\boldsymbol{x}_n)$ is a half-space $S$. In this case, the preimage of $S$, $\phi_\theta^{-1}(S)$, is not guaranteed to be compact. To encourage the preimage of our normal decision region to be compact, we choose $l$ to be a radial basis function: $l(\boldsymbol{z}) = \exp\left(-\|\boldsymbol{z}\|^2\right)$. In this case, (1) becomes

$$\frac{1}{n} \sum_{i=1}^n y_i \|\phi_\theta(\boldsymbol{x}_i)\|^2 - (1-y_i) \log\left(1 - \exp\left(-\|\phi_\theta(\boldsymbol{x}_i)\|^2\right)\right).$$

If there are no anomalies, the HSC loss simplifies to $\frac{1}{n} \sum_{i=1}^n \|\phi_\theta(\boldsymbol{x}_i)\|^2$. For $\boldsymbol{c} = 0$, we thus recover Deep SVDD as a special case. Similar to Deep SVDD/SAD, we define our anomaly score as $s(\boldsymbol{x}) := \|\phi_\theta(\boldsymbol{x})\|^2$. Motivated by robust statistics (Hampel et al., 2005; Huber & Ronchetti, 2009), we also considered replacing $l$ with radial functions that replace the squared-norm with more robust alternatives. Here, we found the pseudo-Huber loss (Charbonnier et al., 1997) to consistently yield the best results. We refer to Appendix E for a detailed analysis.

### 3.3 Contrastive language-image pre-training

To challenge the assumption that transfer learning approaches require OE for state-of-the-art detection performance, we consider a zero-shot approach to AD using the features of the contrastive language-image pre-training (CLIP) model (Radford et al., 2021). CLIP is trained on a massive dataset of 400 million (image, text) pairs with an objective to align corresponding pairs in feature space while keeping other pairs apart. Let $(\boldsymbol{x}_u, \boldsymbol{x}_v)$ denote an (image, text) pair, $\boldsymbol{u} = f_u(\boldsymbol{x}_u)$ and $\boldsymbol{v} = f_v(\boldsymbol{x}_v)$ the corresponding representations obtained by networks $f_u$ and $f_v$, and consequently $\boldsymbol{u}_i$ and $\boldsymbol{v}_i$ the representations of the $i$-th data pair. CLIP uses the following losses: the text-to-image loss $l_i^{(v \to u)}$ and the image-to-text loss $l_i^{(u \to v)}$

$$l_i^{(v \to u)} = -\log \frac{\exp(\langle \boldsymbol{v}_i, \boldsymbol{u}_i \rangle \, e^\tau)}{\sum_{k=1}^N \exp(\langle \boldsymbol{v}_i, \boldsymbol{u}_k \rangle \, e^\tau)}, \quad l_i^{(u \to v)} = -\log \frac{\exp(\langle \boldsymbol{u}_i, \boldsymbol{v}_i \rangle \, e^\tau)}{\sum_{k=1}^N \exp(\langle \boldsymbol{u}_i, \boldsymbol{v}_k \rangle \, e^\tau)}, \tag{2}$$

where $\langle \cdot, \cdot \rangle$ denotes the cosine similarity and $\tau$ is a temperature parameter. CLIP's final objective is

$$\min_{f_u, f_v, \tau} \frac{1}{N} \sum_{i=1}^N \left( l_i^{(v \to u)} + l_i^{(u \to v)} \right) / 2. \tag{3}$$

Radford et al. (2021) report that, without any fine-tuning on the downstream task, CLIP is able to outperform a fully supervised linear classifier with ResNet-50 features on several classification benchmarks, including ImageNet. For this, they use the names of the dataset classes as potential text candidates and predict the class whose text has the largest alignment with a given image. For out-of-distribution detection, Fort et al. (2021) explored using CLIP by taking the in-distribution and out-of-distribution text labels as candidates. We use CLIP in a similar way to perform zero-shot AD, where we use the text pair $(v_1, v_2) = ($"a photo of a {NORMAL_CLASS}", "a photo of something"$)$. For a test image $\boldsymbol{x}$, we compute its anomaly score as

$$s(\boldsymbol{x}) = \frac{\exp(\langle f_u(\boldsymbol{x}), f_v(v_2) \rangle \cdot 100)}{\sum_{k=1}^2 \exp(\langle f_u(\boldsymbol{x}), f_v(v_k) \rangle \cdot 100)}. \tag{4}$$

Fine-tuning CLIP for AD with OE also is straightforward. We simply minimize the score (4) for normal samples and maximize it for OE samples. Since this corresponds to a binary cross-entropy loss, this is an instance of supervised OE, which we term "BCE with CLIP" (or just BCE-CL).

**On the legitimacy of transfer learning for AD** The use of transfer learning improved the performance of deep AD approaches significantly. Yet, it seems at least questionable whether the use of pre-trained models is experimentally sound since there may be a semantic overlap between the pre-training data and the anomalies seen at test time. While it is technically true that the AD model is still unsupervised and does not exploit knowledge of test samples, the reported performance on typical image AD benchmarks might be spurious as the model may not generalize well to other data. Radford et al. (2021), however, have investigated the overlap of data used for pre-training CLIP, where they remove all the data that overlaps with the downstream tasks and observe only an insignificant drop in performance on average. This suggests that our experiments and results based on CLIP reasonably explore generalization performance. We still want to raise awareness for this somewhat problematic trend in deep AD, however, for which we propose future research below.

## 4 Experimental setups

Before we present our results, we explain the experimental setup. In particular, we introduce the common one vs. rest benchmark, the CIFAR-10, ImageNet, CUB, DTD, Fashion-MNIST, and MNIST datasets, and the state-of-the-art AD methods we consider in our experiments.

**One vs. rest benchmark** The one vs. rest evaluation protocol is a ubiquitous benchmark in the deep AD literature (Ruff et al., 2018; Golan & El-Yaniv, 2018; Hendrycks et al., 2019a;b; Ruff et al., 2020; Sohn et al., 2021; Deecke et al., 2021; Liznerski et al., 2021; Reiss et al., 2021). This benchmark constructs AD settings

from classification datasets (e.g., CIFAR-10) by considering the "one" class (e.g., "airplane") as being normal and the "rest" classes (e.g., "automobile", "bird", ...) as being anomalous at test time. In each experiment, we train a model using only the training set of the normal class and samples from an OE set that are not contained in the anomaly classes of the benchmark. We use the same OE auxiliary datasets as suggested in previous works (Hendrycks et al., 2019a;b; Liznerski et al., 2021). To evaluate detection performance, we consider the commonly used Area Under the ROC curve (AUC) on the one vs. rest test sets. This is repeated over classes and multiple random seeds.

**Datasets**   For our experiments we focus on the well-established one vs. rest versions of CIFAR-10 and ImageNet. We also consider less common datasets, for which there are yet no AD results in the literature. If not mentioned otherwise, we use all available classes as our one vs. rest classes.

- CIFAR-10: For CIFAR-10 (Krizhevsky et al., 2009), we use 80 Million Tiny Images (80MTI)(Torralba et al., 2008) as OE, with CIFAR-10 and CIFAR-100 images removed. This follows the experimental setup in Hendrycks et al. (2019b).
- ImageNet-30: For ImageNet (Deng et al., 2009), we use a subset of 30 classes as the one vs. rest classes, which are the same classes used in Hendrycks et al. (2019b). For OE, we use ImageNet-22K with ImageNet-1K removed, again following Hendrycks et al. (2019b).
- CUB: CUB (Caltech-UCSD Birds-200-2011) (Wah et al., 2011) is a more challenging dataset where each of the 200 classes consists of images of a specific bird type (e.g., "black footed albatross", "blue jay", ...). We again use ImageNet-22k as OE.
- DTD: DTD (Describable Textures Dataset) (Cimpoi et al., 2014) is a dataset containing 47 different classes of images of textures like "cracked" and "braided". We use ImageNet-22k as OE.
- Fashion-MNIST: For Fashion-MNIST (Xiao et al., 2017), we consider a grayscale version of CIFAR-100 as OE, as suggested in Liznerski et al. (2021).
- MNIST: For MNIST (Deng, 2012), we use EMNIST (Cohen et al., 2017) as OE.

**End-to-end methods**   We present results from end-to-end methods (without transfer learning) including all methods that achieve state-of-the-art performance on the CIFAR-10 and ImageNet-30 one vs. rest benchmarks.

- Unsupervised: Shorthands for unsupervised methods are DSVDD (Ruff et al., 2018), GT (Golan & El-Yaniv, 2018), GT+ (Hendrycks et al., 2019b), and CSI (Tack et al., 2020).
- Unsupervised OE: We implement HSC from Section 3.2 and DSAD (Ruff et al., 2020) as unsupervised OE methods and also report the results from the unsupervised OE variant of GT+ (Hendrycks et al., 2019b).
- Supervised OE: BCE denotes a standard binary cross-entropy classifier. We also implement the Focal loss classifier with $\gamma = 2$ (Lin et al., 2017), a BCE variant for imbalanced classes that was also presented in Hendrycks et al. (2019b). Results from Hendrycks et al. (2019b) are marked with an asterisk as Focal*.

**Transfer learning-based methods**   We consider the following transfer learning-based methods.

- Unsupervised: We implement a zero-shot anomaly detector using CLIP's feature space as described in Section 3.3 and use DN2 (Bergman et al., 2020) and PANDA (Reiss et al., 2021) as shorthands respectively for these unsupervised methods from the literature.
- Supervised OE: We consider a fine-tuned version of CLIP with a binary cross-entropy classifier, denoted as BCE-CL. When available, we also report the results of ADIP (Deecke et al., 2021) and the supervised OE variant of PANDA (Reiss et al., 2021).

We provide all network architecture and optimization details in Appendix G and investigate the impact of $\gamma$ for the Focal loss in Appendix F. We report the mean AUC performance over all classes and seeds in the main paper. Individual results per class and method are given in Appendix I. Each experiment is averaged over 10 seeds if not mentioned otherwise.

## 5   On the usefulness of samples in deep AD

Traditionally, AD methods utilize an unsupervised approach due to the assumption that it is impossible to characterize everything that is not normal. As mentioned above, later works introduced the idea of including a large collection of random images (OE) during training that serve as auxiliary examples of anomalousness to

improve unsupervised AD methods (Hendrycks et al., 2019a). Models are trained with these OE samples using a loss that is essentially inverting a given unsupervised AD loss. We here look into whether an unsupervised OE approach, instead of a straightforward supervised OE approach using standard binary cross-entropy, is really necessary.

### 5.1 Supervised OE achieves state-of-the-art results

The first work on OE (Hendrycks et al., 2019a) applied an unsupervised OE method to two experimental setups: CIFAR-10 with 80MTI as OE and ImageNet-30 with ImageNet-22K as OE. Here we consider the same experimental setups with the basic HSC and BCE classifiers. We also include experiments on the mostly unexplored one vs. rest versions of CUB, DTD, Fashion-MNIST, and MNIST. The results are shown in Table 1. In this section we only consider end-to-end methods (i.e., no transfer learning).

Table 1: Mean AUC detection performance in % for end-to-end methods on the CIFAR-10, ImageNet-30, CUB, DTD, Fashion-MNIST, and MNIST one vs. rest benchmarks.

|  | Unsupervised | | | Unsupervised OE | | | Supervised OE | | |
|  | DSVDD | GT+* | CSI* | GT+* | DSAD | HSC | Focal* | Focal | BCE |
|---|---|---|---|---|---|---|---|---|---|
| CIFAR-10 | 64.8* | 90.1 | 94.3 | 95.6 | 94.5 | 95.9 | 87.3 | 95.8 | **96.1** |
| ImageNet-30 | 61.1 | 84.8 | 91.6 | 85.7 | 96.7 | 97.3 | 56.1 | 97.5 | **97.7** |
| CUB | 59.0 | × | × | × | 81.1 | 83.2 | × | 83.5 | **84.1** |
| DTD | 56.8 | × | × | × | 72.4 | 72.7 | × | 73.2 | **73.3** |
| Fashion-MNIST | 86.3 | × | × | × | 86.4 | 87.3 | × | **87.7** | 86.4 |
| MNIST | 97.6 | × | × | × | 97.6 | 98.2 | × | **98.4** | **98.4** |

**Discussion** Surprisingly, we find that the choice of AD method has little impact on performance. On CIFAR-10, all unsupervised OE methods yield a comparable detection performance, while the supervised methods (Focal and BCE) show state-of-the-art performance, with BCE attaining the overall best mean AUC. On ImageNet, Deep SAD, HSC, Focal, and BCE all outperform the current state of the art (CSI*) by a significant margin. We are unsure as to why the Focal* performs so poorly in Hendrycks et al. (2019b) since their experimental code is not publicly available. On CUB and DTD, we observe a similar behavior, but the methods perform slightly worse, most likely due to the more fine-grained class categorization in CUB and the more subtle anomalies in DTD. These benchmarks, as they show room for improvement, thus present themselves useful for future AD evaluation. On the rather low-dimensional Fashion-MNIST and MNIST datasets, OE is not able to improve significantly over unsupervised approaches, and the different OE approaches again perform similarly. Overall, our results show that a vanilla classifier using OE outperforms all previous deep AD approaches.

Our experiments seem to suggest that the inclusion of OE does not just improve AD performance, it also changes the problem into a typical supervised classification problem that does not require a compact decision boundary (cf. Figure 1). This stands in contrast to previous observations in shallow AD (Tax, 2001; Görnitz et al., 2013). While there is an abundance of OE data for the sorts of AD problems we investigate here, we want to highlight that this data is likely not very helpful for AD problems where anomalies are more subtle. For example, in the realm of manufacturing, pictures of cats and trucks are not as useful for OE as industrial images, which are not so widely available. Nonetheless, one may still have a few anomalous examples to incorporate during training. As one removes OE data, we would expect the behavior of an unsupervised OE approach to transition to the behavior of an unsupervised method that outperforms supervised OE. Our next experiment investigates this transition.

### 5.2 Few OE samples are sufficient

To investigate the effect of the size of the OE corpus, we perform experiments varying the OE training set size from just *one* sample ($2^0 = 1$) to using the maximal amount of OE data. Our results for the

high-dimensional datasets CIFAR-10, ImageNet-30, and CUB are shown in Figure 2. With sufficiently few samples, HSC outperforms BCE, which seems to indicate a regime where the unsupervised OE approach is advantageous. Interestingly, this regime seems to be quite small, with supervised OE needing only 8 samples to outperform unsupervised OE on CIFAR-10 and only $\sim 32$ samples for such a transition on ImageNet and CUB. Remarkably, BCE classification outperforms previous state-of-the-art methods on ImageNet with only using $\sim 256$ OE samples. A training set with so few outliers and high-dimensional data represents an instance of skewed data (see Figure 1 and caption), where the common intuition suggests that supervised OE should *not* generalize well. As before, we see that the choice of the specific method seems negligible as soon as sufficient OE data is available.

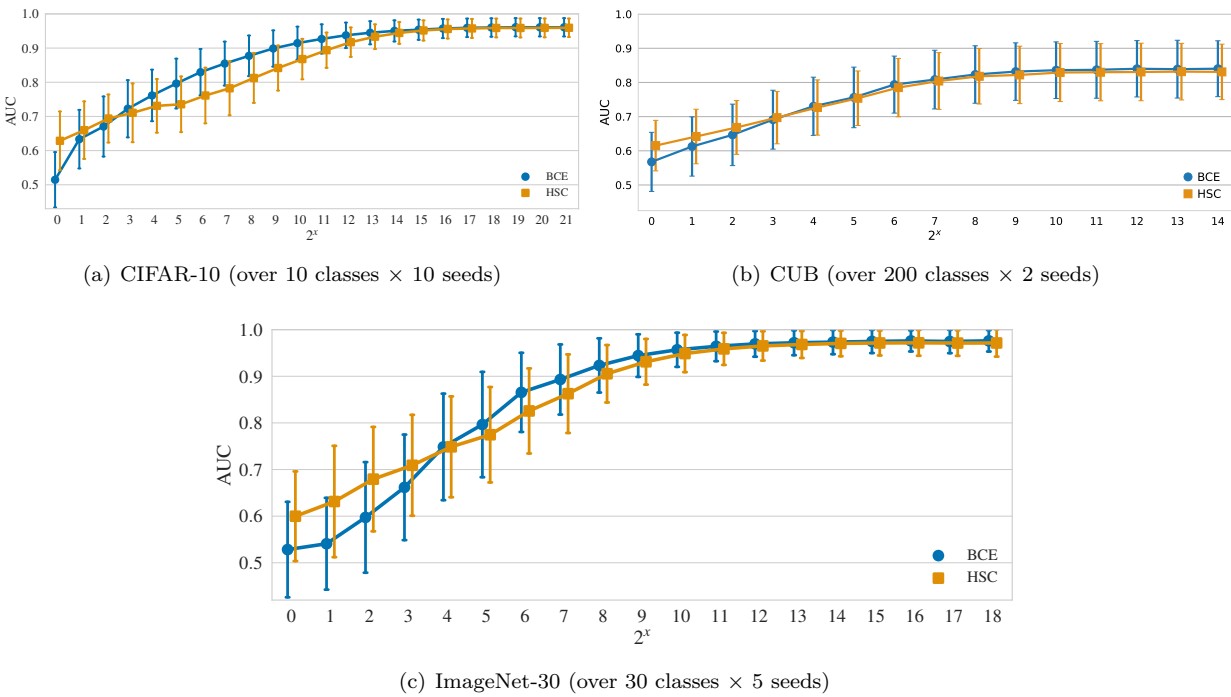

(a) CIFAR-10 (over 10 classes $\times$ 10 seeds)    (b) CUB (over 200 classes $\times$ 2 seeds)

(c) ImageNet-30 (over 30 classes $\times$ 5 seeds)

Figure 2: Mean AUC detection performance in % on the CIFAR-10, CUB, and ImageNet-30 one vs. rest benchmarks when varying the number of 80MTI and ImageNet-22K OE samples, respectively.

Results with varied OE set size for Fashion-MNIST and MNIST are presented in Figure 3. We find that, in contrast to the experiments on high-dimensional datasets, the amount of OE has little impact on performance for the well-performing HSC, whereas BCE requires a lot of OE to perform competitively. It seems that the intuition of Figure 1 does still hold for low-dimensional datasets where shallow methods perform well.

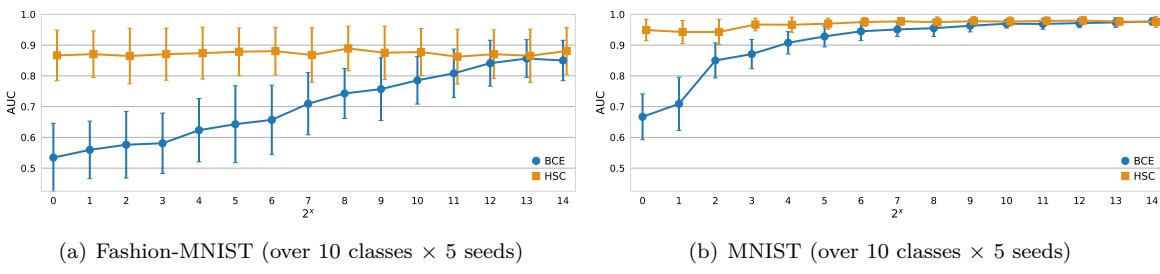

(a) Fashion-MNIST (over 10 classes $\times$ 5 seeds)    (b) MNIST (over 10 classes $\times$ 5 seeds)

Figure 3: Mean AUC detection performance in % on the Fashion-MNIST and MNIST one vs. rest benchmarks when varying the number of grayscale CIFAR-100 and EMNIST OE samples, respectively.

**There are settings that require more OE data**   The one vs. rest image-AD benchmark represents a certain type of typical AD problem where the normal data distribution is roughly unimodal since the normal samples are drawn from just one class. However, one can also create a more challenging, and perhaps less typical, AD benchmark by considering the "rest" classes (e.g., "automobile", "bird", ...) as being normal and the "one" class (e.g., "airplane") as being anomalous at test time. The distribution of normal samples in this leave-one-class-out approach becomes multimodal. Figure 4 shows the AD performance of HSC and BCE on the CIFAR-10 and ImageNet-30 leave-one-class-out image-AD benchmark, when one varies the size of the OE training set.

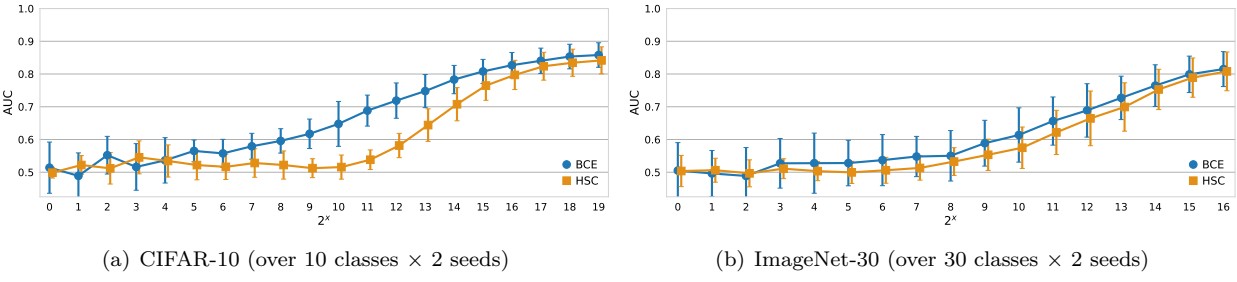

(a) CIFAR-10 (over 10 classes × 2 seeds)     (b) ImageNet-30 (over 30 classes × 2 seeds)

Figure 4: Mean AUC detection performance in % on the CIFAR-10 and ImageNet-30 leave-one-class-out benchmarks when varying the number of 80MTI and ImageNet-22K OE samples, respectively.

Compared to previous experiments it seems that more OE data is required to achieve strong performance. This indicates that more OE samples are necessary when the normal class is not concentrated. We also report results when one uses the full OE dataset in Appendix I.3, where we see that the various methods perform similarly with respect to one another as in the one vs. rest tasks, albeit overall slightly worse due to the more difficult problem setting. Notably BCE is the best performing method, further supporting its effectiveness at AD with OE.

**There are settings where random images are not informative as OE**   As mentioned in Section 2, this paper focuses on natural images because random natural images are likely not informative when used as OE in other settings. To demonstrate this, we apply our methods to the one vs. rest DTD benchmark and the MVTec-AD dataset. DTD contains different classes of textures. MVTec-AD contains image-AD scenarios for detecting manufacturing defects for a variety of object types (e.g., screws, bottles, wires, or sections of carpet). For example, one scenario consists of a training set containing images of normal screws and a test set containing images of normal screws along with images of screws with defects, which serve as anomalies. Figure 5 shows results when varying the number of ImageNet-22k samples used as OE for training. We find that the OE training set size has little impact on the AD performance, especially for MVTec-AD.

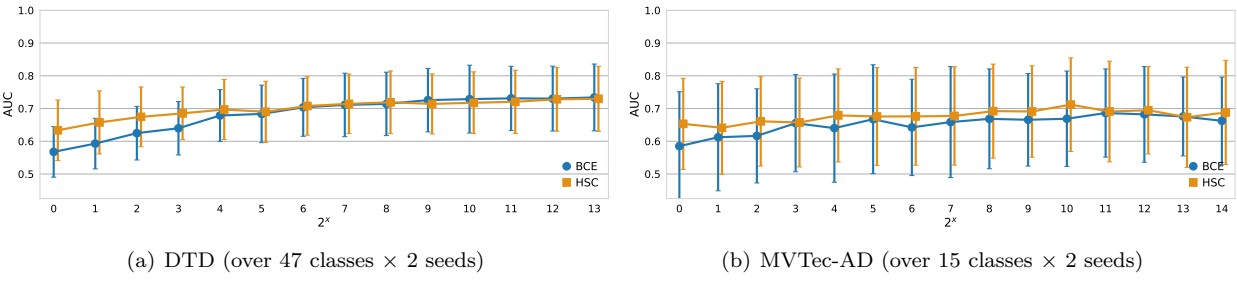

(a) DTD (over 47 classes × 2 seeds)     (b) MVTec-AD (over 15 classes × 2 seeds)

Figure 5: Mean AUC detection performance in % on the DTD one vs. rest and on the MVTec-AD benchmarks when varying the number of ImageNet-22K OE samples.

We also show results using the full ImageNet-22k dataset as OE for various methods for MVTec-AD in Table 18 in Appendix I.4. As expected, OE performs more poorly on MVTec-AD than state-of-the-art methods: HSC scores around 70% while Roth et al. (2022) score around 99% AUC. Contrary to the experiments with CIFAR-10, CUB, and ImageNet-30, we observe that HSC is a bit stronger than BCE (66%). There seems to be some evidence that HSC performs better when the OE data is not very informative, either when there is a dearth of OE data like in Section 5.2, or when the OE data simply isn't relevant to the AD scenario. We explore this a bit further in Section 5.4.

**Diversity of OE data is important**  To measure the impact of data diversity we also experiment with varying the number of OE classes. We defer these results to Appendix A. In summary, we find that performance overall increases with OE data diversity, but already using just one OE class still performs relatively well.

While our results show that surprisingly few samples are needed to achieve competitive performance for end-to-end models on the standard one vs. rest AD benchmarks, more recent methods in deep AD tend to use transfer learning. Interestingly, we find that one can achieve state-of-the-art AD performance using pre-trained models with *no training samples* (i.e., in a zero-shot setting).

### 5.3  Transfer learning enables zero-shot AD with state-of-the-art performance

More recent progress in deep AD has been achieved through transfer learning, which has further improved the state of the art on standard deep AD benchmarks. These methods use a network pre-trained on large datasets to provide rich representations as a starting basis for deep AD. Again, we look into how useful auxiliary data is for these algorithms. Here we investigate the effect of using OE data (vs. no OE data at all) and the implicit use of an extraneous dataset via pre-training. We also consider the situation where *no normal data* is used and find that CLIP outperforms all previous end-to-end methods on virtually all datasets. Table 2 shows the results for transfer learning-based methods on the one vs. rest benchmarks. Note that, since DN2, PANDA, and ADIP employ ImageNet pre-trained networks, they cannot be compared on the ImageNet AD benchmark in a fair way.

**Discussion**  These results highlight the remarkable efficacy of pre-training. Disregarding the CLIP results for now, we see that pre-training improves over previous unsupervised deep results on CIFAR-10. The results of PANDA indicates that OE is still useful for deep AD: OE provides additional information that is not learned in the pre-training task. Undoubtedly, the most interesting result here is the observation that one can outperform all previous state-of-the-art methods using no additional training from the benchmarks.

Table 2: Mean AUC detection performance in % for methods with transfer learning on the CIFAR-10, ImageNet-30, CUB, DTD, Fashion-MNIST, and MNIST one vs. rest benchmark.

|  | Unsupervised | | | Supervised OE | | |
|  | DN2* | PANDA* | CLIP | PANDA* | ADIP* | BCE-CL |
|---|---|---|---|---|---|---|
| CIFAR-10 | 92.5 | 96.2 | 98.5 | 98.9 | 99.1 | **99.6** |
| ImageNet-30 | × | × | 99.88 | × | × | **99.90** |
| CUB | × | × | 97.1 | × | × | **97.5** |
| DTD | × | × | 90.2 | × | × | **94.6** |
| Fashion-MNIST | × | × | 89.0 | × | × | **94.7** |
| MNIST | × | × | 59.0 | × | × | **96.0** |

In a zero-shot setting, CLIP (unsupervised) outperforms all previous end-to-end (see Table 1) and unsupervised methods on all datasets apart from MNIST. Fine-tuning CLIP with OE along with normal samples (BCE-CL), further improves CLIP's results setting a new state of the art on the CIFAR-10 and ImageNet benchmarks. Remarkably, CLIP performs similarly on DTD that consists of textures instead of natural images and, on CUB, outperforms end-to-end methods by a large margin (13.4% AUC), essentially solving this quite challenging benchmark.

Similar to results in other areas of deep learning (Bommasani et al., 2021), the use of large pre-trained networks offers an effective and convenient way to improve performance. Though transfer learning seems to be a natural endpoint for a certain class of deep AD problems, it still leaves a more general question about the difference between supervised and unsupervised approaches to deep AD in settings where transfer learning is not appropriate. This may happen when there are very subtle semantic novelties (Vaze et al., 2022), when one simply cannot use a pre-trained network; for example, when one must train a network from scratch due

to architectural considerations (e.g., when one requires a smaller architecture due to hardware constraints), or when there are security concerns regarding the white-box nature of pre-trained networks (Samek et al., 2021). Additionally, some AD techniques offer no obvious way to utilize a pre-trained network, like the recently introduced explainable one-class variant (Liznerski et al., 2021) or methods based on probabilistic models.

## 5.4 On the robustness of HSC vs. BCE

Our previous experiments (Section 5.2) have shown that, though end-to-end BCE overall outperforms HSC, an unsupervised OE approach is more effective when only very few ($< 32$) OE samples are available or when the OE data is not very informative. This indicates a certain degree of robustness to the anomalous training samples for HSC. This robustness with regards to OE is likely inherited from the learning objectives of unsupervised and semi-supervised AD, which encourage the normal representations to be concentrated thereby avoiding the issue with skewed OE data (see Figure 1). So, even on high-dimensional datasets, there seems to be a regime, albeit a very small one, where the intuition with skewed data in Figure 1 *does* hold.

To demonstrate the robustness of HSC when one has few OE samples we investigate the extreme case where an OE dataset consists of only *one* sample in more detail. How much can a single sample help or hinder the different approaches to deep AD? This is analogous to the experiments in Section 5.2 with the number of OE samples fixed to one. To investigate robustness, we search for the OE sample that gives the worst test AUC for each class and report the average AUC over all classes. While it is desirable for a method to be unaffected by detrimental OE examples, one would still want an AD method to exploit beneficial OE examples. To measure this trade-off, we perform an analogous experiment where we search for the best performing OE sample and again report the average test AUC over all classes.

Table 3: Mean AUC detection performance in % for the best and worst single OE samples on the CIFAR-10 AD benchmark with 80MTI as OE and on the ImageNet-10 AD benchmark with ImageNet-22K (without the 1K classes) as OE.

|  | CIFAR-10 | | ImageNet-10 | |
| --- | --- | --- | --- | --- |
|  | HSC | BCE | HSC | BCE |
| Best OE | 77.7 | 69.9 | 79.3 | 75.5 |
| Worst OE | 43.3 | 31.6 | 39.2 | 26.3 |

As it is computationally prohibitive to test every possible OE sample, we utilize an evolutionary algorithm to attempt to minimize or maximize the class AUC. A detailed description of the algorithm can be found in Appendix H. To ensure that this optimization scheme does not find poor local minima, we also evaluate the AUC of many randomly chosen OE samples (see Appendix D). We find that the evolutionary algorithm almost always finds better optima. We report our results for CIFAR-10 and for the first 10 classes of ImageNet-30 in Table 3. Figure 6 shows optimal OE examples on CIFAR-10 for the normal classes "ship" and "cat," and on ImageNet-10 for "airplane" and "dragonfly."

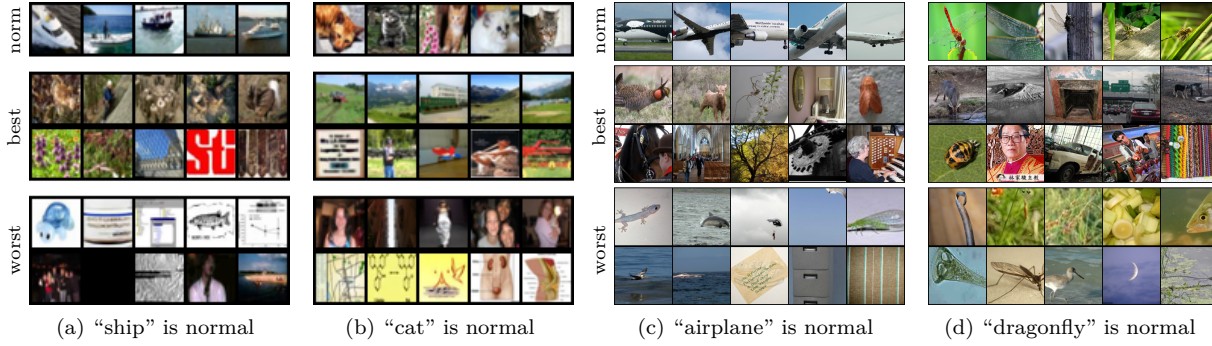

(a) "ship" is normal      (b) "cat" is normal      (c) "airplane" is normal      (d) "dragonfly" is normal

Figure 6: OE samples for CIFAR-10 with 80MTI as OE (a-b) and for ImageNet-10 with ImageNet-22k as OE (c-d). The first row shows normal samples, the next two rows show the best samples for HSC (top) and BCE (bottom), and the last two rows show the worst samples for HSC (top) and BCE (bottom).

**Discussion** On both datasets, we observe that HSC performs better than BCE when using both the best and the worst OE samples. Looking at the samples chosen for the optima, there also appears to be more consistency within a setting (class, dataset, best or worst) for HSC. For example on "best" with "ship," HSC's images are all brownish outdoor photos, whereas the BCE samples vary from greenery to stylized text to an image containing mostly sky. This is likely due to the fact that HSC has, in some sense, an initial notion of anomalousness due to its unsupervised term. For instance, the most useful OE samples are those not already contained in this notion of anomalousness, resulting in HSC having a stable region for selecting OE samples that yield the greatest improvement. BCE lacks this notion so it can benefit from a large variety of OE samples. With HSC, choosing one optimal sample achieves roughly the same performance as using 32 random ones. Interestingly, it seems that near-distribution outliers are less useful as OE samples since samples with similar color patterns as the normal ones occur more often as the "worst" samples. Previous works have found that near-distribution OE samples are useful for OE (Lee et al., 2018; Goyal et al., 2020), however, our results suggest that this may not hold when little OE data is available.

**HSC focuses on low frequency features** To gain further insight into the difference between BCE and HSC, we extend the previous experiments to include frequency-domain corruptions. This sort of analysis has been insightful in other works on deep learning (Yin et al., 2019). We find that HSC is again more robust than BCE since it's generally less affected by the frequency corruptions and tends to focus on low frequency signals in the input. We defer the results and discussion to Appendix B and also show some more examples for the best and worst single OE samples in Appendix I.

## 6 Broader impact

Anomaly detection methods on images may be deployed on tasks which have societal implications such as screening images or automated surveillance, and it is thus imperative that these tasks are done in a fair and transparent way. The use of OE is potentially harmful as there may be OE images biasing the model towards detecting certain entities as anomalous. Our paper aids in this since it demonstrates that no huge corpora are necessary, which enables a controlled selection of OE samples. Further, we have shown that HSC is more robust, and the fact that it chooses optimal OE samples that coincide with human intuition suggests that it is more interpretable than BCE, where the rationale for optimal OE samples is opaque. This makes HSC more suitable for critical applications. We trained some of our models with the 80MTI dataset, which is known to contain problematic data such as offensive labels, but were required to do so to be comparable with the previous line of research.

## 7 Conclusion

We presented surprising results that challenge common assumptions in AD. Neither does deep AD on natural images seem to require specialized AD methodologies nor huge amounts of Outlier Exposure (OE). A standard classifier outperforms all end-to-end methods on the common one vs. rest benchmark, for which it only needs 256 random OE samples on ImageNet and only one well-chosen OE sample for competitive performance. We showed some limitations of the few-OE approaches when applied to settings where the normal dataset is multimodal or the anomalies are more subtle. Using transfer learning, standard classifiers set a new state of the art on CIFAR-10 and, in a zero-shot setting *without using any additional training data,* on ImageNet. Despite the overall strong performance of standard classifiers, we find that semi-supervised one-class methods are more robust to the choice of OE when only few OE examples are available. Our results provide insights about deep AD that are useful for future research.

## 8 Acknowledgements

PL, BJF, and MK acknowledge support by the Carl-Zeiss Foundation, the German Research Foundation (DFG) awards KL 2698/2-1 and KL 2698/5-1, and the German Federal Ministry of Education and Research (BMBF) awards 01|S18051A, 03|B0770E, and 01|S21010C. RV acknowledges support by the Federal Ministry of Education and Research (BMBF) for the Berlin Institute for the Foundations of Learning and Data

(BIFOLD) (01IS18037A). This work was supported in part by the German Ministry for Education and Research under Grant Nos. 01IS14013A-E, 01GQ1115, 01GQ0850, 01IS18025A, 031L0207D, and 01IS18037A. KRM was partly supported by the Institute of Information & Communications Technology Planning & Evaluation (IITP) grants funded by the Korea government(MSIT) (No. 2019-0-00079, Artificial Intelligence Graduate School Program, Korea University and No. 2022-0-00984, Development of Artificial Intelligence Technology for Personalized Plug-and-Play Explanation and Verification of Explanation).

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

## A    Diversity of the Outlier Exposure data

Here we evaluate how data diversity influences detection performance for unsupervised and supervised OE, again comparing HSC to BCE. For this purpose, instead of 80MTI, we now use CIFAR-100 as OE varying the number of anomaly classes available for the CIFAR-10 benchmark. The OE data is varied by choosing $k$ classes at random for each random seed and using the union of these classes as the OE dataset.

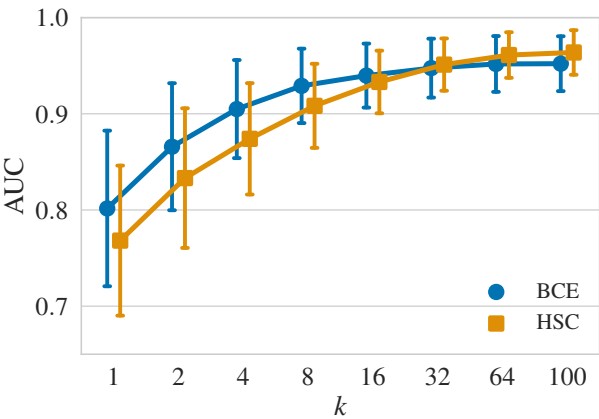

Figure 7: Mean AUC detection performance in % (over 10 classes with 10 seeds per class) on the CIFAR-10 with CIFAR-100 OE one vs. rest benchmark when varying the number of $k$ classes that comprise the OE dataset.

The results are presented in Figure 7. As expected, the performance increases with the diversity of the OE dataset. Interestingly, drawing OE samples from just $k = 1$ class, i.e. binary classification between the normal class and a single OE class (which is not present as an anomaly class at test time!) already yields good detection performance on the CIFAR-10 benchmark. For example, training a standard classification network to discern between automobiles and beavers performs competitively as an automobile anomaly detector, even when no beavers are present as anomalies during test time.

Compared to Section 5.2 in the main paper, where we see a transition to BCE outperforming HSC at $2^3 = 8$ OE samples, we here see that with even one OE class we have passed this transition: there are already enough samples in a single class that BCE outperforms HSC. The takeaway is that OE sample diversity is not as important as one may expect, simply having many OE smaples suffices to enter the regime where BCE outperforms HSC. Again, with many samples, BCE and HSC's performances are comparable.

## B    HSC focuses on low frequency features

To gain further insight into the difference between BCE and HSC and why the best (and worst) found OE samples are quite different for these methods, we repeat the experiment from Section 5.4 in the main paper with frequency-domain corruptions. That is, we low-pass-filter (LPF) or high-pass-filter (HPF) the *entire* dataset (training, testing, and OE) and then proceed exactly as we did in Section 5.4. An LPF removes all higher frequencies and preserves only global information such as a scenery's color. This roughly corresponds to blurring images. An HPF removes all lower frequencies and roughly corresponds to edge detection. The AUC scores for the CIFAR-10 and ImageNet-10 one vs. rest benchmarks are in Table 4. Figure 8 contains examples of the best and worst OE samples and also shows filtered examples. We provide details on the filter implementation in Appendix C, where we evaluate the general performance of BCE and HSC for varying filter magnitudes.

HSC seems to be more robust than BCE since it's generally less affected by the frequency corruptions. Focusing on CIFAR-10, we see that for HSC frequency corruption makes little change to performance except for the Best OE HPF experiment, where HSC's performance is significantly lower than in the LPF variant,

causing HSC to behave similarly to BCE. This may have some implication that when useful signal is contained in the OE sample, corresponding to the "Best OE" experiments, it is concentrated in the low frequency spectrum. It seems as though BCE is not capable of exploiting this data. Interestingly it was also found in Yin et al. (2019) that low frequency features tend do be more robust ("low frequency bias results in improved robustness to corruptions"). On ImageNet we observe that HSC again performs better on the LPF experiments.

Table 4: Mean AUC detection performance in % for the best and worst single OE samples on CIFAR-10 using 80MTI as OE and on ImageNet-10 using ImageNet-22K (with the 1K classes removed) as OE. All images have been either high-pass-filtered (HPF) or low-pass-filtered (LPF) both during training and testing. Arrows indicate the change compared to Table 3.

|  | CIFAR-10 | | ImageNet | |
|  | HSC | BCE | HSC | BCE |
| --- | --- | --- | --- | --- |
| Best OE LPF | 77.5→ | 68.5→ | 77.2↘ | 73.8→ |
| Worst OE LPF | 44.1→ | 31.1→ | 44.6 ↑ | 26.1→ |
| Best OE HPF | 68.8 ↓ | 66.4↘ | 75.0↘ | 77.3→ |
| Worst OE HPF | 43.6→ | 38.0 ↑ | 44.1 ↑ | 27.9→ |

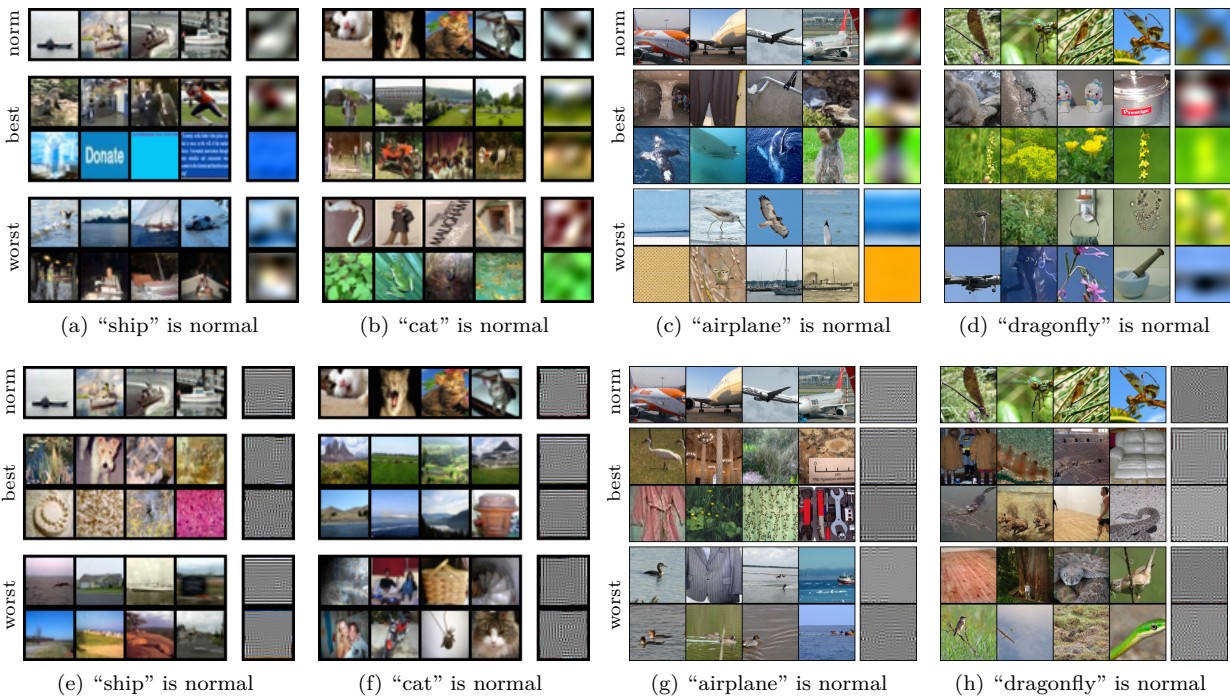

(a) "ship" is normal    (b) "cat" is normal    (c) "airplane" is normal    (d) "dragonfly" is normal

(e) "ship" is normal    (f) "cat" is normal    (g) "airplane" is normal    (h) "dragonfly" is normal

Figure 8: OE samples for low-pass-filtered (a-d) and high-pass-filtered (e-h) versions of CIFAR-10 with 80MTI as OE (a,b,e,f) and ImageNet-10 with ImageNet-22k as OE (c,d,g,h). In each figure, the first row shows normal samples, the next two rows the best samples for HSC (top) and BCE (bottom), and the last two rows the worst samples for HSC (top) and BCE (bottom). The last column shows the filtered version of the images, which is what the network sees during training and testing.

## C  Frequency sensitivity analysis

To understand why so few OE samples are that effective, we investigate the general detection performance of HSC and BCE for images with limited frequency spectra. Analog to Appendix B, we either low-pass (LPF) or

high-pass-filter (HPF) all images, both during training and testing, both normal and anomalous samples. We train and evaluate either HSC or BCE for varying OE dataset sizes and different magnitudes of filters. Note that, due to computational constraints, we decrease the number of epochs and restrict the augmentations for the frequency experiments. Figures 9, 10, 11, and 12 show the results on LPF CIFAR-10, HPF CIFAR-10, LPF ImageNet-30, and HPF ImageNet-30, respectively. Each point in the plots corresponds to the mean AUC detection performance over all classes and 2 seeds per class. Different colors/markers correspond to different amounts of random OE samples used. The magnitudes shown on the horizontal axes correspond to the number of rows and columns removed in the frequency domain. For example, an LPF with a magnitude of $m$ sets the first and last $m$ rows and the first and last $m$ columns of the Fourier-transformed image to zero, before applying the inverse Fourier transformation. A magnitude of 0 corresponds to unfiltered images, a magnitude of 15 on CIFAR-10 images (which have a resolution of $32 \times 32$) corresponds to filtered images where just 4 "pixels" in the center remain in the frequency domain. Similarly, an HPF with a magnitude of $m$ sets the center of size $m \times m$ to zero. The extended OE robustness experiments in Appendix B and Appendix D use a magnitude of 14 for CIFAR-10 and a magnitude of 110 for ImageNet.

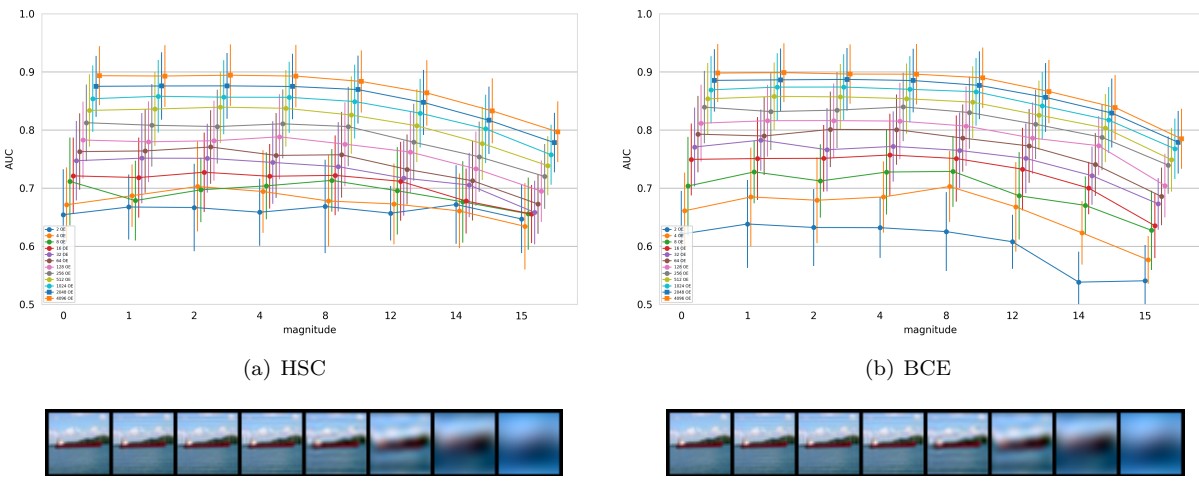

Figure 9: LPF CIFAR-10 with 80MTI OE AD benchmark.

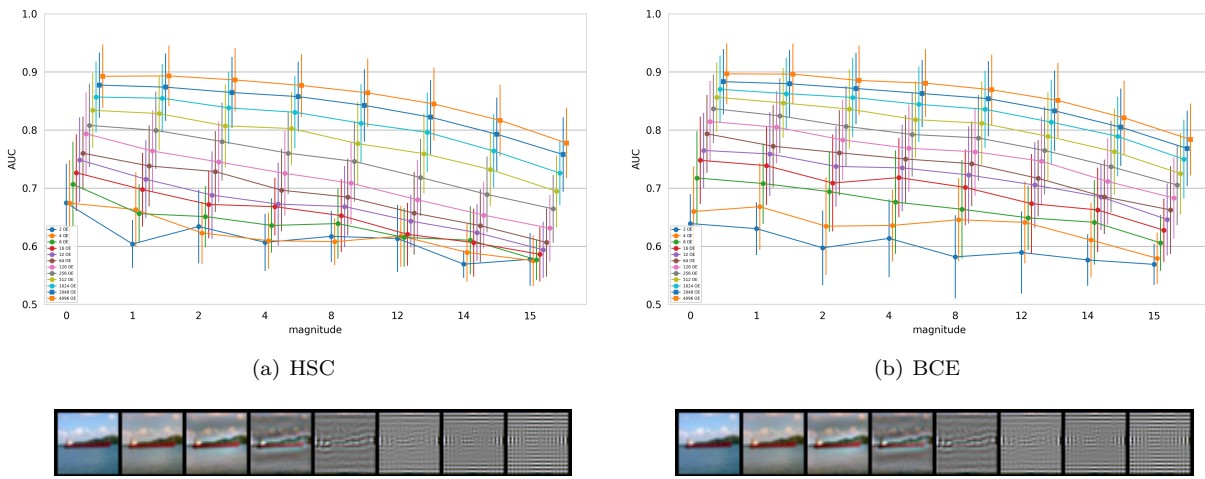

Figure 10: HPF CIFAR-10 with 80MTI OE AD benchmark.

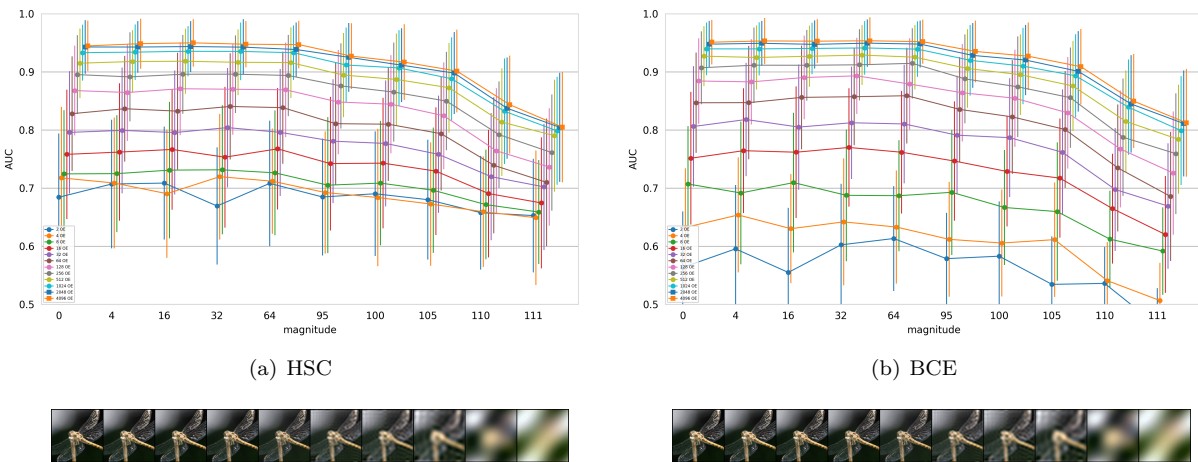

Figure 11: LPF ImageNet-30 with ImageNet-22k (with ImageNet-30 removed) OE AD benchmark.

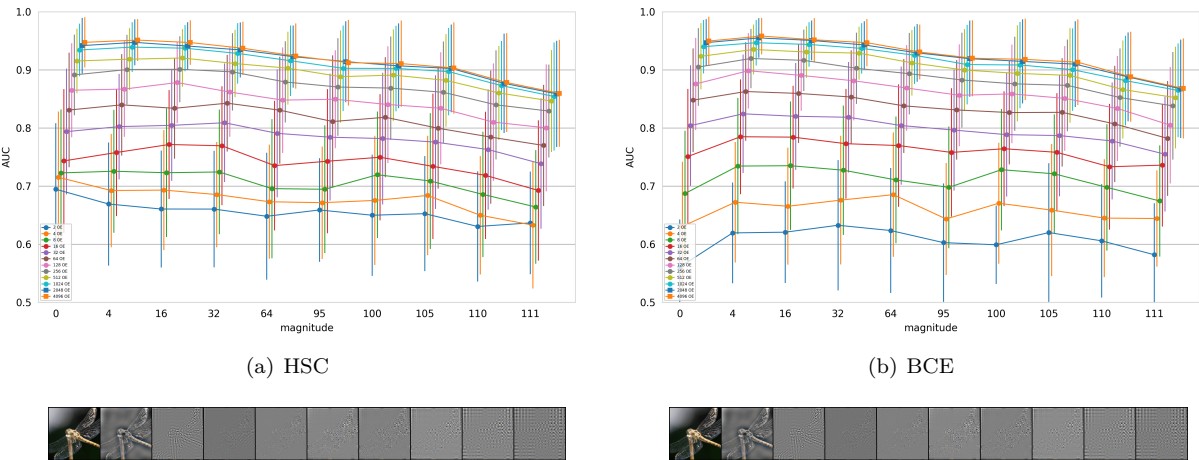

Figure 12: HPF ImageNet-30 with ImageNet-22k (with ImageNet-1k removed) OE AD benchmark.

## D  OE robustness results with random search

In Section 5.4 we use an evolutionary algorithm to find the best and worst single OE samples for AD. The main difference between an evolutionary algorithm and a pure random search is that the former "fine-tunes" samples by picking the next ones depending on the best-performing previous ones. However, the algorithm also explores completely new samples. To show that this is the case and the algorithm doesn't quickly converge to a local optimum, we here present results using a pure random search for the best single OE samples and compare those to the ones in the main paper.

Table 5 shows the mean AUC over ten classes (analogue to Table 3 in the main paper and Table 4 in Appendix B). We see that pure random search consistently yields slightly worse samples on CIFAR-10 and mostly very similar performing samples on ImageNet-10. There is a minor exception for unfiltered images with BCE, where random search found on average 2% better performing single OE samples, resulting in the same conclusion, though. Figure 13 shows examples for the best samples found via a random search for unfiltered and filtered images. We see that the results are quite similar to those found via the evolutionary algorithm.

Table 5: Mean AUC detection performance in % for the best single OE samples on the CIFAR-10 AD benchmark using 80MTI as OE and on the ImageNet-10 AD benchmark using ImageNet-22K (with the 1K classes removed) as OE. All images have been either unfiltered, high-pass-filtered (HPF), or low-pass-filtered (LPF). Here we use pure random search instead of an evolutionary algorithm.

|  | CIFAR-10 | | ImageNet | |
|---|---|---|---|---|
|  | HSC | BCE | HSC | BCE |
| Best OE | 76.5 | 68.7 | 79.5 | 77.8 |
| Best OE LPF | 77.0 | 66.6 | 76.7 | 73.3 |
| Best OE HPF | 68.1 | 65.9 | 75.0 | 77.2 |

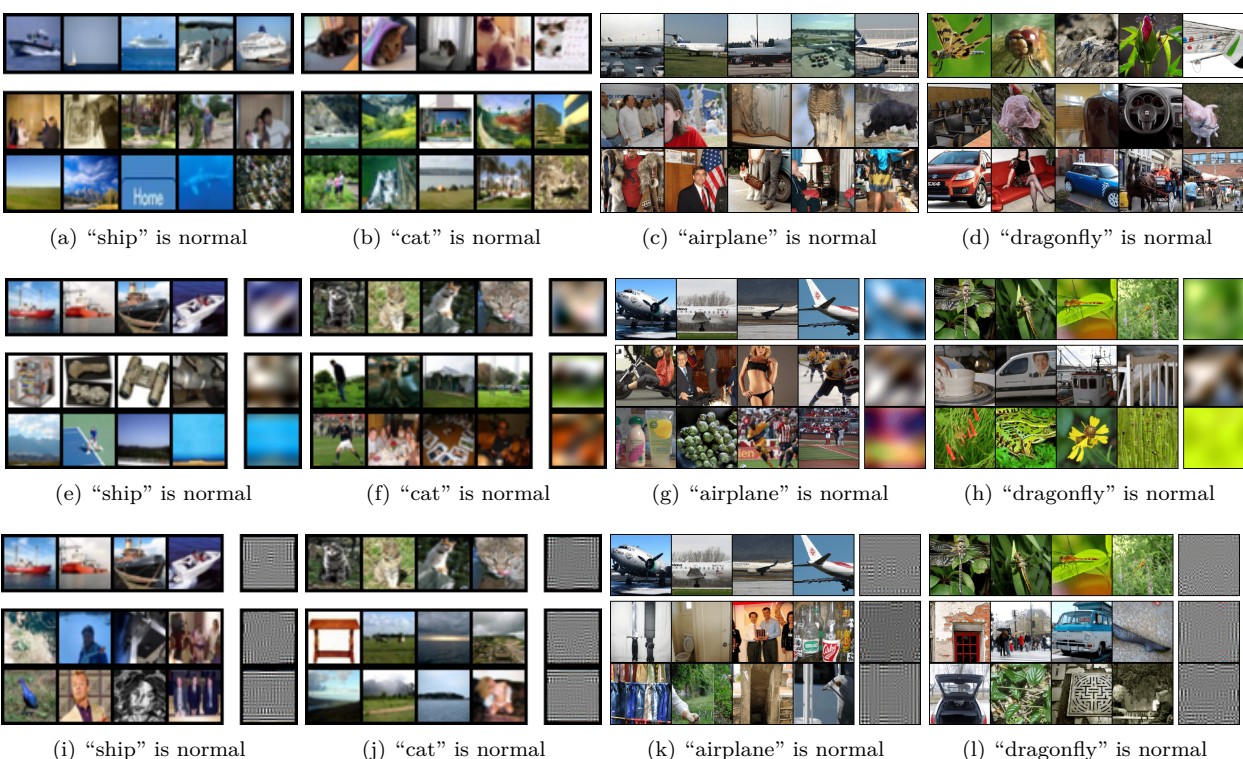

(a) "ship" is normal    (b) "cat" is normal    (c) "airplane" is normal    (d) "dragonfly" is normal

(e) "ship" is normal    (f) "cat" is normal    (g) "airplane" is normal    (h) "dragonfly" is normal

(i) "ship" is normal    (j) "cat" is normal    (k) "airplane" is normal    (l) "dragonfly" is normal

Figure 13: Best OE samples for CIFAR-10 with 80MTI as OE (a-b, e-f, i-j) and ImageNet-10 with ImageNet-22k as OE (c-d, g-h, k-l). In each figure, the first row shows normal samples, and the next two rows the best samples found via HSC (top) and BCE (bottom). In Figures (e-h), all samples (train, test, anomalous, and normal) are low-pass-filtered, in Figures (i-l) they are high-pass-filtered. These figures also contain a separate last column showing the filtered version of the images, which is what the network sees during training or testing. Here we use pure random search instead of an evolutionary algorithm.

## E Hypersphere Classifier sensitivity analysis

In this section, we show results for the Hypersphere Classifier (HSC) (Section 3.2) when varying the radial function $l(\boldsymbol{z}) = \exp\left(-h(\boldsymbol{z})\right)$. For this, we run the CIFAR-10 one vs. rest benchmark with 80MTI as OE, as presented in Table 1 in the main paper, for different functions $h : \mathbb{R}^r \to [0, \infty), \boldsymbol{z} \mapsto h(\boldsymbol{z})$. We also alter training to be with or without data augmentation in these experiments. The results are presented in Table 6. We see that data augmentation leads to an improvement in performance even in the case where we have the full 80MTI dataset as OE. HSC shows the overall best performance with data augmentation and using the robust Pseudo-Huber loss $h(\boldsymbol{z}) = \sqrt{\|\boldsymbol{z}\|^2 + 1} - 1$.

Table 6: Mean AUC detection performance in % (over 10 seeds) on the CIFAR-10 one vs. rest benchmark using 80MTI as OE for different choices of $h(\boldsymbol{z})$ in the radial function $l$ of HSC.

| Data augment. | $\|\boldsymbol{z}\|_1$ | $\|\boldsymbol{z}\|_2$ | $\|\boldsymbol{z}\|_2^2$ | $\sqrt{\|\boldsymbol{z}\|^2 + 1} - 1$ |
|---|---|---|---|---|
| w/o | 90.6 | 92.3 | 89.1 | 91.8 |
| w/ | 92.5 | 94.1 | 94.5 | 96.1 |

## F    Focal Loss with varying $\gamma$

Here we include results showing how mean AUC detection performance changes with $\gamma$ on the Focal loss. Since we balance every batch to contain 128 normal and 128 OE samples during training, we set the weighting factor $\alpha$ to be $\alpha = 0.5$ (Lin et al., 2017). Again, note that $\gamma = 0$ corresponds to standard binary cross entropy. Figure 14 shows that mean AUC performance is insensitive to the choice of $\gamma$ on the CIFAR-10 and ImageNet-30 one vs. rest benchmarks.

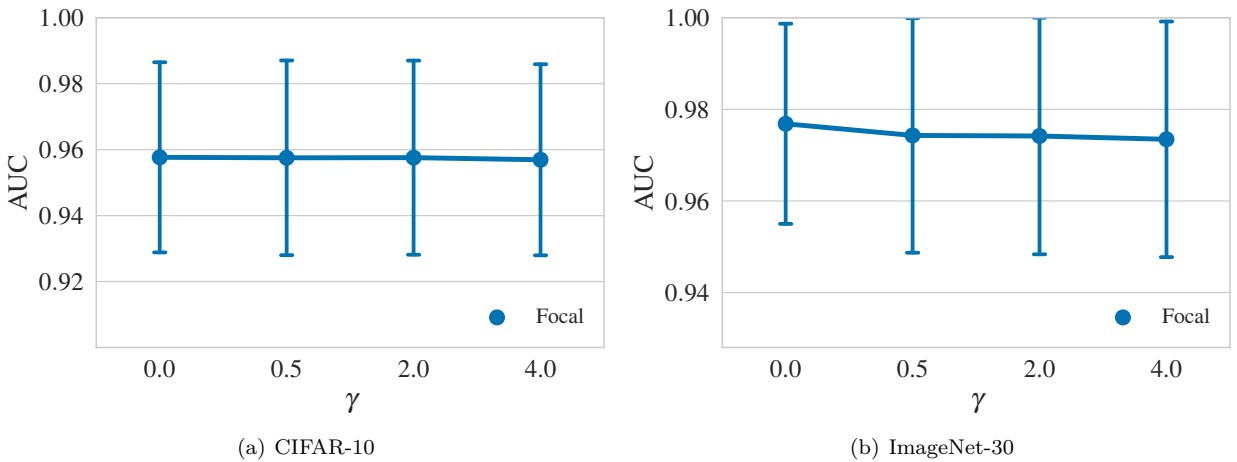

(a) CIFAR-10                                    (b) ImageNet-30

Figure 14: Focal loss detection performance in mean AUC in % when varying $\gamma$ on the CIFAR-10 with 80MTI OE (a) and ImageNet-30 with ImageNet-22K OE (b) one vs. rest benchmarks.

## G    Network architectures and optimization

We provide details on the network architectures and optimization below, where we distinguish between end-to-end methods and the ones that use transfer learning (CLIP-based).

### G.1    End-to-end methods

We always use the same underlying network $\phi_\theta$ in each experimental setting for our HSC, Deep SAD, Focal, and BCE implementations to control architectural effects. For Focal and BCE, the output of the network $\phi_\theta$ is followed by a linear layer with sigmoid activation. For the experiments on CIFAR-10 and (Fashion-)MNIST we use standard LeNet-style networks (LeCun et al., 1990) having three convolutional layers followed by two fully connected layers. We use batch normalization (Ioffe & Szegedy, 2015) and (leaky) ReLU activations in these networks. For our experiments on ImageNet, CUB, DTD, and MVTec-AD we use the same WideResNet (Zagoruyko & Komodakis, 2016) as Hendrycks et al. (2019b), which has ResNet-18 as its architectural backbone. We use Adam (Kingma & Ba, 2015) for optimization and balance every batch to contain 128 normal and 128 OE samples during training. For data augmentation, we use standard color jitter, random cropping, horizontal flipping, and Gaussian pixel noise. Due to computational constraints, there is

an exception for the OE robustness experiments (Section 5.4), the frequency analysis (Appendices B and C), and the leave-one-class-out experiment with varying OE sizes (Figure 4), where we use no augmentations for CIFAR-10 and only random cropping for ImageNet. We provide further dataset-specific details below.

**CIFAR-10**  On CIFAR-10, we use LeNet-style networks having three convolutional layers and two fully connected layers. Each convolutional layer is followed by batch normalization, a leaky ReLU activation, and max-pooling. The first fully connected layer is followed by batch normalization, and a leaky ReLU activation, while the last layer is just a linear transformation. The number of kernels in the convolutional layers are, from first to last: 32-64-128. The fully connected layers have 512-256 units respectively. We use Adam (Kingma & Ba, 2015) for optimization and balance every batch to contain 128 normal and 128 OE samples during training. We train for 200 epochs starting with a learning rate of $\eta = 0.001$ and have learning rate milestones at 100 and 150 epochs. The learning rate is reduced by a factor of 10 at every milestone. For the OE robustness experiments (Section 5.4), the frequency analysis (Appendices B and C), and the leave-one-class-out experiment with varying OE sizes (Figure 4), we trained for 30 epochs with a milestone at 25 instead.

**ImageNet**  On ImageNet, we use exactly the same WideResNet (Zagoruyko & Komodakis, 2016) as was used in Hendrycks et al. (2019b), which has a ResNet-18 as architectural backbone. We use Adam (Kingma & Ba, 2015) for optimization and balance every batch to contain 128 normal and 128 OE samples during training. We train for 150 epochs starting with a learning rate of $\eta = 0.001$ and milestones at epochs 100 and 125. The learning rate is reduced by a factor of 10 at every milestone. For the OE robustness experiments (Section 5.4), the frequency analysis (Appendices B and C), and the leave-one-class-out experiment with varying OE sizes (Figure 4) we trained for 30 epochs with a milestone at 25 instead.

**CUB**  On CUB, we use the same setup as we used on ImageNet. However, we balance every batch to contain 30 normal and 30 OE samples during training since there are only 30 training samples per class. Also, we used a subset of ImageNet-22k where we sampled 2 images per ImageNet-22k class as OE. This speeds up data loading but should not have an impact on the performance because with 150 epochs and 30 training samples the trainer only sees up to 4500 OE samples.

**DTD**  On DTD, similar to CUB, we use the same setup as we used on ImageNet but balance every batch to contain 40 normal and 40 OE samples during training since there are only 40 training samples per class. Again, we use the subset of ImageNet-22k as OE.

**MVTec-AD**  On MVTec-AD, we use almost the same setup as we used on ImageNet. However, we train for 300 epochs starting with a learning rate of $\eta = 0.001$ and milestones at epochs 200 and 250.

**Fashion-MNIST**  On Fashion-MNIST, we use a similar setup as we used on CIFAR-10. Here the network consists of only two convolutional layers with 16 and 32 kernels, respectively.

**MNIST**  On MNIST, we use the same setup as on Fashion-MNIST but don't use any data augmentation.

### G.2  CLIP

Apart from the following changes, we use the same setup for our CLIP-based implementations as for the end-to-end methods. One of the changes is that we use the pre-trained CLIP network architecture (Radford et al., 2021). For fine-tuning CLIP with a BEC classifier, we use SGD with Nesterov momentum instead of ADAM, train for 80 epochs starting with a learning rate of $\eta = 0.0001$, and have learning rate milestones at 50, 60, 70, and 75 epochs. The learning rate is reduced by a factor of 10 at every milestone. This applies completely to ImageNet and MVTec-AD. For CIFAR-10, CUB, DTD, Fashion-MNIST, and MNIST, we do the same but start with a learning rate of $\eta = 0.00002$ instead.

# H    The evolutionary algorithm for finding OE samples

In Section 5.4 we search for a sample in the OE dataset that is either particularly harmful or useful for the AD model. A common approach to perform a discrete search in a large set is to use evolutionary algorithms (Yu & Gen, 2010; Fortin et al., 2012). We here stick to a simple version that uses tournament selection (Blickle & Thiele, 1995) with three competitors. In our scenario, the individuals are images of the OE dataset and randomly initialized.

---

**Algorithm 1** Evolve OE samples

---

**Input:** AD model $\phi$, OE dataset $D$, normal train set $X$, full test set $X_{test}$.
**Output:** A collection of optimal single training outliers (individuals) $D'$.
**Define:**

  COIN($p$): returns TRUE with chance $p$, else FALSE.
  RND($D, k$): selects $k$ samples in set $D$ randomly.
  TRAIN($\phi, d$): randomly initializes $\phi$'s weights, then trains $\phi$ with normal dataset $X$ and the single training outlier $d$ as OE.
  EVAL($\phi$): computes and returns the test AUC of $\phi$ using the one vs. rest approach on $X_{test}$.

**Algorithm:**

  # Initialize:
  $D' \leftarrow$ RND($D, 64$)
  **for all** $d' \in D'$ **do**
    TRAIN($\phi, d'$)
    $a \leftarrow$ EVAL($\phi$)
    Set fitness of $d'$ to $a$
  **end for**

  **for** 50 iterations **do**

    # Select:
    $D_{temp} \leftarrow D'$
    **for all** $d_{temp} \in D_{temp}$ **do**
      Replace $d_{temp}$ in $D_{temp}$ with the best in RND($D', 3$).    [best according to fitness]
    **end for**
    $D' \leftarrow D_{temp}$

    # Mate:
    **for all** $i \in [0, 2, 4, ..., |D'|]$ **do**
      **if** COIN($0.05$) **then**
        $d'_1 \leftarrow D'[i]$
        $d'_2 \leftarrow D'[i+1]$
        **for all** $j \in [0, 1]$ **do**
          $P \leftarrow$ RND($D, 10000$)
          sort($P$) according to ($\|p - d'_1\|^2 + \|p - d'_2\|^2$) for every $p \in P$.
          $P \leftarrow P[: 50]$, get the 50 samples with least average distance to $d'_1$ and $d'_2$.
          Replace $D'[i+j]$ with RND($P, 1$).
        **end for**
      **end if**
    **end for**

    # Mutate:
    **for all** $d' \in D'$ **do**
      **if** COIN($0.55$) **then**
        $P \leftarrow$ RND($D, 10000$)
        sort($P$) according to $\|p - d'\|^2$ for every $p \in P$.
        $P \leftarrow P[: 50]$, get the 50 samples with least distance to $d'$.
        Replace $d'$ in $D'$ with RND($P, 1$).
      **end if**
    **end for**

    # Evaluate:
    **for all** $d' \in D'$ **do**
      TRAIN($\phi, d'$)
      $a \leftarrow$ EVAL($\phi$)
      Set fitness of $d'$ to $a$
    **end for**

  **end for**

---

Mutating an individual works by first sampling random images from the OE set and then randomly picking one among those 50 in that subset that have the least $L^2$ distance to the individual. Mating works similarly, it picks one with the least $L^2$ distance to both parents. The fitness of an individual is the test AUC of an AD model trained with the full normal dataset and just the individual as the only training outlier. The algorithm's objective is to either maximize ("best" OE samples) or minimize ("worst" OE samples) the average fitness of its final generation.

This experimental setup is exceptionally computationally expensive, with the evaluation of a single individual's fitness requiring a full training of a neural network. Because of this we decrease the number of training epochs, restrict the augmentations, and average over just two different random seeds to evaluate an individual's fitness (see Appendix G). On ImageNet, we only consider the first 10 classes (see Appendix I) out of the 30 that are used in Hendrycks et al. (2019a), but evaluate these with the full 30-classes one vs. rest benchmark.

Algorithm 1 provides a detailed explanation of the evolutionary algorithm. For the sake of readability, we fix the evolutionary hyperparameters to the values used in our experiments. We set the generation size to 64, the number of generations to 50, the tournament size to 3, the mating chance to 0.05, the mutation chance to 0.55, the initial candidate pool size to 10000, and the final candidate pool size to 50. We apply this algorithm to each class separately, thereby changing the normal set $X$ and test set $X_{test}$. For finding the worst single OE samples instead of the best ones, we change the selection procedure to simply replace $d_{temp}$ with the worst in $\text{RND}(D', 3)$ instead of the best.

# I  Results on individual classes

We report results averaged over all classes in the main paper. Here we provide full class-wise results for all experiments of Sections 5.1, 5.3, 5.4, and Appendix B. We also include figures for each class for the varying amount of OE experiment (Section 5.2) for CIFAR-10 and ImageNet-30.

## I.1  Image AD with end-to-end methods on the one vs. rest benchmark

Here, we report results for the one vs. rest benchmark experiments from Section 5.1 for all individual classes and *end-to-end* methods. For CIFAR-10, we report results in Table 7, including results from the literature. For CIFAR-10, ImageNet-30, Fashion-MNIST, and MNIST, we report results with standard deviations for our implementations in Tables 8, 9, 10, and 11, respectively. For CUB and DTD, we report class-wise results instead in form of a plot in Figures 15 and 16, as there are too many classes (47 for DTD and 200 for CUB) to report concisely in a Table.

Table 7: Mean AUC detection performance in % (over 10 seeds) for all individual classes and end-to-end methods on the CIFAR-10 one vs. rest benchmark with 80MTI OE from Section 5.1.

| Class | Unsupervised | | | | | Unsupervised OE | | | Supervised OE | | |
|---|---|---|---|---|---|---|---|---|---|---|---|
| | SVDD* | DSVDD* | GT* | GT+* | CSI* | GT+* | DSAD | HSC | Focal* | Focal | BCE |
| Airplane | 65.6 | 61.7 | 74.7 | 77.5 | 89.9 | 90.4 | 94.2 | 96.3 | 87.6 | 95.9 | **96.4** |
| Automobile | 40.9 | 65.9 | 95.7 | 96.9 | 99.1 | **99.3** | 98.1 | 98.7 | 93.9 | 98.7 | 98.8 |
| Bird | 65.3 | 50.8 | 78.1 | 87.3 | 93.1 | **93.7** | 89.8 | 92.7 | 78.6 | 92.3 | 93.0 |
| Cat | 50.1 | 59.1 | 72.4 | 80.9 | 86.4 | 88.1 | 87.4 | 89.8 | 79.9 | 88.8 | **90.0** |
| Deer | 75.2 | 60.9 | 87.8 | 92.7 | 93.9 | **97.4** | 95.0 | 96.6 | 81.7 | 96.6 | 97.1 |
| Dog | 51.2 | 65.7 | 87.8 | 90.2 | 93.2 | **94.3** | 93.0 | 94.2 | 85.6 | 94.1 | 94.2 |
| Frog | 71.8 | 67.7 | 83.4 | 90.9 | 95.1 | 97.1 | 96.9 | 97.9 | 93.3 | 97.8 | **98.0** |
| Horse | 51.2 | 67.3 | 95.5 | 96.5 | 98.7 | **98.8** | 96.8 | 97.6 | 87.9 | 97.6 | 97.6 |
| Ship | 67.9 | 75.9 | 93.3 | 95.2 | 97.9 | **98.7** | 97.1 | 98.2 | 92.6 | 98.0 | 98.1 |
| Truck | 48.5 | 73.1 | 91.3 | 93.3 | 95.5 | **98.5** | 96.2 | 97.4 | 92.1 | 97.5 | 97.7 |
| Mean AUC | 58.8 | 64.8 | 86.0 | 90.1 | 94.3 | 95.6 | 94.5 | 95.9 | 87.3 | 95.8 | **96.1** |

Table 8: Mean AUC detection performance in % (over 10 seeds) *with standard deviations* for all individual classes for our end-to-end implementations on the CIFAR-10 one vs. rest benchmark with 80MTI OE from Section 5.1.

| Class | Unsupervised OE | | Supervised OE | |
|---|---|---|---|---|
| | DSAD | HSC | Focal | BCE |
| Airplane | 94.2 ± 0.34 | 96.3 ± 0.13 | 95.9 ± 0.11 | 96.4 ± 0.17 |
| Automobile | 98.1 ± 0.19 | 98.7 ± 0.07 | 98.7 ± 0.09 | 98.8 ± 0.06 |
| Bird | 89.8 ± 0.54 | 92.7 ± 0.27 | 92.3 ± 0.32 | 93.0 ± 0.14 |
| Cat | 87.4 ± 0.38 | 89.8 ± 0.27 | 88.8 ± 0.33 | 90.0 ± 0.27 |
| Deer | 95.0 ± 0.22 | 96.6 ± 0.17 | 96.6 ± 0.10 | 97.1 ± 0.10 |
| Dog | 93.0 ± 0.30 | 94.2 ± 0.13 | 94.1 ± 0.21 | 94.2 ± 0.12 |
| Frog | 96.9 ± 0.22 | 97.9 ± 0.08 | 97.8 ± 0.07 | 98.0 ± 0.09 |
| Horse | 96.8 ± 0.15 | 97.6 ± 0.10 | 97.6 ± 0.16 | 97.6 ± 0.09 |
| Ship | 97.1 ± 0.21 | 98.2 ± 0.09 | 98.0 ± 0.11 | 98.1 ± 0.08 |
| Truck | 96.2 ± 0.22 | 97.4 ± 0.13 | 97.5 ± 0.12 | 97.7 ± 0.16 |
| Mean AUC | 94.5 ± 3.30 | 95.9 ± 2.68 | 95.8 ± 2.97 | 96.1 ± 2.71 |

Table 9: Mean AUC detection performance in % (over 10 seeds) for all individual classes for our end-to-end implementations on the ImageNet-30 one vs. rest benchmark with ImageNet-22K OE from Section 5.1. Note that for GT+*, Focal*, and CSI* as reported in Table 1 in the main paper, Hendrycks et al. (2019b) and Tack et al. (2020) do not provide results on a per class basis.

| Class | Unsupervised DSVDD | Unsupervised OE | | Supervised OE | |
|---|---|---|---|---|---|
| | | DSAD | HSC | Focal | BCE |
| Acorn | $62.7 \pm 2.94$ | $98.5 \pm 0.28$ | $98.8 \pm 0.42$ | $99.0 \pm 0.15$ | $99.0 \pm 0.19$ |
| Airliner | $55.8 \pm 1.76$ | $99.6 \pm 0.16$ | $99.8 \pm 0.10$ | $99.9 \pm 0.02$ | $99.8 \pm 0.04$ |
| Ambulance | $47.3 \pm 3.58$ | $99.0 \pm 0.13$ | $99.8 \pm 0.13$ | $99.2 \pm 0.14$ | $99.9 \pm 0.07$ |
| American alligator | $73.0 \pm 0.65$ | $92.9 \pm 1.06$ | $98.0 \pm 0.32$ | $94.7 \pm 0.67$ | $98.2 \pm 0.27$ |
| Banjo | $56.8 \pm 2.22$ | $97.0 \pm 0.51$ | $98.2 \pm 0.41$ | $97.0 \pm 0.33$ | $98.7 \pm 0.22$ |
| Barn | $67.6 \pm 1.32$ | $98.5 \pm 0.29$ | $99.8 \pm 0.05$ | $98.7 \pm 0.24$ | $99.8 \pm 0.08$ |
| Bikini | $59.7 \pm 2.81$ | $96.5 \pm 0.84$ | $98.6 \pm 0.57$ | $97.2 \pm 0.89$ | $99.1 \pm 0.30$ |
| Digital clock | $61.2 \pm 2.08$ | $99.4 \pm 0.33$ | $96.8 \pm 0.79$ | $99.8 \pm 0.03$ | $97.2 \pm 0.29$ |
| Dragonfly | $61.9 \pm 3.02$ | $98.8 \pm 0.28$ | $98.4 \pm 0.16$ | $99.1 \pm 0.21$ | $98.3 \pm 0.04$ |
| Dumbbell | $51.3 \pm 1.09$ | $93.0 \pm 0.53$ | $91.6 \pm 0.88$ | $94.0 \pm 0.04$ | $92.6 \pm 0.97$ |
| Forklift | $48.0 \pm 5.00$ | $90.6 \pm 1.43$ | $99.1 \pm 0.33$ | $94.2 \pm 0.90$ | $99.5 \pm 0.09$ |
| Goblet | $63.3 \pm 0.25$ | $92.4 \pm 1.05$ | $93.8 \pm 0.38$ | $93.8 \pm 0.27$ | $94.7 \pm 1.43$ |
| Grand piano | $58.4 \pm 0.16$ | $99.7 \pm 0.06$ | $97.4 \pm 0.37$ | $99.9 \pm 0.04$ | $97.6 \pm 0.34$ |
| Hotdog | $62.0 \pm 2.46$ | $95.9 \pm 2.01$ | $98.5 \pm 0.34$ | $97.2 \pm 0.05$ | $98.8 \pm 0.34$ |
| Hourglass | $60.2 \pm 2.93$ | $96.3 \pm 0.37$ | $96.9 \pm 0.26$ | $97.5 \pm 0.17$ | $97.6 \pm 0.48$ |
| Manhole cover | $63.9 \pm 0.56$ | $98.5 \pm 0.29$ | $99.6 \pm 0.34$ | $99.2 \pm 0.09$ | $99.8 \pm 0.01$ |
| Mosque | $73.8 \pm 0.35$ | $98.6 \pm 0.29$ | $99.1 \pm 0.26$ | $98.9 \pm 0.30$ | $99.3 \pm 0.15$ |
| Nail | $49.4 \pm 1.05$ | $92.8 \pm 0.80$ | $94.0 \pm 0.76$ | $93.5 \pm 0.32$ | $94.5 \pm 1.37$ |
| Parking meter | $46.3 \pm 2.64$ | $98.5 \pm 0.29$ | $93.3 \pm 1.64$ | $99.3 \pm 0.04$ | $94.7 \pm 0.76$ |
| Pillow | $40.8 \pm 1.22$ | $99.3 \pm 0.14$ | $94.0 \pm 0.47$ | $99.2 \pm 0.14$ | $94.2 \pm 0.42$ |
| Revolver | $50.6 \pm 1.59$ | $98.2 \pm 0.30$ | $97.6 \pm 0.25$ | $98.6 \pm 0.11$ | $97.7 \pm 0.68$ |
| Rotary dial telephone | $59.1 \pm 1.68$ | $90.4 \pm 1.99$ | $97.7 \pm 0.50$ | $92.2 \pm 0.33$ | $98.3 \pm 0.75$ |
| Schooner | $78.0 \pm 1.94$ | $99.1 \pm 0.18$ | $99.2 \pm 0.20$ | $99.6 \pm 0.02$ | $99.1 \pm 0.26$ |
| Snowmobile | $70.9 \pm 0.79$ | $97.7 \pm 0.86$ | $99.0 \pm 0.22$ | $98.1 \pm 0.15$ | $99.1 \pm 0.25$ |
| Soccer ball | $56.0 \pm 1.68$ | $97.3 \pm 1.70$ | $92.9 \pm 1.18$ | $98.6 \pm 0.13$ | $93.6 \pm 0.61$ |
| Stingray | $79.5 \pm 0.98$ | $99.3 \pm 0.20$ | $99.1 \pm 0.33$ | $99.7 \pm 0.04$ | $99.2 \pm 0.10$ |
| Strawberry | $65.0 \pm 3.32$ | $97.7 \pm 0.64$ | $99.1 \pm 0.20$ | $99.1 \pm 0.03$ | $99.2 \pm 0.22$ |
| Tank | $69.5 \pm 1.83$ | $97.3 \pm 0.51$ | $98.6 \pm 0.18$ | $97.3 \pm 0.47$ | $98.9 \pm 0.13$ |
| Toaster | $53.1 \pm 0.42$ | $97.7 \pm 0.56$ | $92.2 \pm 0.78$ | $98.3 \pm 0.05$ | $92.2 \pm 0.65$ |
| Volcano | $88.0 \pm 3.27$ | $89.6 \pm 0.44$ | $99.5 \pm 0.09$ | $91.6 \pm 0.90$ | $99.4 \pm 0.19$ |
| Mean | $61.1 \pm 10.61$ | $96.7 \pm 2.98$ | $97.3 \pm 2.53$ | $97.5 \pm 2.43$ | $97.7 \pm 2.34$ |

Table 10: Mean AUC detection performance in % (over 10 seeds) for all individual classes for our end-to-end implementations on the Fashion-MNIST one vs. rest benchmark with grayscale CIFAR-100 OE from Section 5.1.

| Class | Unsupervised DSVDD | Unsupervised OE | | Supervised OE | |
|---|---|---|---|---|---|
| | | DSAD | HSC | Focal | BCE |
| Top | $82.8 \pm 1.10$ | $85.2 \pm 2.00$ | $83.5 \pm 4.13$ | $78.6 \pm 2.41$ | $75.3 \pm 2.12$ |
| Trouser | $94.8 \pm 4.92$ | $96.1 \pm 1.90$ | $98.1 \pm 0.18$ | $95.9 \pm 1.30$ | $93.5 \pm 0.81$ |
| Pullover | $79.3 \pm 2.42$ | $82.4 \pm 0.95$ | $83.6 \pm 2.59$ | $88.9 \pm 1.55$ | $89.4 \pm 1.37$ |
| Dress | $88.8 \pm 2.22$ | $88.1 \pm 1.96$ | $88.0 \pm 2.26$ | $84.5 \pm 2.04$ | $82.6 \pm 2.88$ |
| Coat | $82.9 \pm 3.71$ | $84.1 \pm 2.04$ | $85.7 \pm 1.71$ | $86.2 \pm 0.62$ | $83.1 \pm 2.20$ |
| Sandal | $75.1 \pm 9.60$ | $91.2 \pm 1.18$ | $90.5 \pm 0.76$ | $92.7 \pm 0.97$ | $91.7 \pm 2.09$ |
| Shirt | $75.5 \pm 1.02$ | $73.0 \pm 2.40$ | $74.5 \pm 3.16$ | $78.4 \pm 0.74$ | $74.7 \pm 1.58$ |
| Sneaker | $96.6 \pm 0.53$ | $96.0 \pm 0.65$ | $97.1 \pm 0.49$ | $95.1 \pm 0.69$ | $95.0 \pm 0.31$ |
| Bag | $88.9 \pm 2.67$ | $75.9 \pm 5.94$ | $78.3 \pm 3.86$ | $88.0 \pm 0.53$ | $89.4 \pm 2.01$ |
| Ankle boot | $98.2 \pm 0.23$ | $91.6 \pm 1.91$ | $94.0 \pm 1.36$ | $88.9 \pm 1.78$ | $89.8 \pm 1.79$ |
| Mean | $86.3 \pm 8.06$ | $86.4 \pm 7.44$ | $87.3 \pm 7.38$ | $87.7 \pm 5.76$ | $86.4 \pm 6.84$ |

Table 11: Mean AUC detection performance in % (over 10 seeds) for all individual classes for our end-to-end implementations on the MNIST one vs. rest benchmark with EMNIST OE from Section 5.1.

| | Unsupervised | Unsupervised OE | | Supervised OE | |
|---|---|---|---|---|---|
| Class | DSVDD | DSAD | HSC | Focal | BCE |
| Zero | $99.0 \pm 0.27$ | $99.3 \pm 0.39$ | $99.6 \pm 0.03$ | $99.4 \pm 0.34$ | $99.4 \pm 0.16$ |
| One | $99.8 \pm 0.02$ | $99.7 \pm 0.12$ | $99.9 \pm 0.02$ | $99.7 \pm 0.18$ | $99.3 \pm 0.34$ |
| Two | $92.3 \pm 1.17$ | $98.0 \pm 0.12$ | $98.0 \pm 0.49$ | $97.6 \pm 0.68$ | $97.5 \pm 0.23$ |
| Three | $94.9 \pm 0.36$ | $98.6 \pm 0.35$ | $98.5 \pm 0.94$ | $99.5 \pm 0.10$ | $99.6 \pm 0.13$ |
| Four | $94.4 \pm 0.50$ | $96.5 \pm 1.02$ | $96.7 \pm 0.74$ | $97.5 \pm 1.01$ | $98.2 \pm 0.52$ |
| Five | $94.2 \pm 1.35$ | $97.6 \pm 0.24$ | $97.8 \pm 0.90$ | $97.9 \pm 0.25$ | $97.3 \pm 0.69$ |
| Six | $99.0 \pm 0.09$ | $99.6 \pm 0.08$ | $99.6 \pm 0.08$ | $99.7 \pm 0.07$ | $99.7 \pm 0.07$ |
| Seven | $96.6 \pm 0.57$ | $97.1 \pm 0.39$ | $97.4 \pm 0.63$ | $97.6 \pm 0.65$ | $97.7 \pm 0.45$ |
| Eight | $90.9 \pm 1.47$ | $93.5 \pm 2.49$ | $96.6 \pm 0.81$ | $97.3 \pm 0.32$ | $97.3 \pm 0.57$ |
| Nine | $96.9 \pm 0.36$ | $96.3 \pm 1.06$ | $97.6 \pm 0.27$ | $97.8 \pm 0.31$ | $97.7 \pm 0.57$ |
| Mean | $95.8 \pm 2.83$ | $97.6 \pm 1.81$ | $98.2 \pm 1.14$ | $98.4 \pm 0.97$ | $98.4 \pm 0.95$ |

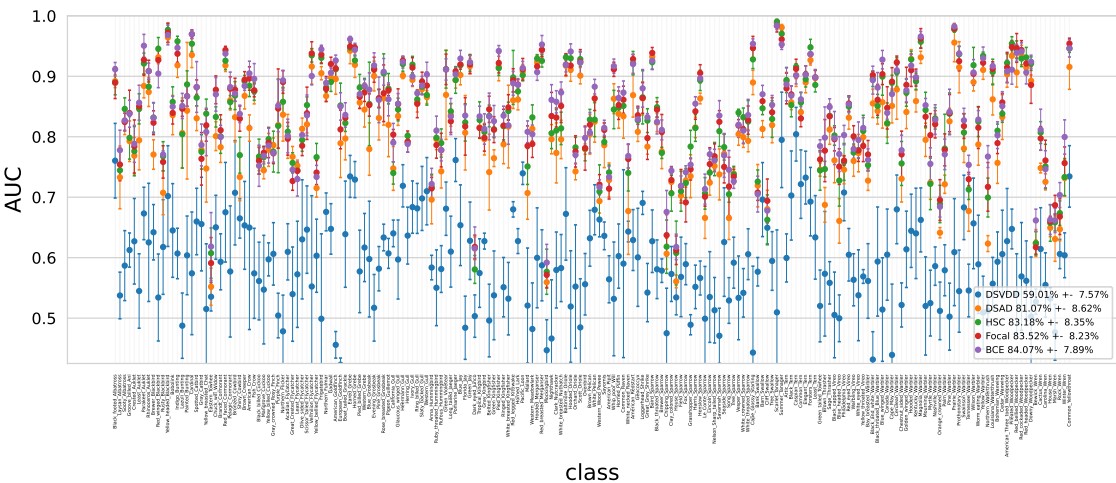

Figure 15: Mean AUC detection performance in % (over 10 seeds) for all individual classes for our end-to-end implementations on the CUB one vs. rest benchmark with ImageNet-22k OE from Section 5.1.

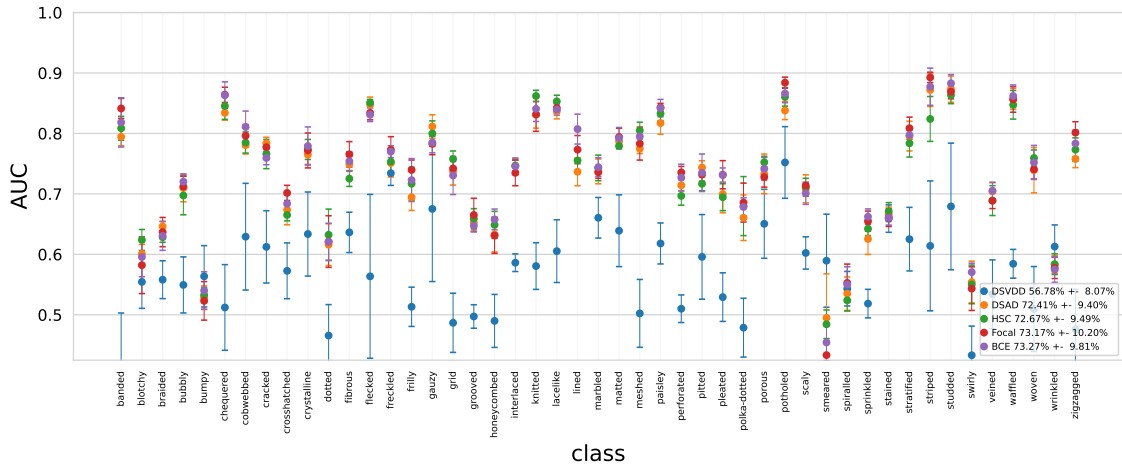

Figure 16: Mean AUC detection performance in % (over 10 seeds) for all individual classes for our end-to-end implementations on the DTD one vs. rest benchmark with ImageNet-22k OE from Section 5.1.

**I.2 Image AD with transfer learning-based methods on the one vs. rest benchmark**

Here we report results for the one vs. rest benchmark experiments from Section 5.3 for all individual classes and *transfer learning-based* methods. For CIFAR-10, we report results in Table 12, including results from the literature (left) and results with standard deviations for our implementation (right). For ImageNet-30, Fashion-MNIST, and MNIST, we report results with standard deviations for our implementations in Tables 14, 13, and 15, respectively. For CUB and DTD, we report class-wise results instead in form of a plot in Figures 17 and 18, as there are too many classes (47 for DTD and 200 for CUB) to report concisely in a Table.

Table 12: Left: Mean AUC detection performance in % (over 10 seeds) for all individual classes and transfer learning-based methods on the CIFAR-10 one vs. rest benchmark with 80MTI OE from Section 5.3. Right: Our transfer learning-based implementations for the same setup.

| | Unsupervised | | | Supervised OE | | | | Unsupervised | Supervised OE |
| Class | DN2* | PANDA* | CLIP | PANDA* | ADIP* | BCE-CL | Class | CLIP | BCE-CL |
|---|---|---|---|---|---|---|---|---|---|
| Airplane | 93.9 | × | 99.4 | × | 99.2 | **99.7** | Airplane | 99.40 ± 0.00 | 99.74 ± 0.02 |
| Automobile | 97.7 | × | 99.4 | × | **99.8** | **99.8** | Automobile | 99.37 ± 0.00 | 99.82 ± 0.01 |
| Bird | 85.5 | × | 97.4 | × | 98.6 | **99.2** | Bird | 97.37 ± 0.00 | 99.29 ± 0.04 |
| Cat | 85.5 | × | 97.0 | × | 97.0 | **98.8** | Cat | 96.99 ± 0.00 | 98.86 ± 0.04 |
| Deer | 93.6 | × | 98.1 | × | 99.3 | **99.6** | Deer | 98.05 ± 0.00 | 99.62 ± 0.02 |
| Dog | 91.3 | × | 97.4 | × | 98.2 | **99.2** | Dog | 97.36 ± 0.00 | 99.26 ± 0.04 |
| Frog | 94.3 | × | 98.1 | × | 99.6 | **99.9** | Frog | 98.05 ± 0.00 | 99.89 ± 0.01 |
| Horse | 93.6 | × | 99.0 | × | **99.8** | **99.8** | Horse | 98.99 ± 0.00 | 99.84 ± 0.02 |
| Ship | 95.1 | × | 99.7 | × | 99.6 | **99.8** | Ship | 99.71 ± 0.00 | 99.86 ± 0.01 |
| Truck | 95.3 | × | 99.3 | × | 99.5 | **99.9** | Truck | 99.30 ± 0.00 | 99.90 ± 0.01 |
| Mean AUC | 92.5 | 96.2 | 98.5 | 98.9 | 99.1 | **99.6** | Mean AUC | 98.46 ± 0.96 | 99.61 ± 0.34 |

Table 13: Mean AUC detection performance in % (over 10 seeds) for all individual classes for our transfer learning-based implementations on the Fashion-MNIST one vs. rest benchmark with grayscale CIFAR-100 OE from Section 5.3.

| | Unsupervised | Supervised OE |
| Class | CLIP | BCE-CL |
|---|---|---|
| Top | 82.9 ± 0.00 | 84.7 ± 0.34 |
| Trouser | 87.7 ± 0.00 | 98.5 ± 0.03 |
| Pullover | 91.6 ± 0.00 | 93.8 ± 0.06 |
| Dress | 87.0 ± 0.00 | 97.0 ± 0.07 |
| Coat | 91.8 ± 0.00 | 91.8 ± 0.34 |
| Sandal | 78.7 ± 0.00 | 99.3 ± 0.02 |
| Shirt | 81.3 ± 0.00 | 84.9 ± 0.22 |
| Sneaker | 97.3 ± 0.00 | 97.8 ± 0.09 |
| Bag | 92.4 ± 0.00 | 99.6 ± 0.02 |
| Ankle boot | 99.1 ± 0.00 | 99.4 ± 0.04 |
| Mean | 89.0 ± 6.36 | 94.7 ± 5.49 |

Table 14: Mean AUC detection performance in % (over 10 seeds) for all individual classes for our transfer learning-based implementations on the ImageNet-30 one vs. rest benchmark with ImageNet-22K OE from Section 5.3.

| Class | Unsupervised CLIP | Supervised OE BCE-CL |
|---|---|---|
| Acorn | 99.78 ± 0.00 | 99.96 ± 0.01 |
| Airliner | 100.00 ± 0.00 | 100.00 ± 0.00 |
| Ambulance | 100.00 ± 0.00 | 100.00 ± 0.00 |
| American alligator | 99.98 ± 0.00 | 100.00 ± 0.00 |
| Banjo | 100.00 ± 0.00 | 100.00 ± 0.00 |
| Barn | 100.00 ± 0.00 | 100.00 ± 0.00 |
| Bikini | 99.87 ± 0.00 | 100.00 ± 0.00 |
| Digital clock | 99.69 ± 0.00 | 99.93 ± 0.02 |
| Dragonfly | 100.00 ± 0.00 | 100.00 ± 0.00 |
| Dumbbell | 99.91 ± 0.00 | 99.97 ± 0.02 |
| Forklift | 100.00 ± 0.00 | 100.00 ± 0.00 |
| Goblet | 99.29 ± 0.00 | 99.81 ± 0.04 |
| Grand piano | 100.00 ± 0.00 | 98.36 ± 4.87 |
| Hotdog | 99.99 ± 0.00 | 100.00 ± 0.00 |
| Hourglass | 99.69 ± 0.00 | 99.97 ± 0.02 |
| Manhole cover | 100.00 ± 0.00 | 100.00 ± 0.00 |
| Mosque | 100.00 ± 0.00 | 100.00 ± 0.00 |
| Nail | 99.61 ± 0.00 | 99.97 ± 0.01 |
| Parking meter | 99.52 ± 0.00 | 99.97 ± 0.01 |
| Pillow | 99.95 ± 0.00 | 100.00 ± 0.00 |
| Revolver | 100.00 ± 0.00 | 100.00 ± 0.00 |
| Rotary dial telephone | 99.84 ± 0.00 | 99.21 ± 1.75 |
| Schooner | 100.00 ± 0.00 | 100.00 ± 0.00 |
| Snowmobile | 99.99 ± 0.00 | 100.00 ± 0.00 |
| Soccer ball | 99.97 ± 0.00 | 99.98 ± 0.05 |
| Stingray | 100.00 ± 0.00 | 100.00 ± 0.00 |
| Strawberry | 99.81 ± 0.00 | 99.97 ± 0.07 |
| Tank | 100.00 ± 0.00 | 100.00 ± 0.00 |
| Toaster | 99.44 ± 0.00 | 99.88 ± 0.01 |
| Volcano | 100.00 ± 0.00 | 99.99 ± 0.04 |
| Mean AUC | 99.88 ± 0.19 | 99.90 ± 0.32 |

Table 15: Mean AUC detection performance in % (over 10 seeds) for all individual classes for our transfer learning-based implementations on the MNIST one vs. rest benchmark with EMNIST OE from Section 5.1.

| Class | Unsupervised CLIP | Supervised OE BCE-CL |
|---|---|---|
| Zero | 73.8 ± 0.00 | 98.2 ± 2.22 |
| One | 38.3 ± 0.00 | 99.6 ± 0.08 |
| Two | 69.8 ± 0.00 | 89.6 ± 11.61 |
| Three | 85.2 ± 0.00 | 99.4 ± 0.06 |
| Four | 64.8 ± 0.00 | 98.4 ± 0.15 |
| Five | 49.4 ± 0.00 | 96.6 ± 0.33 |
| Six | 24.7 ± 0.00 | 88.5 ± 21.08 |
| Seven | 77.9 ± 0.00 | 97.8 ± 0.06 |
| Eight | 40.0 ± 0.00 | 95.6 ± 0.37 |
| Nine | 66.1 ± 0.00 | 96.9 ± 0.43 |
| Mean | 59.0 ± 18.76 | 96.0 ± 3.70 |

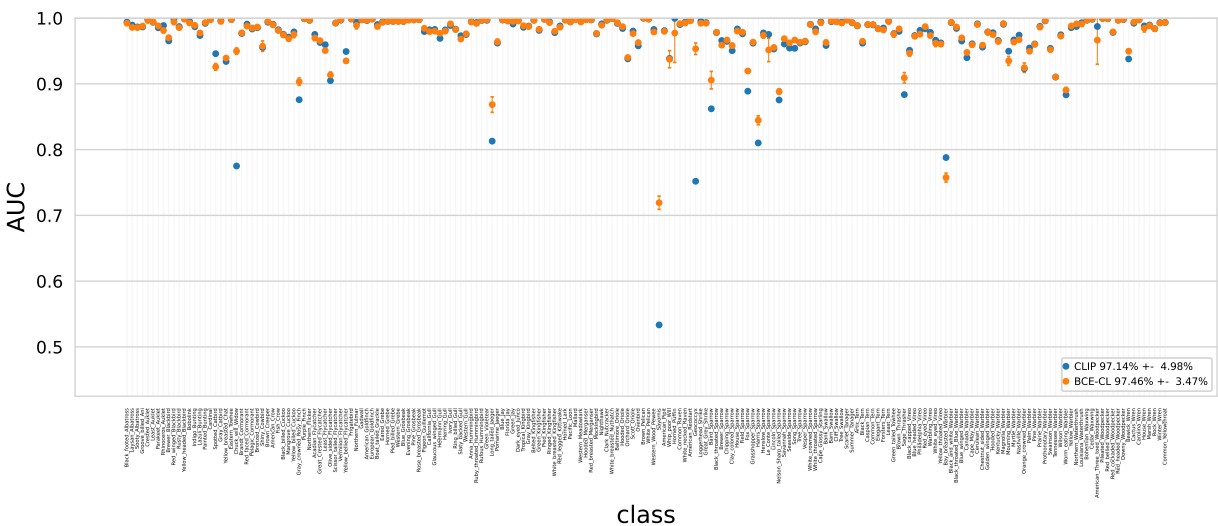

Figure 17: Mean AUC detection performance in % (over 10 seeds) for all individual classes for our transfer learning-based implementations on the CUB one vs. rest benchmark with ImageNet-22k OE from Section 5.3.

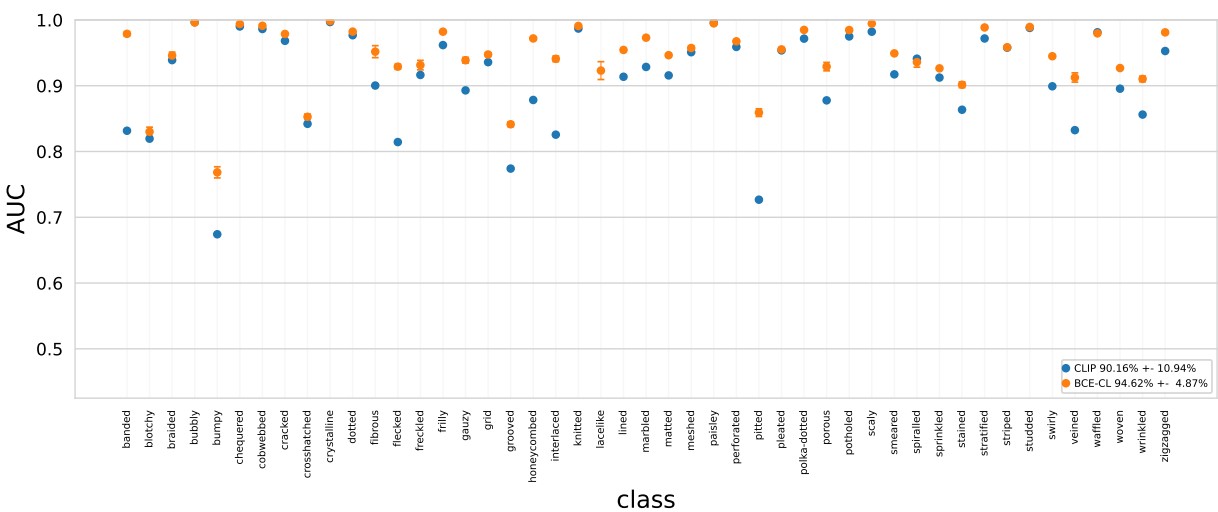

Figure 18: Mean AUC detection performance in % (over 10 seeds) for all individual classes for our transfer learning-based implementations on the DTD one vs. rest benchmark with ImageNet-22k OE from Section 5.3.

### I.3 Image AD on the leave-one-class-out benchmark

In the main paper, we evaluate our methods on the ubiquitous one vs. rest image-AD benchmark (see Section 4). In Section 5.2 we also present results on the more uncommon leave-one-class-out benchmark when varying the OE training set size. Here, we present results when using the full OE training set size, analogue to the experiments in Sections 5.1 and 5.3. Tables 17 and 16 show results for leave-one-class-out CIFAR-10 and leave-one-class-out ImageNet-30, respectively.

Note that, to apply CLIP to the leave-one-class-out setting with $K$ normal classes, we use the text tuple $(v_1, \ldots, v_{K+1}) = $ ("a photo of a {NORMAL_CLASS_1}", ..., "a photo of a {NORMAL_CLASS_K}", "a photo of something"). For a test image $\boldsymbol{x}$, we compute its anomaly score as

$$s(\boldsymbol{x}) = \frac{\exp(\langle f_u(\boldsymbol{x}), f_v(v_{K+1})\rangle \cdot 100)}{\sum_{k=1}^{K+1} \exp(\langle f_u(\boldsymbol{x}), f_v(v_k)\rangle \cdot 100)}. \tag{5}$$

Table 16: Mean AUC detection performance in % (over 2 seeds) with standard deviations for all individual classes for our implementations on the ImageNet-30 leave-one-class-out AD benchmark with ImageNet-22k (with ImageNet-1k removed) OE.

| Anomaly class | Unsupervised | | Unsupervised OE | | Supervised OE | | |
|---|---|---|---|---|---|---|---|
| | DSVDD | CLIP | DSAD | HSC | Focal | BCE | BCE-CL |
| Acorn | 49.7 ± 1.02 | 98.8 ± 0.00 | 92.0 ± 0.21 | 92.6 ± 0.54 | 92.2 ± 0.51 | 90.4 ± 0.02 | 99.7 ± 0.05 |
| Airliner | 47.6 ± 3.67 | 99.9 ± 0.00 | 93.8 ± 0.89 | 92.9 ± 1.71 | 94.7 ± 0.22 | 93.9 ± 0.09 | 99.9 ± 0.01 |
| Ambulance | 50.5 ± 1.57 | 99.7 ± 0.00 | 93.4 ± 0.97 | 92.3 ± 0.92 | 93.1 ± 1.46 | 93.2 ± 1.66 | 99.9 ± 0.02 |
| American alligator | 53.4 ± 4.66 | 98.6 ± 0.00 | 91.7 ± 0.89 | 91.3 ± 0.12 | 91.5 ± 1.01 | 92.1 ± 0.01 | 99.7 ± 0.07 |
| Banjo | 50.3 ± 1.30 | 99.4 ± 0.00 | 85.3 ± 0.98 | 85.6 ± 0.97 | 85.3 ± 0.54 | 86.0 ± 0.04 | 99.9 ± 0.02 |
| Barn | 51.5 ± 0.96 | 98.3 ± 0.00 | 92.0 ± 0.73 | 90.6 ± 0.95 | 91.3 ± 0.17 | 89.3 ± 2.04 | 99.8 ± 0.05 |
| Bikini | 50.6 ± 1.03 | 97.9 ± 0.00 | 89.8 ± 0.04 | 90.0 ± 0.49 | 87.4 ± 0.72 | 90.6 ± 0.44 | 99.7 ± 0.03 |
| Digital clock | 49.1 ± 1.33 | 94.9 ± 0.00 | 86.0 ± 0.42 | 84.8 ± 1.80 | 84.7 ± 0.79 | 85.1 ± 0.83 | 98.5 ± 0.01 |
| Dragonfly | 51.2 ± 3.23 | 99.8 ± 0.00 | 94.2 ± 0.01 | 93.9 ± 0.24 | 93.8 ± 0.58 | 93.5 ± 0.72 | 99.9 ± 0.01 |
| Dumbbell | 46.1 ± 2.87 | 97.7 ± 0.00 | 80.7 ± 1.12 | 82.7 ± 0.78 | 81.7 ± 1.39 | 83.3 ± 0.71 | 99.4 ± 0.07 |
| Forklift | 50.6 ± 1.20 | 96.8 ± 0.00 | 82.1 ± 0.37 | 82.0 ± 0.42 | 82.9 ± 0.34 | 81.9 ± 0.71 | 99.6 ± 0.12 |
| Goblet | 46.8 ± 0.13 | 96.3 ± 0.00 | 84.2 ± 2.64 | 83.9 ± 0.15 | 84.4 ± 1.30 | 83.5 ± 0.99 | 98.8 ± 0.14 |
| Grand piano | 46.2 ± 2.79 | 99.5 ± 0.00 | 82.2 ± 1.00 | 82.4 ± 0.01 | 83.4 ± 0.71 | 82.6 ± 1.08 | 99.9 ± 0.02 |
| Hotdog | 50.4 ± 1.74 | 99.7 ± 0.00 | 93.2 ± 0.62 | 93.9 ± 0.49 | 94.3 ± 0.27 | 94.2 ± 0.23 | 99.7 ± 0.03 |
| Hourglass | 50.9 ± 2.30 | 87.3 ± 0.00 | 87.1 ± 0.89 | 85.9 ± 0.62 | 86.9 ± 0.71 | 85.5 ± 1.14 | 96.0 ± 0.29 |
| Manhole cover | 51.8 ± 5.59 | 93.0 ± 0.00 | 84.7 ± 1.74 | 84.9 ± 1.47 | 87.8 ± 0.48 | 88.3 ± 0.09 | 99.6 ± 0.04 |
| Mosque | 49.4 ± 6.93 | 99.7 ± 0.00 | 91.7 ± 0.04 | 89.4 ± 1.22 | 88.1 ± 0.84 | 89.1 ± 0.29 | 99.9 ± 0.06 |
| Nail | 49.3 ± 0.87 | 97.1 ± 0.00 | 86.2 ± 0.44 | 87.7 ± 0.74 | 86.6 ± 0.08 | 85.4 ± 1.49 | 97.0 ± 0.05 |
| Parking meter | 51.1 ± 2.91 | 92.4 ± 0.00 | 85.1 ± 0.96 | 82.1 ± 0.26 | 81.9 ± 0.87 | 81.3 ± 0.17 | 96.3 ± 0.55 |
| Pillow | 47.9 ± 0.37 | 99.5 ± 0.00 | 91.6 ± 0.65 | 90.6 ± 0.84 | 91.0 ± 0.65 | 90.6 ± 0.36 | 99.9 ± 0.02 |
| Revolver | 54.7 ± 0.47 | 99.6 ± 0.00 | 91.3 ± 0.66 | 88.9 ± 0.91 | 89.4 ± 0.15 | 90.5 ± 0.65 | 99.8 ± 0.04 |
| Rotary dial telephone | 50.0 ± 0.68 | 97.3 ± 0.00 | 85.2 ± 1.38 | 84.9 ± 1.09 | 82.9 ± 0.38 | 82.7 ± 0.41 | 99.3 ± 0.10 |
| Schooner | 42.0 ± 0.25 | 99.9 ± 0.00 | 95.6 ± 0.12 | 94.2 ± 0.65 | 95.6 ± 0.18 | 94.4 ± 0.61 | 99.9 ± 0.01 |
| Snowmobile | 48.2 ± 2.40 | 99.5 ± 0.00 | 86.6 ± 0.01 | 88.0 ± 0.12 | 87.5 ± 1.40 | 86.5 ± 0.16 | 99.9 ± 0.01 |
| Soccer ball | 51.1 ± 0.20 | 99.4 ± 0.00 | 90.3 ± 0.46 | 89.8 ± 0.67 | 90.5 ± 0.51 | 90.2 ± 0.92 | 99.8 ± 0.00 |
| Stingray | 48.6 ± 6.35 | 97.1 ± 0.00 | 91.8 ± 0.17 | 90.9 ± 1.21 | 90.8 ± 0.53 | 90.4 ± 0.39 | 99.8 ± 0.01 |
| Strawberry | 52.0 ± 2.28 | 99.5 ± 0.00 | 95.5 ± 0.95 | 96.4 ± 0.34 | 95.5 ± 0.23 | 95.0 ± 0.48 | 99.8 ± 0.01 |
| Tank | 51.8 ± 2.40 | 98.4 ± 0.00 | 86.8 ± 1.76 | 82.7 ± 0.72 | 84.9 ± 0.19 | 84.6 ± 1.32 | 99.8 ± 0.01 |
| Toaster | 48.3 ± 1.40 | 97.2 ± 0.00 | 84.0 ± 0.08 | 83.3 ± 1.03 | 86.1 ± 0.72 | 84.3 ± 0.32 | 98.0 ± 0.19 |
| Volcano | 51.3 ± 2.07 | 99.6 ± 0.00 | 89.2 ± 1.10 | 89.2 ± 0.77 | 88.5 ± 0.65 | 88.4 ± 0.94 | 99.9 ± 0.01 |
| Mean | 49.7 ± 2.41 | 97.8 ± 2.73 | 88.8 ± 4.20 | 88.3 ± 4.16 | 88.5 ± 4.14 | 88.2 ± 4.12 | 99.3 ± 1.05 |

Table 17: Mean AUC detection performance in % (over 2 seeds) with standard deviations for all individual classes for our implementations on the CIFAR-10 leave-one-class-out AD benchmark with 80MTI OE.

| | Unsupervised | | Unsupervised OE | | Supervised OE | | |
|---|---|---|---|---|---|---|---|
| Class | DSVDD | CLIP | DSAD | HSC | Focal | BCE | BCE-CL |
| Airplane | $62.3 \pm 8.90$ | $96.2 \pm 0.00$ | $85.5 \pm 0.52$ | $87.0 \pm 0.52$ | $88.4 \pm 0.28$ | $87.9 \pm 0.17$ | $99.2 \pm 0.03$ |
| Automobile | $52.7 \pm 0.69$ | $96.0 \pm 0.00$ | $85.6 \pm 0.81$ | $87.2 \pm 0.56$ | $89.9 \pm 0.71$ | $90.3 \pm 0.01$ | $99.3 \pm 0.02$ |
| Bird | $54.2 \pm 4.00$ | $93.5 \pm 0.00$ | $83.6 \pm 0.04$ | $83.8 \pm 0.03$ | $86.2 \pm 0.49$ | $86.0 \pm 0.45$ | $97.9 \pm 0.15$ |
| Cat | $39.6 \pm 2.98$ | $90.5 \pm 0.00$ | $82.4 \pm 0.10$ | $82.4 \pm 0.80$ | $83.4 \pm 0.52$ | $84.1 \pm 0.36$ | $97.9 \pm 0.01$ |
| Deer | $56.6 \pm 0.49$ | $79.5 \pm 0.00$ | $75.6 \pm 0.11$ | $74.9 \pm 0.06$ | $76.9 \pm 0.42$ | $77.0 \pm 0.13$ | $96.6 \pm 0.03$ |
| Dog | $49.9 \pm 4.06$ | $90.6 \pm 0.00$ | $82.7 \pm 0.15$ | $82.8 \pm 0.09$ | $84.1 \pm 0.75$ | $84.6 \pm 0.03$ | $97.9 \pm 0.00$ |
| Frog | $53.2 \pm 11.75$ | $94.0 \pm 0.00$ | $85.1 \pm 0.10$ | $85.5 \pm 0.35$ | $86.1 \pm 0.28$ | $86.4 \pm 0.31$ | $98.3 \pm 0.12$ |
| Horse | $47.7 \pm 0.69$ | $93.4 \pm 0.00$ | $85.2 \pm 0.57$ | $85.9 \pm 0.08$ | $88.4 \pm 0.04$ | $88.2 \pm 0.52$ | $98.6 \pm 0.09$ |
| Ship | $54.7 \pm 5.87$ | $97.8 \pm 0.00$ | $89.7 \pm 0.12$ | $90.7 \pm 0.06$ | $91.3 \pm 0.17$ | $92.0 \pm 0.35$ | $99.4 \pm 0.03$ |
| Truck | $51.1 \pm 1.86$ | $90.0 \pm 0.00$ | $86.7 \pm 0.23$ | $88.0 \pm 0.20$ | $88.9 \pm 0.41$ | $89.2 \pm 0.37$ | $99.0 \pm 0.01$ |
| Mean | $52.2 \pm 5.66$ | $92.2 \pm 4.89$ | $84.2 \pm 3.51$ | $84.8 \pm 4.08$ | $86.4 \pm 3.94$ | $86.6 \pm 3.96$ | $98.4 \pm 0.83$ |

## I.4 Image AD on the MVTec-AD benchmark

As mentioned in Section 2, our paper focuses on natural images because random images from the web are likely not informative as OE for other data. In Section 5.2 we demonstrate this on the manufacturing dataset MVTec-AD, where anomalies are rather subtle, for varying OE set sizes. Here we show results using the full ImageNet-22k dataset as OE in Table 18.

Table 18: Mean AUC detection performance in % (over 2 seeds) with standard deviations for all individual classes for our implementations on the MVTec-AD benchmark with ImageNet22k OE.

| | Unsupervised | | Unsupervised OE | | Supervised OE | | |
|---|---|---|---|---|---|---|---|
| Class | DSVDD | CLIP | DSAD | HSC | Focal | BCE | BCE-CL |
| Bottle | $83.0 \pm 2.38$ | $34.8 \pm 0.00$ | $78.0 \pm 6.07$ | $73.6 \pm 0.95$ | $66.9 \pm 6.47$ | $71.9 \pm 5.28$ | $76.0 \pm 4.92$ |
| Cable | $60.7 \pm 3.02$ | $48.7 \pm 0.00$ | $71.3 \pm 2.74$ | $69.1 \pm 0.42$ | $61.5 \pm 3.47$ | $54.8 \pm 4.76$ | $72.1 \pm 4.82$ |
| Capsule | $61.4 \pm 4.09$ | $43.3 \pm 0.00$ | $53.2 \pm 7.64$ | $59.7 \pm 5.88$ | $56.0 \pm 11.81$ | $57.5 \pm 3.97$ | $59.2 \pm 1.07$ |
| Carpet | $54.0 \pm 1.89$ | $72.5 \pm 0.00$ | $55.6 \pm 2.96$ | $63.5 \pm 5.12$ | $55.3 \pm 0.22$ | $57.1 \pm 0.52$ | $88.1 \pm 0.63$ |
| Grid | $47.0 \pm 2.09$ | $60.6 \pm 0.00$ | $48.9 \pm 16.75$ | $59.4 \pm 1.46$ | $46.0 \pm 4.93$ | $49.9 \pm 3.63$ | $72.7 \pm 0.92$ |
| Hazelnut | $80.5 \pm 0.11$ | $48.1 \pm 0.00$ | $70.2 \pm 6.56$ | $79.2 \pm 3.38$ | $73.2 \pm 5.14$ | $78.8 \pm 2.82$ | $62.8 \pm 1.12$ |
| Leather | $69.3 \pm 3.21$ | $99.9 \pm 0.00$ | $80.1 \pm 3.87$ | $84.6 \pm 0.48$ | $87.3 \pm 4.52$ | $87.3 \pm 1.61$ | $87.4 \pm 12.50$ |
| Metal nut | $55.7 \pm 0.20$ | $45.0 \pm 0.00$ | $47.9 \pm 5.33$ | $62.7 \pm 5.76$ | $61.1 \pm 7.14$ | $54.5 \pm 4.42$ | $75.3 \pm 7.61$ |
| Pill | $61.3 \pm 2.17$ | $69.4 \pm 0.00$ | $58.8 \pm 10.27$ | $55.8 \pm 4.95$ | $70.3 \pm 2.96$ | $53.8 \pm 11.25$ | $64.5 \pm 1.20$ |
| Screw | $49.5 \pm 1.52$ | $64.4 \pm 0.00$ | $39.0 \pm 1.30$ | $41.9 \pm 1.84$ | $59.5 \pm 4.08$ | $62.0 \pm 7.20$ | $76.6 \pm 1.33$ |
| Tile | $63.7 \pm 0.05$ | $78.0 \pm 0.00$ | $95.4 \pm 1.06$ | $95.3 \pm 1.15$ | $94.3 \pm 2.09$ | $95.5 \pm 1.82$ | $98.2 \pm 0.20$ |
| Toothbrush | $52.9 \pm 6.53$ | $58.3 \pm 0.00$ | $70.3 \pm 1.94$ | $88.3 \pm 2.99$ | $59.6 \pm 1.81$ | $66.9 \pm 4.72$ | $68.0 \pm 6.76$ |
| Transistor | $61.9 \pm 6.73$ | $51.0 \pm 0.00$ | $70.4 \pm 7.19$ | $64.9 \pm 2.83$ | $64.6 \pm 3.87$ | $66.9 \pm 0.08$ | $72.5 \pm 6.65$ |
| Wood | $87.7 \pm 0.70$ | $35.5 \pm 0.00$ | $85.1 \pm 11.27$ | $93.8 \pm 1.67$ | $73.6 \pm 9.12$ | $75.7 \pm 20.57$ | $94.3 \pm 0.73$ |
| Zipper | $70.8 \pm 1.52$ | $36.5 \pm 0.00$ | $67.1 \pm 6.22$ | $60.3 \pm 10.50$ | $51.8 \pm 25.64$ | $58.6 \pm 0.54$ | $74.5 \pm 0.67$ |
| Mean | $64.0 \pm 11.81$ | $56.4 \pm 17.57$ | $66.1 \pm 14.89$ | $70.1 \pm 14.78$ | $65.4 \pm 12.44$ | $66.1 \pm 12.93$ | $76.2 \pm 10.96$ |

## I.5 Varying the OE size

For the experiments on varying the number of OE samples (Section 5.2), we include plots for all individual classes in Figure 21 for CIFAR-10 and in Figures 19 and 20 for ImageNet-30, respectively. Additionally, for the experiments on varying the diversity of OE data on CIFAR-10 with CIFAR-100 OE, we added the plots for all individual classes in Figure 22.

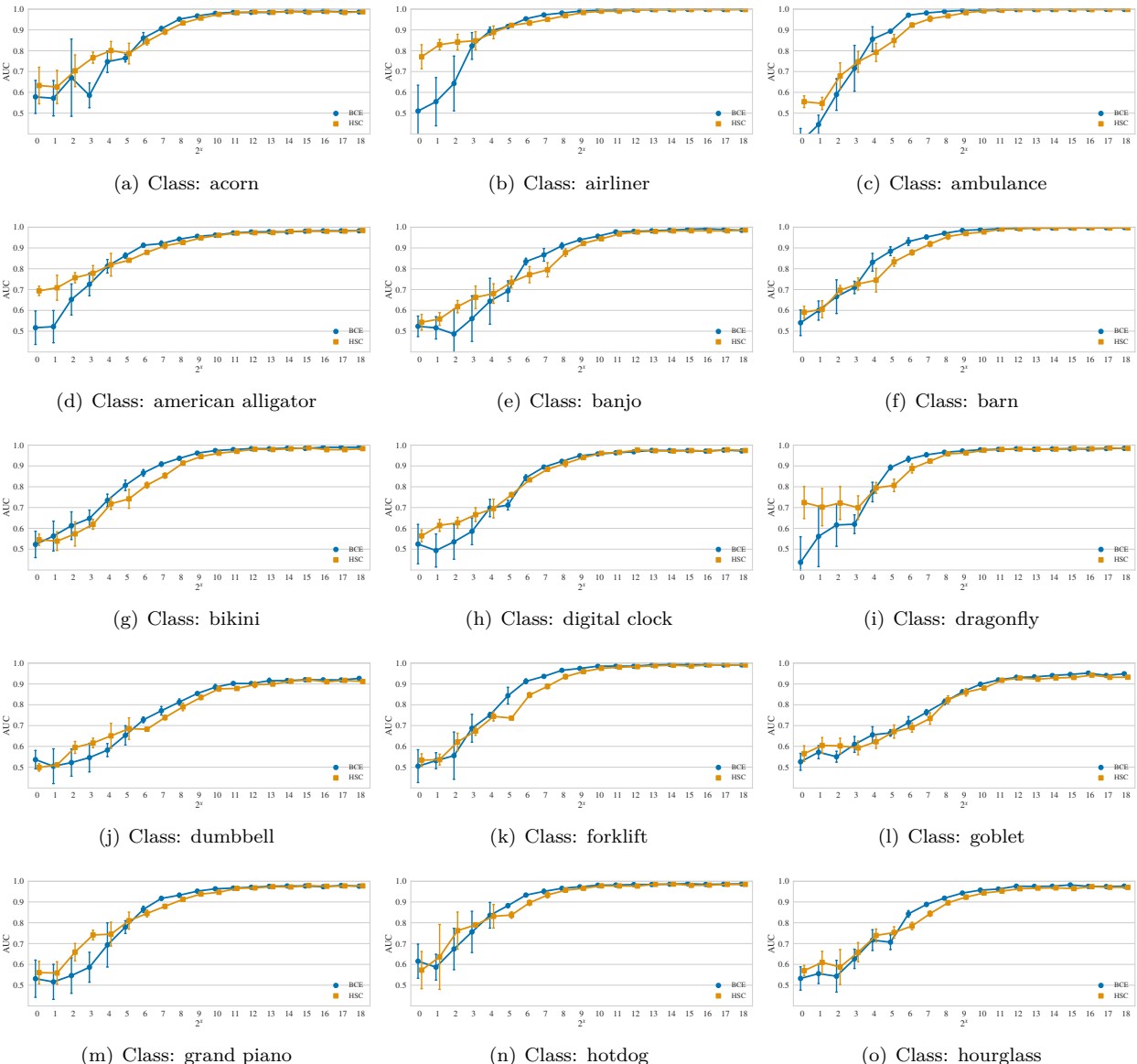

Figure 19: Mean AUC detection performance in % (over 5 seeds) for all classes of the ImageNet-30 one vs. rest benchmark from Section 5.2 when varying the number of ImageNet-22K OE samples. These plots correspond to Figure 2(c), but here we report the results for all individual classes (from class 1 (acorn) to class 15 (hourglass)).

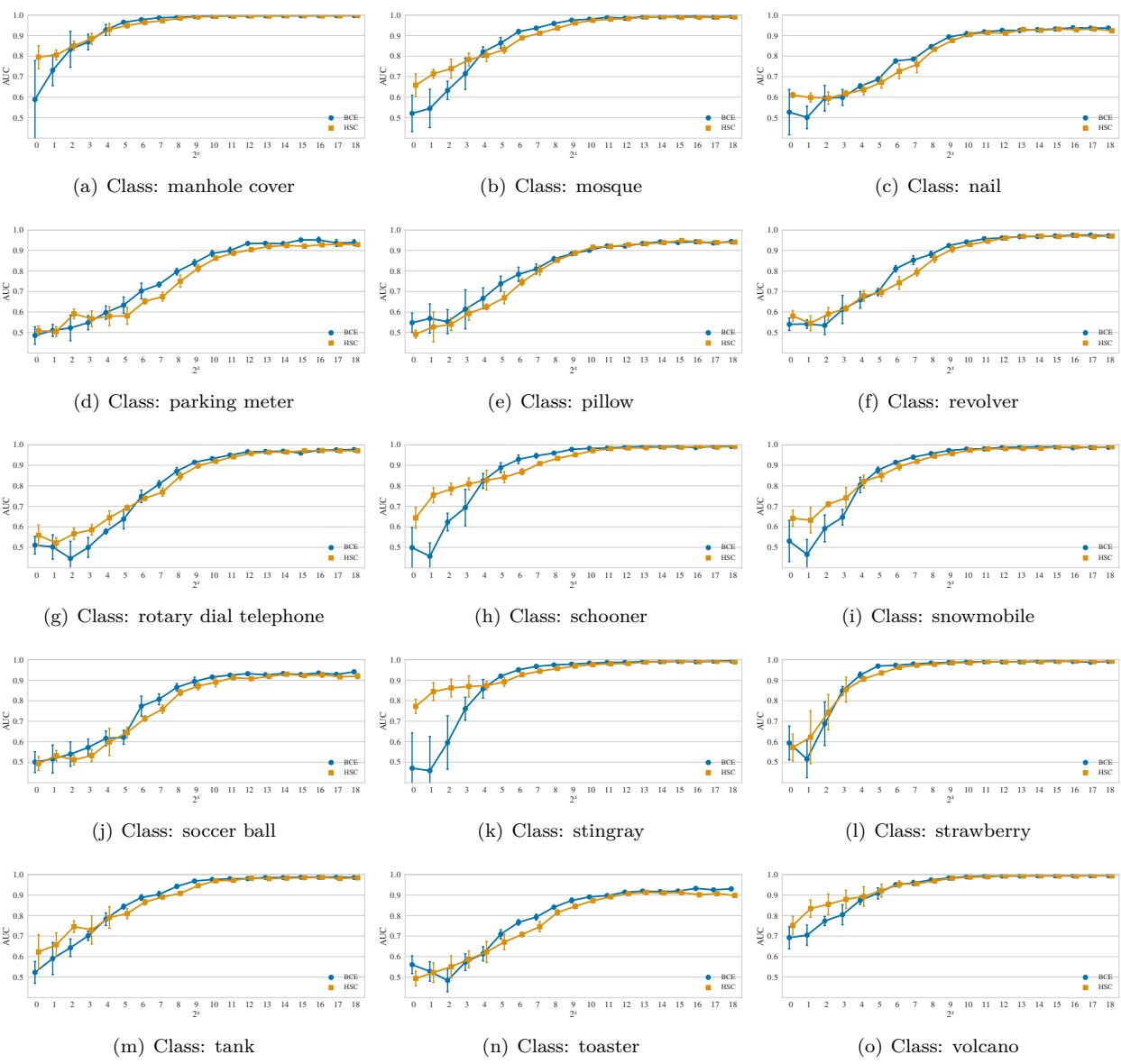

Figure 20: Mean AUC detection performance in % (over 5 seeds) for all classes of the ImageNet-30 one vs. rest benchmark from Section 5.2 when varying the number of ImageNet-22K OE samples. These plots correspond to Figure 2(c), but here we report the results for all individual classes (from class 16 (manhole cover) to class 30 (volcano)).

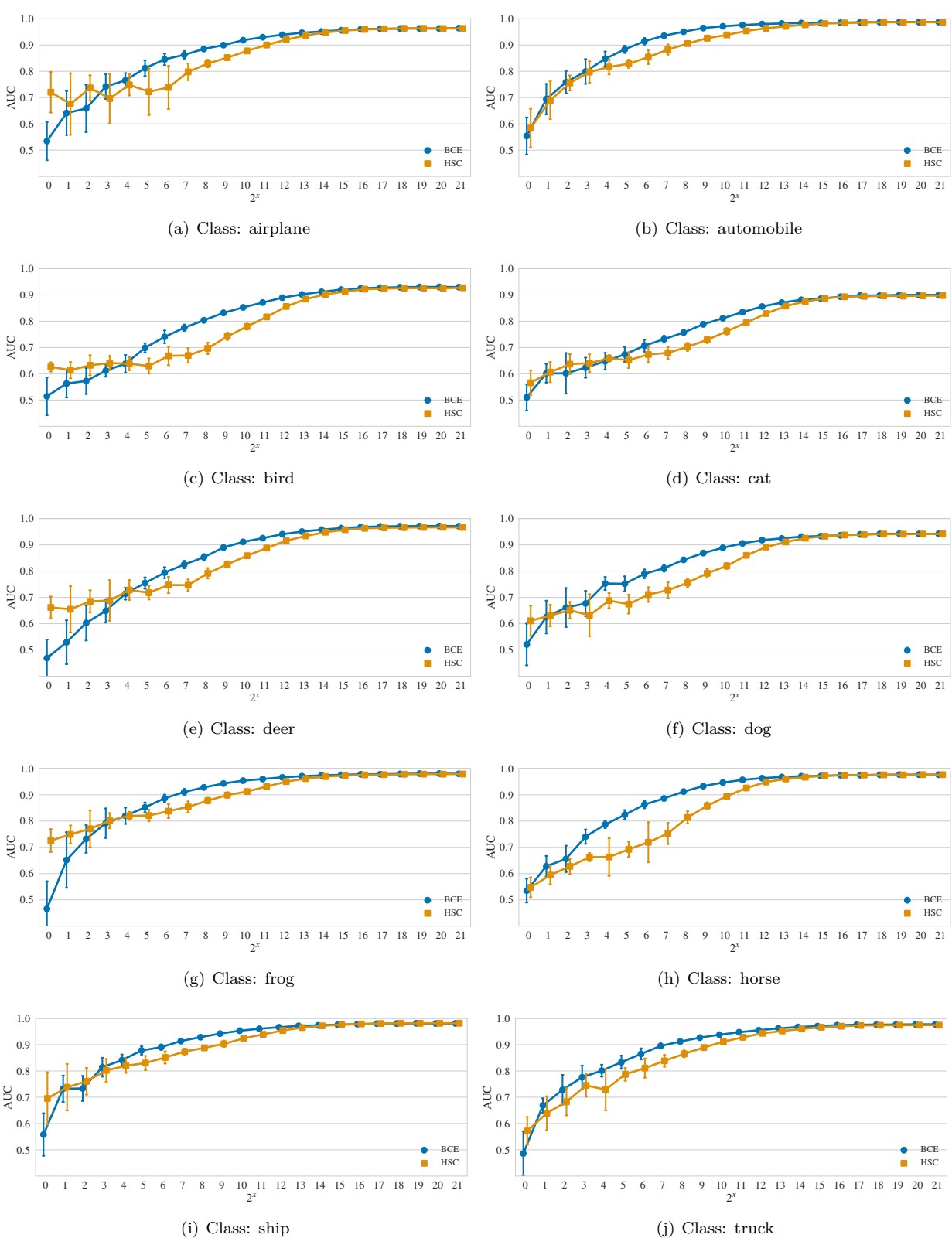

Figure 21: Mean AUC detection performance in % (over 10 seeds) for all classes of the CIFAR-10 one vs. rest benchmark from Section 5.2 when varying the number of 80MTI OE samples. These plots correspond to Figure 2(a), but here we report the results for all individual classes.

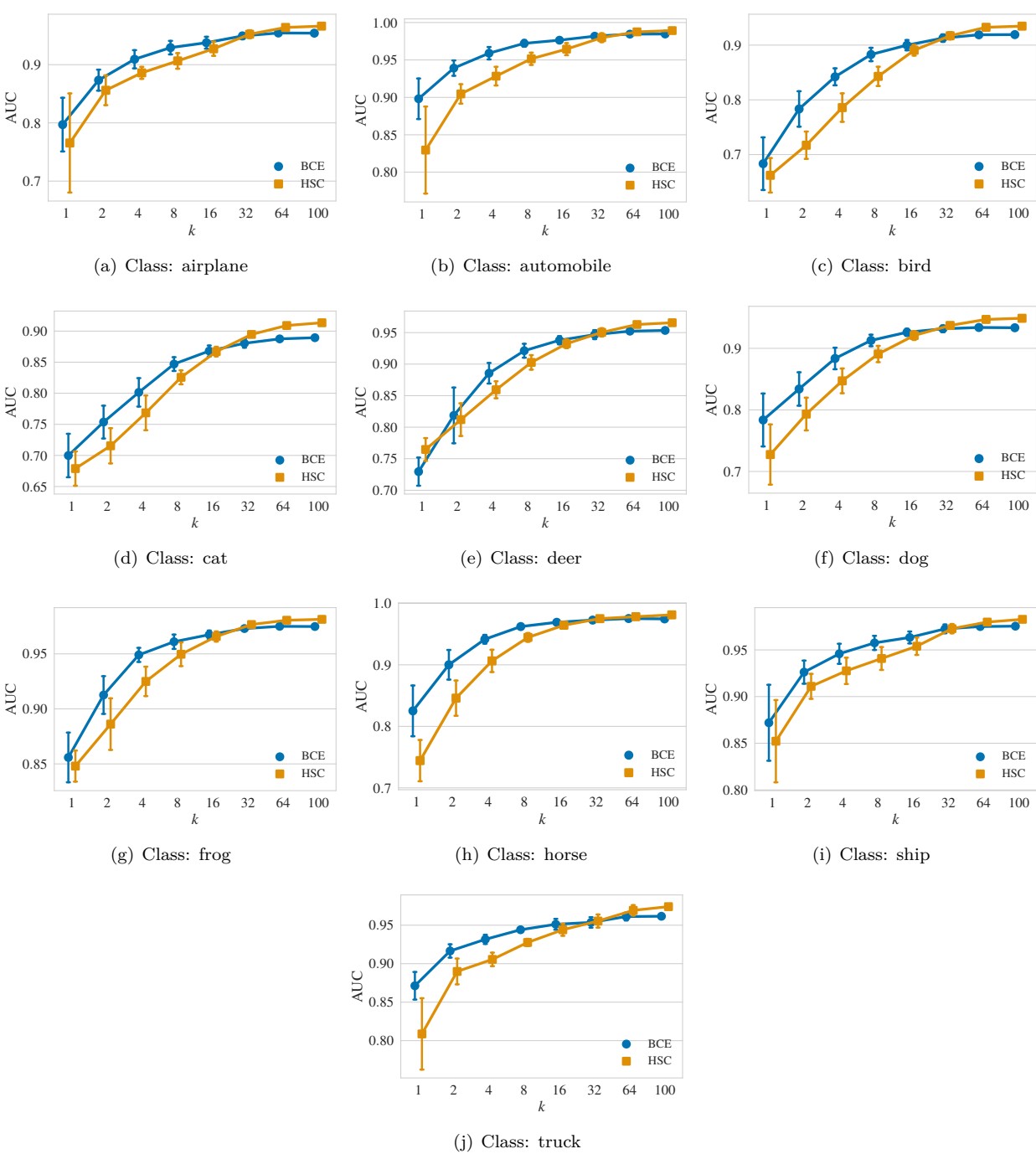

Figure 22: Mean AUC detection performance in % (over 10 seeds) for all CIFAR-10 classes from the experiment in Appendix A on varying the number of classes $k$ of the CIFAR-100 OE data. These plots correspond to Figure 7, but here we report the results for all individual classes.

### I.6 Robustness towards choice of OE samples

Here we provide class-wise results for the best and worst single OE samples found via an evolutionary algorithm (Section 5.4). On ImageNet, we only consider the first 10 classes of the ones used by (Hendrycks et al., 2019b): "Acorn", "Airliner", "Ambulance", "American alligator", "Banjo", "Barn", "Bikini", "Digital clock", "Dragonfly", "Dumbbell". However, the full 30-class one vs. rest benchmark was used to evaluate for each of these ten classes. Tables 19 and 20 show the performance of the best or worst single OE sample (from 80MTI) for each class in CIFAR-10, while the former shows results for unfiltered images and the latter for low-pass-filtered or high-pass-filtered images. Tables 21 and 22 show the same for ImageNet-10 with ImageNet-22K as OE. Figures 23, 24, 25, 26, 27, and 28 show the 4-5 best and worst samples for HSC and BCE, on CIFAR-10 or ImageNet-10, for unfiltered or filtered images, respectively.

Table 19: Class-wise AUC detection performance in % for the best and worst single OE samples found via an evolutionary algorithm (Section 5.4) on the CIFAR-10 one vs. rest AD benchmark using 80MTI as OE.

| | Best OE | | Worst OE | |
|---|---|---|---|---|
| Class | HSC | BCE | HSC | BCE |
| Airplane | 85.2 | 76.1 | 37.4 | 29.9 |
| Automobile | 78.1 | 71.5 | 36.4 | 32.9 |
| Bird | 74.0 | 67.9 | 47.2 | 35.8 |
| Cat | 72.5 | 64.9 | 38.7 | 35.1 |
| Deer | 79.9 | 68.2 | 55.1 | 29.8 |
| Dog | 71.6 | 70.6 | 41.0 | 35.0 |
| Frog | 84.2 | 71.7 | 47.7 | 24.8 |
| Horse | 66.2 | 66.2 | 41.9 | 38.7 |
| Ship | 86.5 | 72.8 | 46.2 | 27.1 |
| Truck | 78.8 | 69.1 | 41.2 | 26.9 |
| Mean AUC | 77.7 | 69.9 | 43.3 | 31.6 |

Table 20: Class-wise AUC detection performance in % for the best and worst single OE samples found via an evolutionary algorithm (Section 5.4) on the CIFAR-10 one vs. rest AD benchmark using 80MTI as OE. All images are either low-pass-filtered (LPF) or high-pass-filtered (HPF), both during training and testing.

| | Best OE | | | | Worst OE | | | |
|---|---|---|---|---|---|---|---|---|
| Class | HSC LPF | BCE LPF | HSC HPF | BCE HPF | HSC LPF | BCE LPF | HSC HPF | BCE HPF |
| Airplane | 83.0 | 68.5 | 68.5 | 64.6 | 36.0 | 22.1 | 40.7 | 32.5 |
| Automobile | 75.0 | 69.5 | 60.1 | 69.0 | 47.5 | 32.6 | 40.2 | 40.3 |
| Bird | 71.1 | 63.5 | 67.9 | 61.6 | 40.3 | 39.1 | 46.4 | 39.2 |
| Cat | 73.6 | 63.2 | 65.5 | 63.3 | 44.0 | 34.2 | 48.6 | 39.4 |
| Deer | 76.9 | 69.2 | 68.5 | 63.8 | 38.0 | 31.7 | 47.2 | 38.9 |
| Dog | 74.2 | 64.3 | 69.4 | 67.1 | 43.0 | 34.1 | 53.3 | 37.3 |
| Frog | 83.5 | 73.1 | 78.2 | 70.3 | 40.1 | 28.2 | 38.0 | 37.0 |
| Horse | 72.5 | 66.5 | 66.7 | 63.0 | 50.9 | 35.5 | 49.0 | 40.3 |
| Ship | 85.7 | 75.8 | 74.8 | 72.2 | 47.8 | 23.0 | 35.5 | 35.8 |
| Truck | 79.9 | 71.0 | 67.9 | 69.2 | 53.0 | 30.1 | 37.3 | 39.7 |
| Mean AUC | 77.5 | 68.5 | 68.8 | 66.4 | 44.1 | 31.1 | 43.6 | 38.0 |

Table 21: Class-wise AUC detection performance in % for the best and worst single OE samples found via an evolutionary algorithm (Section 5.4) on the ImageNet-30 one vs. rest AD benchmark using ImageNet-22k (with the 1K classes removed) as OE.

| | Best OE | | Worst OE | |
|---|---|---|---|---|
| Class | HSC | BCE | HSC | BCE |
| Acorn | 84.0 | 76.6 | 43.6 | 22.3 |
| Airliner | 91.1 | 80.8 | 41.3 | 15.7 |
| Ambulance | 84.0 | 86.4 | 24.0 | 22.9 |
| American alligator | 81.2 | 75.7 | 54.6 | 30.1 |
| Banjo | 81.3 | 75.8 | 37.2 | 26.8 |
| Barn | 78.4 | 68.0 | 40.0 | 29.0 |
| Bikini | 66.7 | 67.0 | 41.5 | 38.9 |
| Digital clock | 67.7 | 73.2 | 32.5 | 28.1 |
| Dragonfly | 86.6 | 84.8 | 36.1 | 13.6 |
| Dumbbell | 71.8 | 66.5 | 40.8 | 35.9 |
| Mean AUC | 79.3 | 75.5 | 39.2 | 26.3 |

Table 22: Class-wise AUC detection performance in % for the best and worst single OE samples found via an evolutionary algorithm (Section 5.4) on the ImageNet-30 one vs. rest AD benchmark using ImageNet-22k (with the 1K classes removed) as OE. All images are either low-pass-filtered (LPF) or high-pass-filtered (HPF), both during training and testing.

| | Best OE | | | | Worst OE | | | |
|---|---|---|---|---|---|---|---|---|
| Class | HSC LPF | BCE LPF | HSC HPF | BCE HPF | HSC LPF | BCE LPF | HSC HPF | BCE HPF |
| Acorn | 82.8 | 83.8 | 80.4 | 75.9 | 41.2 | 20.0 | 45.8 | 31.0 |
| Airliner | 78.2 | 77.3 | 83.6 | 86.3 | 44.6 | 21.7 | 52.5 | 18.0 |
| Ambulance | 86.7 | 76.3 | 79.1 | 83.3 | 53.6 | 26.1 | 44.4 | 29.2 |
| American alligator | 69.3 | 67.4 | 78.3 | 80.1 | 43.4 | 32.3 | 42.2 | 29.0 |
| Banjo | 83.0 | 74.8 | 71.3 | 75.9 | 48.0 | 21.5 | 39.1 | 31.9 |
| Barn | 79.6 | 71.5 | 73.0 | 77.6 | 45.2 | 25.6 | 44.4 | 26.2 |
| Bikini | 70.4 | 68.0 | 68.6 | 65.9 | 47.4 | 35.6 | 45.9 | 34.9 |
| Digital clock | 63.3 | 65.7 | 68.4 | 76.2 | 41.2 | 35.9 | 34.9 | 26.0 |
| Dragonfly | 85.0 | 85.6 | 83.7 | 84.6 | 33.0 | 12.9 | 51.4 | 20.2 |
| Dumbbell | 73.2 | 68.1 | 63.8 | 67.2 | 48.1 | 29.1 | 40.0 | 32.8 |
| Mean AUC | 77.2 | 73.8 | 75.0 | 77.3 | 44.6 | 26.1 | 44.1 | 27.9 |

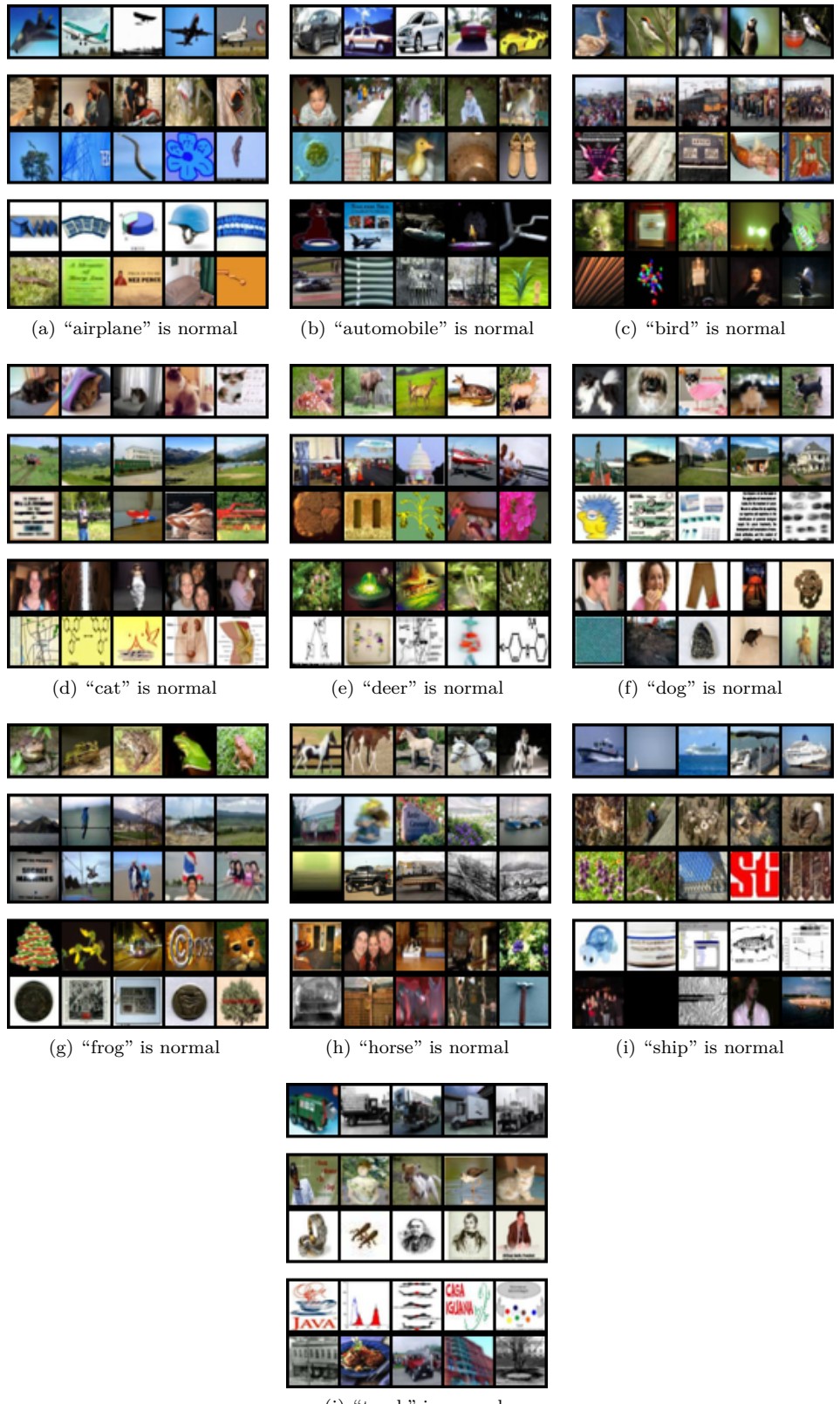

Figure 23: Optimal OE samples for CIFAR-10 with 80MTI as OE. The first row shows normal samples, the next two rows the best samples found via HSC (top) and BCE (bottom), and the last two rows the worst samples found via HSC (top) and BCE (bottom).

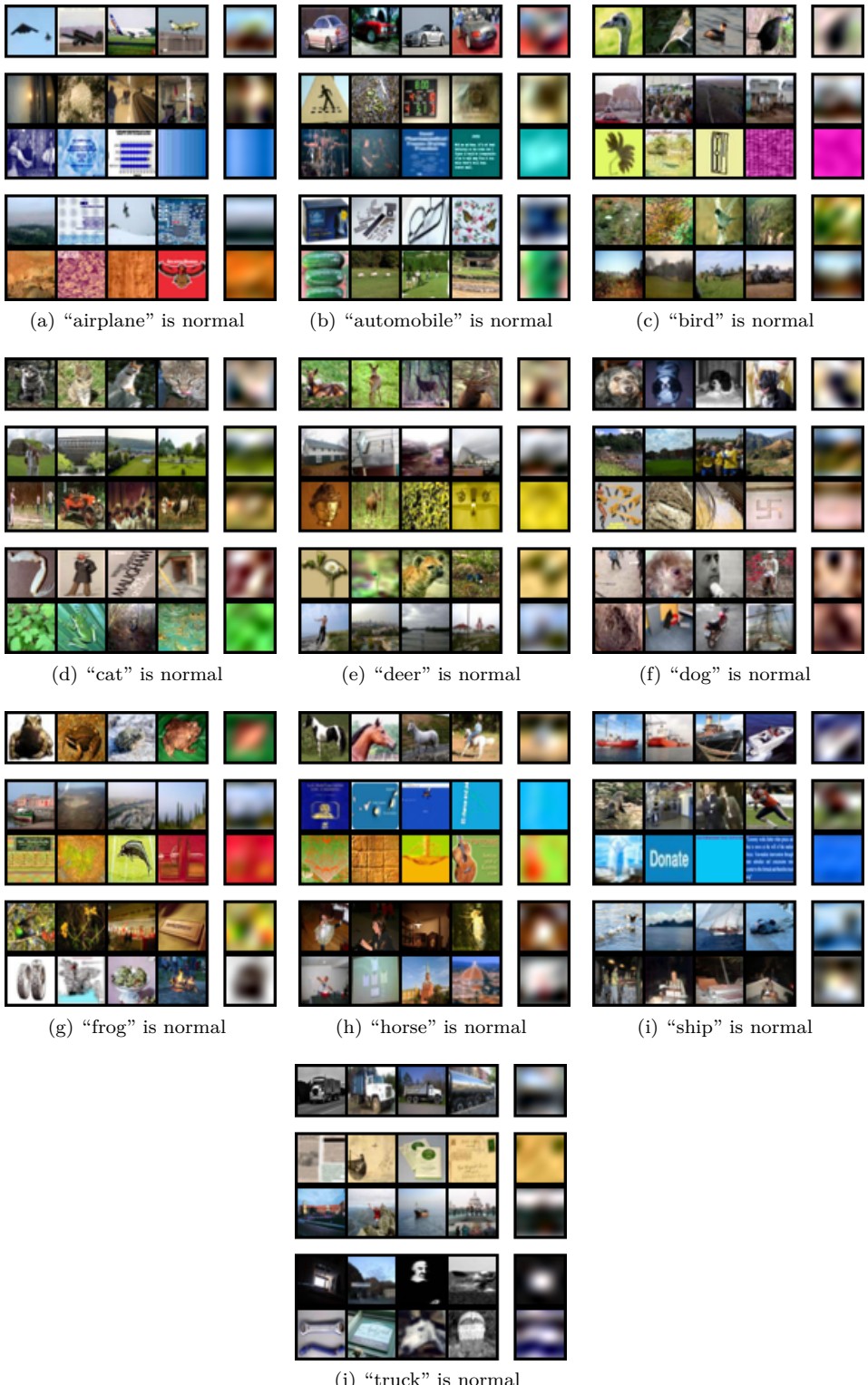

(a) "airplane" is normal    (b) "automobile" is normal    (c) "bird" is normal

(d) "cat" is normal    (e) "deer" is normal    (f) "dog" is normal

(g) "frog" is normal    (h) "horse" is normal    (i) "ship" is normal

(j) "truck" is normal

Figure 24: Optimal OE samples for low-pass-filtered CIFAR-10 with low-pass-filtered 80MTI as OE. The first row shows normal samples, the next two rows the best samples found via HSC (top) and BCE (bottom), and the last two rows the worst samples found via HSC (top) and BCE (bottom). The last column shows the low-pass-filtered version of the images, which is what the network sees during training and testing.

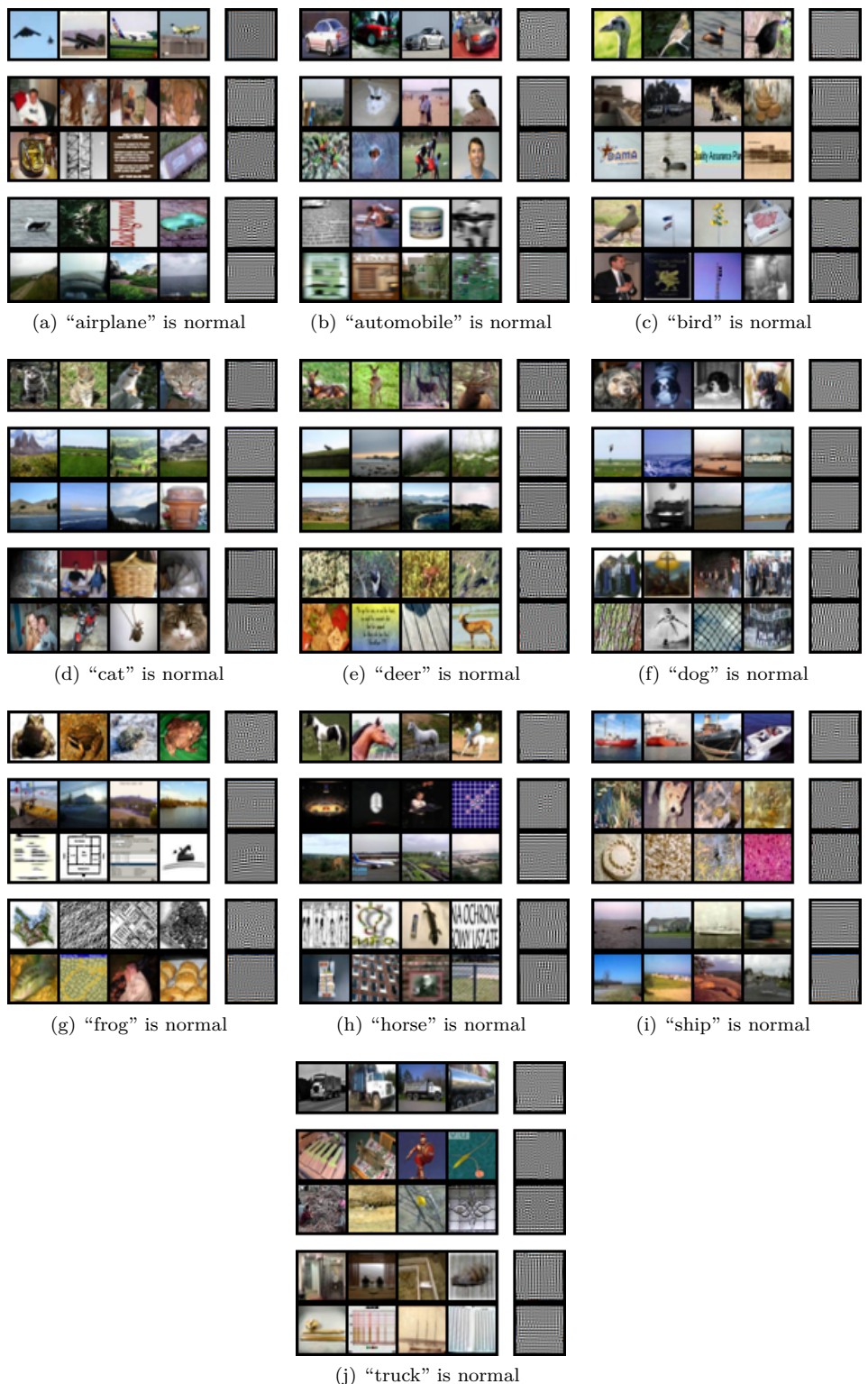

Figure 25: Optimal OE samples for high-pass-filtered CIFAR-10 with high-pass-filtered 80MTI as OE. The first row shows normal samples, the next two rows the best samples found via HSC (top) and BCE (bottom), and the last two rows the worst samples found via HSC (top) and BCE (bottom). The last column shows the high-pass-filtered version of the images, which is what the network sees during training and testing.

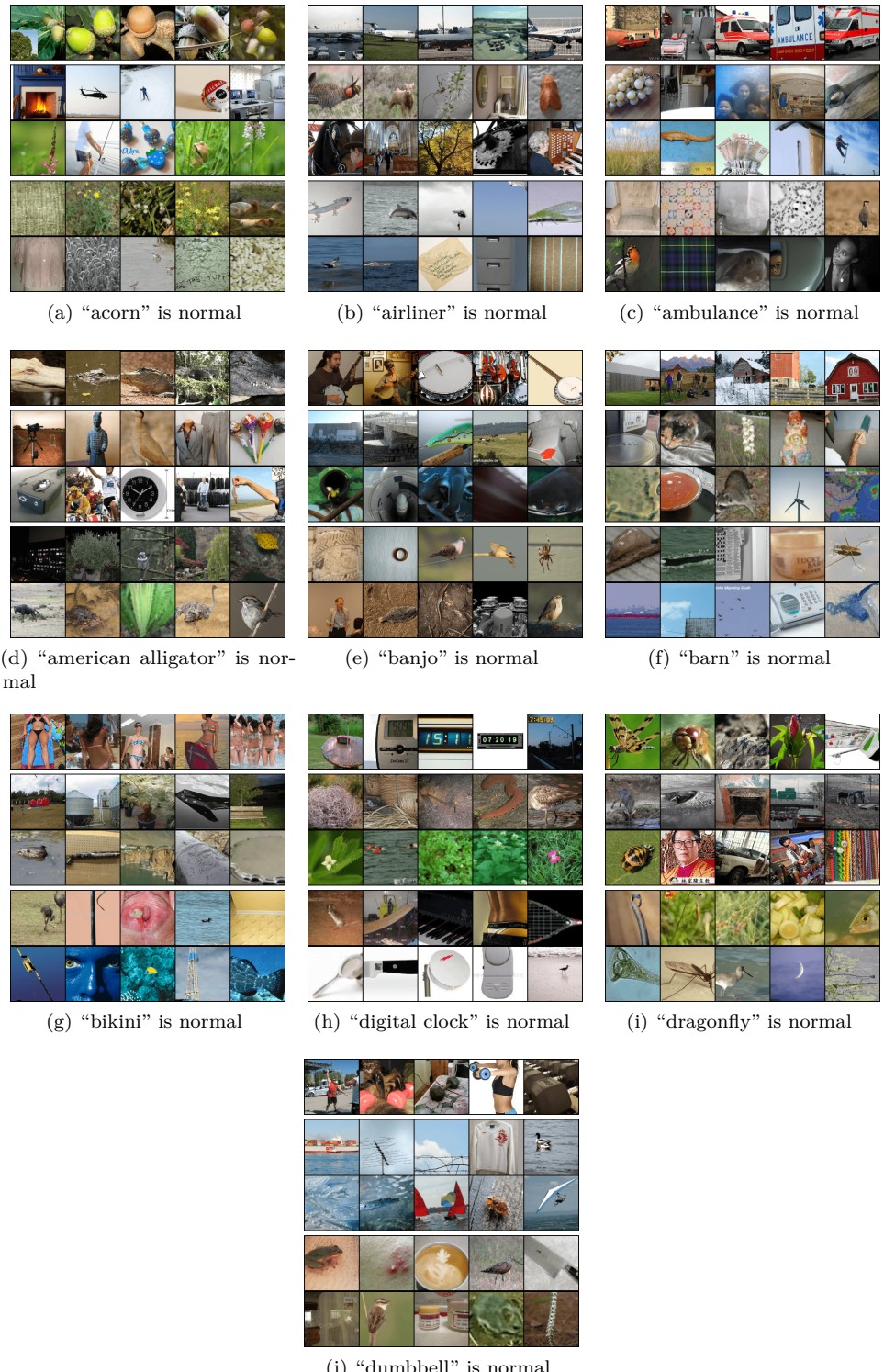

Figure 26: Optimal OE samples for ImageNet1k with ImageNet22k (with the 1K classes removed) as OE. The first row shows normal samples, the next two rows the best samples found via HSC (top) and BCE (bottom), and the last two rows the worst samples found via HSC (top) and BCE (bottom).

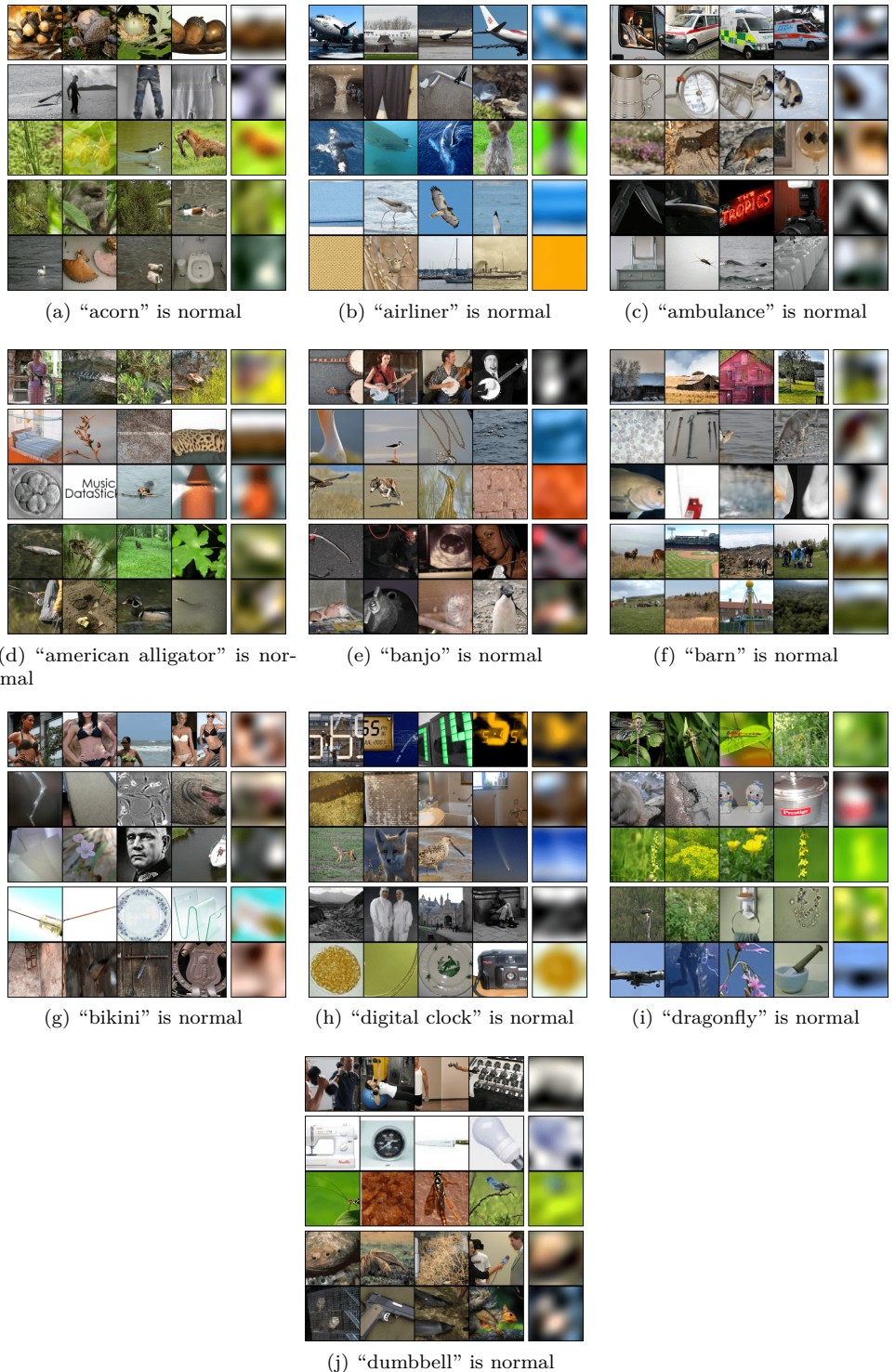

(a) "acorn" is normal

(b) "airliner" is normal

(c) "ambulance" is normal

(d) "american alligator" is normal

(e) "banjo" is normal

(f) "barn" is normal

(g) "bikini" is normal

(h) "digital clock" is normal

(i) "dragonfly" is normal

(j) "dumbbell" is normal

Figure 27: Optimal OE samples for low-pass-filtered ImageNet1k with low-pass-filtered ImageNet22k (with the 1K classes removed) as OE. The first row shows normal samples, the next two rows the best samples found via HSC (top) and BCE (bottom), and the last two rows the worst samples found via HSC (top) and BCE (bottom). The last column shows the low-pass-filtered version of the images, which is what the network sees during training and testing.

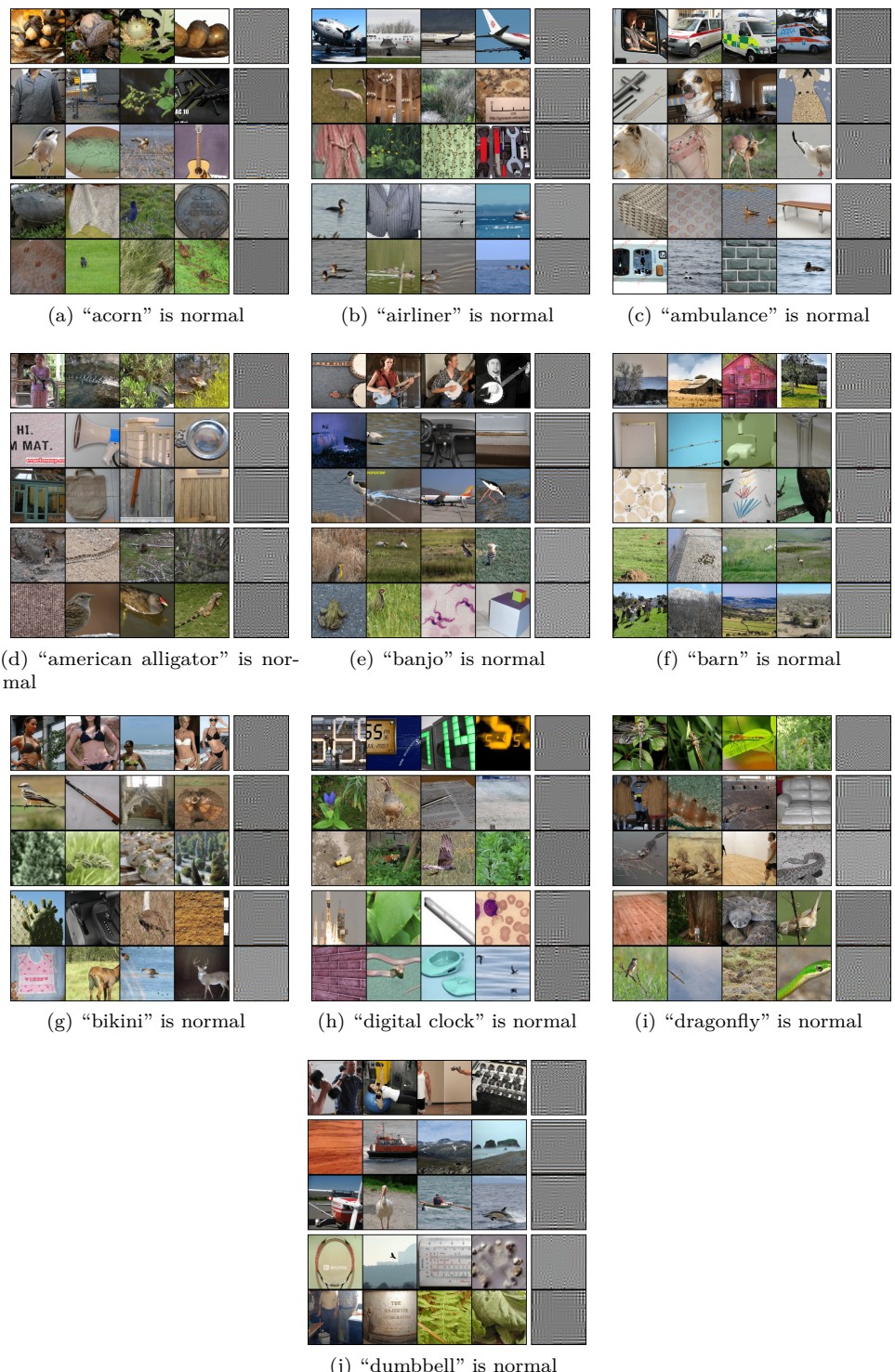

Figure 28: Optimal OE samples for high-pass-filtered ImageNet1k with high-pass-filtered ImageNet22k (with the 1K classes removed) as OE. The first row shows normal samples, the next two rows the best samples found via HSC (top) and BCE (bottom), and the last two rows the worst samples found via HSC (top) and BCE (bottom). The last column shows the high-pass-filtered version of the images, which is what the network sees during training and testing.

