# OpenReview forum: "Exposing Outlier Exposure: What Can Be Learned From Few, One, and Zero Outlier Images"
_TMLR — Accepted by TMLR_

### Review · Reviewer_tVkG · 2022-08-17

**Summary Of Contributions:**

This paper investigates outlier exposure (OE) for anomaly detection (AD) on two commonly used object classification datasets (Cifar10 and ImageNet). Three main settings are investigated: unsupervised, supervised and pretrained. Different OE hyparameters are tested: number and diversity of images. Additionally a zero-shot setting is evaluated which merely relies on the supervision of the normal class label. Results show that even small OE dataset can achieve strong performance and that standard AD losses eg. SVDD nearly do not help at all when OE are available.

New knowledge:
* While the knowledge that not many OE images are needed is not really new given Deecke et al. and Reiss et al., I am not aware of a precise investigation of the accuracy vs. number of images in OCC case.
* The exact interplay between SVDD + OE losses has not been sufficiently investigated before, although the results are expected.
* SOTA numbers for transfer OE using CLIP weights.
* I think the zero-shot CLIP experiment is new for the OCC setting (although some results were presented in Fort et al., NeurIPS'21)

**Broader Impact Concerns:**

Broader impact is sufficiently addressed

**Requested Changes:**

While the contributions of this work are modest - it adds more knowledge. None the less, a precise contributions statement with the claimed contributions should be added. The discussion above can serve as aid for their specification.

OE should be examined on other datasets beyond Cifar10 and ImageNet. In my experience it works far less effectively on other datasets and current papers give a misleading sense of success. There are many datasets that can be evaluated such as CUB, Dynamic textures, MVTec (already mentioned in the text as failuure case, but a proper experiment should be performed), F/MNIST etc. The leave-one-class-out setting should also be investigated as both SVDD and OE will be less effective there. This reviewer does not think an OE paper is acceptable without such an investigation.

This reviewer does not find the small number of images needed for OE surprizing - this is highly expected given the SOTA in few-shot learning and can probably even be improved further. I suggest removing the "surprise" from the text.

**Strengths And Weaknesses:**

This submission is very well written and gives a nice overview if the SOTA in the field. I found it a pleasure to read, and I believe that its background can serve newcomers to the field very well.

The scope of the contributions of this submission is somewhat modest - an empirical investigation of several aspects of OE. Still, it is able to disambiguate several aspects of current methods e.g. if unsupervised losses are also used, how well OE works without transferring external weights. Also the precise experiments for OE accuracy as a function of OE dataset diversity and size is informative.

Other advantages are the new SOTA number of OE with CLIP weights. I found the zero-shot CLIP results interesting - however it s limited scope should be acknowledged as it only applies when a single class name for the normal data exists and is known - which may not always be the case.

The submission has some weaknesses. I think the primary one is that results are only presented for Cifar10 and ImageNet when OE works well (as OE similar to the anomalies can be found). OE is less successful when the OE dataset is not similar to the anomalies - this is the main reason why it cannot be used in many real cases. This is only partially acknowledged and not investigated. In this reviewer's view, this is a critical limitation.

The unsupervised method used here (SVDD) is no longer the leading self-supervised AD method, it is conceivable that a combination of CSI/DROC with OE might give more favorable tradeoffs for the unsupervised setting. So the investigation is incomplete in this aspect.

Finally, the limited scope of the publication is a limitation but in my understanding of the TMLR guidelines, not a critical one.

---

> ### Author Response · Authors · 2022-09-14
> **Response to Reviewer tVkG - Part 1**
>
> Thank you for your constructive review. In the following, we address the main concerns.
>
>
> ### "While the knowledge that not many OE images are needed is not really new given Deecke et al. and Reiss et al., I am not aware of a precise investigation of the accuracy vs. number of images in OCC case."
>
> Reiss et al. and Deecke et al. both rely on transfer learning. We have demonstrated that this phenomenon is exhibited in general AD paradigms, which we think is a useful observation for the deep AD community.
> The methods presented in Deecke et al. and the OE-based method in Reiss et al. both utilize the full OE dataset (millions of images) so the observation that "not many OE images are needed" is novel to our knowledge.
>
> ### "OE should be examined on other datasets beyond Cifar10 and ImageNet. In my experience it works far less effectively on other datasets and current papers give a misleading sense of success. There are many datasets that can be evaluated such as CUB, Dynamic textures, MVTec (already mentioned in the text as failuure case, but a proper experiment should be performed), F/MNIST etc. The leave-one-class-out setting should also be investigated as both SVDD and OE will be less effective there. This reviewer does not think an OE paper is acceptable without such an investigation."
>
> We have performed several of the requested experiments and added the results to the manuscript, which we agree strengthen the contribution of the paper. We investigate the OE performance on MVTec-AD as well as the leave-one-class-out (LOCO) setting. Results using the full OE datasets are also included in the table below for convenience. Results for varying the amount OE for CIFAR-10 LOCO, ImageNet LOCO, and MVTec-AD can also be found in the updated manuscript (see Figures 3 and 4). We briefly discuss the results here.
>
> **MVTec-AD**: As hypothesized, using ImageNet22k as OE performs more poorly on MVTec-AD than specialized, state-of-the-art methods: HSC scores around 70% compared to [1], which achieve around 99% AuROC. Contrary to the experiments with CIFAR-10 and ImageNet-30, we observe that HSC performs a bit better than BCE, again revealing some robustness of HSC w.r.t. the choice of outliers, which are rather unfitting in this case. We also find that varying the size of the OE dataset for MVTec-AD does not have a significant effect.
>
> **CIFAR-10 LOCO and ImageNet LOCO full OE:** We observe that the various methods perform similarly in relation to one another as in the one-vs-rest tasks, thereby supporting our findings, albeit they perform slightly worse overall. This is presumably due to the more difficult problem setting since the normal data is more heterogeneous in the LOCO setting.
>
> **CIFAR-10 LOCO and ImageNet LOCO varying OE:** We find that more OE data is required in this setting compared to one vs. rest to achieve top performance, which again is likely due to the greater heterogeneity of the normal data. Otherwise, it still holds that BCE performs better.
>
> **Conclusion:** As hypothesized, there are situations where one requires a lot of OE data to perform well. Our core message, however, remains intact and we see strengthened by the added experiments: One does not benefit from an unsupervised approach when enough OE data is available. We have integrated the points above into the paper.
>
> |                 | DSVDD | GT+* | DSAD | HSC  | Focal* | Focal | BCE  | CLIP | BCE-CL |
> |-----------------|-------|------|------|------|--------|-------|------|------|--------|
> | MvTec-AD        | 64.0  | x    | 66.1 | 70.1 | x      | 65.4  | 66.1 | 56.4 | 76.2   |
> | CIFAR-10 LOCO   | 52.2  | x    | 84.2 | 84.8 | x      | 86.4  | 86.6 | 92.2 | 98.4   |
> | ImageNet-30 LOCO | 49.7     | x    | x    | 85.0 | x      | x     | 86.0 | 97.8    | x      |
>
> [1] Roth, K., Pemula, L., Zepeda, J., Schölkopf, B., Brox, T., & Gehler, P. (2022). Towards total recall in industrial anomaly detection. In Proceedings of the IEEE/CVF Conference on Computer Vision and Pattern Recognition (pp. 14318-14328).

---

> ### Author Response · Authors · 2022-09-14
> **Response to Reviewer tVkG - Part 2**
>
> ### "The unsupervised method used here (SVDD) is no longer the leading self-supervised AD method, it is conceivable that a combination of CSI/DROC with OE might give more favorable tradeoffs for the unsupervised setting. So the investigation is incomplete in this aspect."
> We are only considering methods that have an established OE/semi-supervised variant. Both DSVDD and GT+ have rather simple objectives with established OE variants.
>
> Combining CSI with OE is an intriguing idea that might yield the strongest performance but such a version of CSI has not been proposed and studied yet to our knowledge. Furthermore, CSI combines several different losses and we see no single obvious natural way to integrate OE.
>
> DROCC is essentially a method for generating OE synthetically by cyclically perturbing training samples in an adversarial manner to be anomalous and then using those samples to train a new binary classifier. There does not exist an OE variant for DROCC. We suppose the "natural" way to integrate OE would be to include the OE samples with the perturbed samples during training. This results in a method that is the standard BCE method in our submission with a type of one-sided reverse adversarial training (adversarially perturbed normal samples are trained as anomalies). In our opinion, the inclusion of this method would likely not be illuminating. Again, it might be possible that one could concoct an effective OE variant of DROCC, but developing such a method is outside of the scope of this paper.
>
> We mentioned these plausible OE extensions as possible future work.
>
> ### "While the contributions of this work are modest - it adds more knowledge. None the less, a precise contributions statement with the claimed contributions should be added. The discussion above can serve as aid for their specification."
>
> We revised the last two paragraphs of the introduction in the updated manuscript, which summarize the contributions of the paper. We copy them here for convenience.
>
> >In this paper we present surprising experimental results that challenge the assumption that deep AD on images
> needs an unsupervised approach (with or without OE). Using the same OE setup as Hendrycks et al. (2019b),
> which is common in the literature, we find that a standard classifier outperforms current state-of-the-art AD
> methods on the one vs. rest AD benchmark with CIFAR-10 and ImageNet. The one vs. rest benchmark
> has been recommended as a standard evaluation protocol to validate AD methods (Emmott et al., 2013)
> and is used as a litmus test in virtually all deep AD papers published at top-tier venues; see e.g. (Ruff
> et al., 2018; Deecke et al., 2018; Golan & El-Yaniv, 2018; Akcay et al., 2018; Hendrycks et al., 2019b; Abati
> et al., 2019; Perera et al., 2019; Wang et al., 2019a; Ruff et al., 2020; Bergman & Hoshen, 2020; Kim et al.,
> 2020; Liznerski et al., 2021; Deecke et al., 2021). Further challenging common assumptions, we find that OE
> does not seem to require huge amounts of data to represent “anomalousness.” A classifier requires only 256
> random OE samples to outperform the state of the art on ImageNet and only one well-chosen OE sample to
> score reasonably compared to unsupervised methods and classical AD approaches. This approach, however,
> does not solve all types of AD problems, in particular when the normal dataset is highly diverse or when
> anomalies are very subtle. For instance, we demonstrate that the methods need more OE samples to achieve
> top performance on the less common leave-one-class-out AD benchmark Bergman & Hoshen (2020); Deecke
> et al. (2021) where many classes are combined to form a multi-modal normal class. Further, on MVTec-AD, a recent manufacturing dataset, we show that random natural images are not very informative as OE. We also
> investigate transfer learning approaches to AD that have recently improved AD on images (Reiss et al., 2021;
> Deecke et al., 2021). Using CLIP (Radford et al., 2021), a recent foundation model, we find that it is possible
> to set a state of the art on CIFAR-10 and ImageNet without any additional training data. While transfer
> learning and standard classification work well, we still show advantages of unsupervised OE over supervised
> OE. When there are very few OE samples or the OE samples are not very informative, unsupervised OE
> approaches outperform classifiers, indicating a certain robustness with respect to the training outliers.
> In conclusion, the primary message of this paper is neither that we propose yet another state-of-the-art
> method nor that one of our investigated methodologies is of general superiority, but that there is a surprisingly
> strong performance of off-the-shelf classifiers, transfer learning, and just a few OE samples–contradicting
> widespread common assumptions for deep AD on well-established AD benchmarks. Through this work, we
> want to encourage rethinking how previous AD results extend to deep learning.

---

### Review · Reviewer_x73K · 2022-08-29

**Summary Of Contributions:**

The paper focuses on implementing Outlier Exposure (OE) in anomaly detection (AD) as a binary classification problem, and considers to cross-entropy based losses, namely BCE and HSC, where the latter adopts the radial basis kernel and turns out to be an extension of Deep-SVDD with negative (outlier) samples. With these losses, the paper observes that (a) this “supervised” version of OE can outperform OE, and (b) the full effectiveness of OE can be actually made with less (not much less in my opinion, though) outlier samples. Next, the paper considers a simple zero-shot AD method via CLIP, showing that this can achieve near-perfect performances on the one-class benchmarks of CIFAR-10 and ImageNet-30.

**Broader Impact Concerns:**

None that I am aware of.

**Requested Changes:**

Major requests

- I feel the experiment in Section 5.2 could be strengthen if it can provide results from the standard OE, i.e., results OE with fewer samples. If the proposed method can show better few-shot efficiency, it will be a good addition.
- Results on more datasets and diverse evaluation metrics would directly strengthen the paper.
- It is quite confusing that the paper conducts one-class experiments on ImageNet-30 or (the full) ImageNet-1k - although Section 4 says as the former (30), but the captions on the remaining tables says as the latter (1k). The paper should clarify that.
- Missing error bars in all the tables, while the paper states that all the experiments are averaged over 10 seeds.

Minor requests

- I think Section 3 (”Methods”) could be better organized - namely, as Preliminaries and Methods. For me, it is quite hard to extract technical novelty from the existing works given the current text.
- For the few-shot experiments, I think it would be interesting if the paper could explore the effect of data augmentation techniques to the detection performances.

**Strengths And Weaknesses:**

Strengths

- The paper is clearly written.
- Several observations made in the paper could be of interest to the community, e.g., the zero-shot AD performance of CLIP, and OE with fewer samples.

Weaknesses

- The paper proposes a new form of OE, but its novelty over Deep-SVDD is not actually delineated, as I feel it is quite a direct extension of Deep-SVDD. Also, I do not get why the proposed method is more “principled modification” over the previous approaches, that as mentioned in Section 3.2.
- The experiments are conducted with comparably narrow one-class benchmarks of CIFAR-10 and ImageNet-30, with a single major performance metrics of AUC. I generally feel that the claims made in the paper requires a more extensive evaluation: e.g., to support whether OE indeed needs fewer samples to improve semantic anomaly detection.
- The technical novelty of the method seems limited: for the supervised OE results (Section 5.1), the improvements of the proposed losses is quite marginal over the standard OE. In case of few-shot results (Section 5.2), I feel the results are not as dramatic as claimed - it still requires ~32K OE samples to reliably recover the full-shot results, and the few-shot results (I consider it as from 2^1 ~ 2^4) do not actually improve much from the zero-shot results. Also, the experiments do not verify whether this trends will persists on other OOD benchmarks.
- The results in Section 5.4 is somewhat inconclusive: nether HSC and BCE shows much robustness on varying OE samples - for such a claim, I personally think the evaluation benchmark should need more diversity and thoroughness.

---

> ### Author Response · Authors · 2022-09-14
> **Response to Reviewer x73K**
>
> Thank you for your review and suggestions. In the following we address the main concerns.
>
> ### Lack of novelty in proposed methods. "The paper proposes a new form of OE, but its novelty over Deep-SVDD is not actually delineated, as I feel it is quite a direct extension of Deep-SVDD." "The technical novelty of the method seems limited: for the supervised OE results (Section 5.1), the improvements of the proposed losses is quite marginal over the standard OE..."
>
> We agree that our main contribution is not technical (though we introduce the HSC loss, derived from using a radial kernel with BCE). Instead, we see our main contribution in the extensive empirical evaluation and new insights about OE drawn from it.
> We aimed to address this concern in the last paragraph of the introduction of the paper:
> >In conclusion, the primary message of this paper is neither that we propose yet another state-of-the-art
> method nor that one of our investigated methodologies is of general superiority, but that there is a surprisingly
> strong performance of off-the-shelf classifiers, transfer learning, and just a few OE samples–contradicting
> widespread common assumptions for deep AD on well-established AD benchmarks. Through this work, we
> want to encourage rethinking how previous AD results extend to deep learning.
>
> ### Lack of evaluation: "The experiments are conducted with comparably narrow one-class benchmarks of CIFAR-10 and ImageNet-30, with a single major performance metrics of AUC." "Results on more datasets and diverse evaluation metrics would directly strengthen the paper."
>
> Following your and Reviewer tVkG's concerns, we have performed additional experiments. We have included experiments for MVTec-AD and the leave-one-class-out setting for CIFAR-10 and ImageNet. The new results are presented in the updated manuscript and in the response to reviewer tVkG.
>
> Regarding metrics, we find that the AUC (AuROC) metric is overwhelmingly the go-to metric for evaluating deep anomaly detection (see examples Ruff et al., 2018; Deecke et al., 2018; Golan & El-Yaniv, 2018; Akcay et al., 2018; Hendrycks et al., 2019b; Abati et al., 2019; Perera et al., 2019; Wang et al., 2019a; Ruff et al., 2020; Bergman & Hoshen, 2020; Kim et al., 2020; Liznerski et al., 2021; Deecke et al., 2021). For comparability with previous works (as we include some results from the literature), we kept this metric to report.
>
> ### Few-shot results: "The results in Section 5.4 is somewhat inconclusive: nether HSC and BCE shows much robustness on varying OE samples - for such a claim, I personally think the evaluation benchmark should need more diversity and thoroughness."
>
> Our intention here was to illustrate that HSC is more robust than BCE. The beginning of Section 5.4 states:
>
> >Our previous experiments (Section 5.2) have shown that, though end-to-end BCE overall outperforms HSC, an unsupervised OE approach is more effective when only very few (< 32) OE samples are available. This indicates a certain degree of robustness to the anomalous training samples for HSC.
>
> Later in Section 5.4, we state that:
>
> >On both datasets, we observe that HSC performs better than BCE when using both the best
> and the worst OE samples.
>
> The added experiments further strengthen this conclusion (see MVTec) that HSC seems to be consistently better and more robust than BCE in all limited OE settings.
> We hope this clarification and the added evidence alleviates your concern.
>
> ### ImageNet-30 and ImageNet-1k: "It is quite confusing that the paper conducts one-class experiments on ImageNet-30 or (the full) ImageNet-1k - although Section 4 says as the former (30), but the captions on the remaining tables says as the latter (1k). The paper should clarify that."
> Sorry for the confusion. All the experiments use ImageNet-30. We fixed this typo in the updated manuscript.
>
> ### Error bars: "Missing error bars in all the tables, while the paper states that all the experiments are averaged over 10 seeds."
> Due to space constraints, we report these in the Appendix.

---

### Review · Reviewer_Lp9d · 2022-09-01

**Summary Of Contributions:**

This paper shows very interesting results that challenge common assumptions in deep anomaly detection. They found only a small number of Outlier exposure data is enough to outperform all end-to-end methods on common AD benchmarks. By leveraging transfer learning techniques, they propose zero-shot learning with CLIP methods and show superior performance on imagenet.  In addition, they found that semi-supervised one-class methods are more robust to the choice of OE when only few OE data are available.

**Broader Impact Concerns:**

I did not find broader impact concerns

**Requested Changes:**

Overall, the paper is well-written and easy to follow. I am glad to read these interesting findings and recommend to accept.

In the current stage, I have no big concerns to change but it would be helpful if the authors can well address the weakness I listed above.

**Strengths And Weaknesses:**

Strength

1. The paper poses a very interesting angle about AD problems and several common assumptions are contradicted.
2. They provide surprising results that outperform the SOTA benchmarks by using only few OE examples, which sounds exciting.
3. The paper offers a detailed discussion about the motivation, challenges, and current SOTA works, which is very helpful to follow their ideas.

Weakness
1. The empirical studies are strong enough but may lack some theoretical insights, and it will be great if the authors can provide additional explanations why the results look so surprising either from the theoretical side or intuitive side.
2. The motivation for why you use CLIP is not clear. From the current version, just inspired by the previous work [Fort 2021]?
3. The AD assumption in real-world applications.  In fact, OE examples are not easy to get in practice since we probably need additional efforts to define normal and anomaly. Unlike imagenet, we are easy to define which samples are normal or not but if we do not know, does the method still works well?   It involves the basic assumption of AD setting but we may have many challenges in defining OE in real-world applications even though only few samples are needed as the author mentioned.

---

> ### Author Response · Authors · 2022-09-14
> **Response to Reviewer Lp9d - Part 1**
>
> Thank you for your positive review. We engage in further discussion below.
>
> ### "The empirical studies are strong enough but may lack some theoretical insights, and it will be great if the authors can provide additional explanations why the results look so surprising either from the theoretical side or intuitive side."
>
> We agree and have made some attempts to find explanations for the results we see. The frequency analysis we mention in Section 5.4 was an attempt understanding the difference between HSC and BCE, in particular the robustness we observe in Section 5.4. The results of this analysis can be found in Appendices B and C. One can possibly conclude from these experiments that HSC makes better use of low-frequency features which generalizes better, but we don't think the results here are conclusive enough to assert that point. From the collection of "best" (Figure 5 in the current draft, Figure 3 in the original submission) OE samples we can see that the most informative images for HSC for a fixed normal class have a somewhat consistent appearance, especially with regards to color, whereas BCE does not. We think that this might be because HSC is indeed providing a *slight* useful bias for the problem. From the paper:
>
> >This is likely due to the fact that HSC has, in some sense, an initial notion
> of anomalousness due to its unsupervised term. For instance, the most useful OE samples are those not already contained in this notion of anomalousness, resulting in HSC having a stable region for selecting OE samples that yield the greatest improvement. BCE lacks this notion so it can benefit from a large variety of OE samples. This is also supported by Figure 2.
>
> Our guess is that the intuition from Figure 1 (d) actually *does* hold, but only for an exceptionally limited regime with very little OE data, but then the OE data in some sense fills the space as in Figure 1 (d) rather quickly. The quote above alludes to this point as does the following in Section 5.2:
>
> > With sufficiently few samples, HSC outperforms BCE, which seems to indicate a regime where the unsupervised OE approach is advantageous. Interestingly, this regime seems to be quite small...
>
> We agree that a more conclusive explanation would be nice. Hopefully some of the observations we've made in this paper will assist future research in getting to the bottom of this.
>
> ### "The motivation for why you use CLIP is not clear. From the current version, just inspired by the previous work [Fort 2021]? "
> Our paper studies the sample utilization for deep AD. It mostly focuses on the OE samples, but, given the strong performance of recent methods that use transfer learning, we were curious to see how few samples a strong foundation model would require to outperform end-to-end methods. We found that it does not only need no OE samples, but an appropriate pre-trained model even requires no samples at all. We conducted these experiments before we became aware of Fort et al., so perhaps one may consider these as contemporaneous findings.

---

> ### Author Response · Authors · 2022-09-14
> **Response to Reviewer Lp9d - Part 2**
>
> ### "The AD assumption in real-world applications. In fact, OE examples are not easy to get in practice since we probably need additional efforts to define normal and anomaly. Unlike imagenet, we are easy to define which samples are normal or not but if we do not know, does the method still works well? It involves the basic assumption of AD setting but we may have many challenges in defining OE in real-world applications even though only few samples are needed as the author mentioned."
> The OE approach with random natural images works best in settings where the anomalies are on a semantic level (e.g., cat vs. dog etc.) rather than on a low (pixel) level (e.g., hazelnut vs. scratched hazelnut etc.). More details on this differentiation can be found in [1]. Therefore,  we expected the methods to not work well on MVTec-AD, which is a manufacturing datasets with local defects as anomalies. In the updated manuscript, we also empirically validate this hypothesis; see the additional experiments in the updated manuscript. However, for such low-level AD scenarios one might try to generate synthetic outliers and use those during training. For example, [2] used a so-called "confetti noise", [3] used a "cutpaste" algorithm, and very recently [4] used diffusion models to generate synthetic training outliers for MVTec-AD, achieving quite impressive detection results. The main idea is to perturbate the normal samples until they become anomalous (but are still close to the in-distribution) and can serve as negative examples.
>
> Another problematic case for OE is when the normal distribution is very diverse, which is a rather atypical scenario, but there are certainly situations where this occurs (e.g., in patient variability in medicine). The AD model requires a larger amount of OE samples to successfully detect anomalies. We investigated this by following the suggestions of reviewer tVkG. We evaluated the methods on the leave-one-class-out setting, where all classes but one are normal. The results can be found in the updated manuscript. The intuition here is that the more diverse the normal distribution is, which is defined via the given training set, the more OE samples are required.
>
>
> [1] Ruff, L., Kauffmann, J. R., Vandermeulen, R. A., Montavon, G., Samek, W., Kloft, M., ... & Müller, K. R. (2021). A unifying review of deep and shallow anomaly detection. Proceedings of the IEEE, 109(5), 756-795.
>
> [2] Liznerski, P., Ruff, L., Vandermeulen, R. A., Franks, B. J., Kloft, M., & Müller, K. R. (2020). Explainable deep one-class classification. arXiv preprint arXiv:2007.01760.
>
> [3] Li, C. L., Sohn, K., Yoon, J., & Pfister, T. (2021). Cutpaste: Self-supervised learning for anomaly detection and localization. In Proceedings of the IEEE/CVF Conference on Computer Vision and Pattern Recognition (pp. 9664-9674).
>
> [4] Mirzaei, H., Salehi, M., Shahabi, S., Gavves, E., Snoek, C. G., Sabokrou, M., & Rohban, M. H. (2022). Fake It Till You Make It: Near-Distribution Novelty Detection by Score-Based Generative Models. arXiv preprint arXiv:2205.14297.

---

### Decision · Action_Editors · 2022-10-03

**Recommendation:** Accept with minor revision

**Comment:**

The paper received one Accept (Reviewer Lp9d), one Leaning Accept (Reviewer tVkG), and one Leaning Reject (Reviewer x73K). Reviewer x73K's main concerns are related to the technical novelty of the paper. The authors addressed these concerns in their answer, but did not manage to convince the reviewer.

Nevertheless, all reviewers (including Reviewer x73K) acknowledge that some of the observations made by the authors in their study were unknown in the community and are of interest. The reviewers also acknowledge that the paper is very well written and that the experimental evaluation is mostly solid.

As mentioned before, there is however and exception to this, with Reviewer tVkG requesting additional results on datasets mentioned in their review for the readers to get (quoting the reviewer) "a better sense of the (limited) success of OE". We believe this to be a valid suggestion and therefore recommend the authors to follow it for the final version of the paper.

**Audience:**

Yes, this paper will interest at least a portion of the TMLR's audience.

**Claims And Evidence:**

Yes, the claims made in the submission are supported by clear evidence, with one exception noted by Reviewer tVkG. Below is an excerpt from Reviewer tVkG's final recommendation (which I am not sure the authors can see):

"The paper still gives a somewhat misleading sense of the effectiveness of OE. While new results were presented on the LOCO setting and the MVTec dataset, which are good step in the right direction, I still think that showing several more datasets will give a better sense of the (limited) success of OE. This experiment need not take much time or compute. Multiple suggested datasets were included in my review."